# Aligning the Brain with Language Models Through a Nonlinear and Multimodal Approach

## Abstract

Self-supervised language and audio models effectively predict brain responses to speech. However, while nonlinear approaches have become standard in vision encoding, speech encoding models still predominantly rely on linear mappings from unimodal features. This linear approach fails to capture the complex integration of auditory signals with linguistic information across widespread brain networks during speech comprehension. Here, we introduce a nonlinear, multimodal prediction model that combines audio and linguistic features from pre-trained models (e.g., Llama, Whisper). Our approach achieves a 17.2% and 17.9% improvement in prediction performance (unnormalized and normalized correlation) over traditional unimodal linear models, as well as a 7.7% and 14.4% improvement over prior state-of-the-art models relying on weighted averaging of linear unimodal predictions. These substantial improvements not only represent a major step towards future robust in-silico testing and improved decoding performance, but also reveal distributed multimodal processing patterns across the cortex that aligns with key neurolinguistic theories including the Motor Theory of Speech Perception, Convergence-Divergence Zone model, and embodied semantics. Overall, our work highlights the often neglected potential of nonlinear and multimodal approaches to speech encoding, paving the way for future studies to embrace these strategies in naturalistic neurolinguistics research.

## 1 Introduction

Speech encoding models, which predict voxel-wise cortical activity from naturalistic speech, are a powerful tool for probing the neural processes of speech comprehension (Naselaris et al., 2011; Jain & Huth, 2018; LeBel et al., 2021; Vaidya et al., 2022; Goldstein et al., 2022; Tang et al., 2023). They also enable important applications such as in-silico experiments to test brain function without additional data (Wehbe et al., 2016; Bashivan et al., 2019; Jain et al., 2024) and the development of decoding models for language comprehension (Tang et al., 2023).

Most existing approaches rely on unimodal linearized models, where features from language (e.g., Llama; Touvron et al. (2023a)) or speech models (e.g., Whisper; Radford et al. (2023)) are linearly mapped to brain activity(Naselaris et al., 2011). Linearized models is efficient, work well with limited neuroscience datasets, and allow straightforward feature attribution. However, with the advent of larger datasets(LeBel et al., 2023), more sophisticated modeling approaches are now feasible, offering the potential to uncover new insights into neural speech processing.

One key direction is to capture the inherently multimodal nature of speech comprehension. The brain integrates acoustic, linguistic, and motor information across distributed neural networks (McGettigan et al., 2012; Ghazanfar & Schroeder, 2006). While some studies have combined linguistic and visual features (Oota et al., 2022; Wang et al., 2022; Scotti et al., 2024), the integration of advanced speech and language models remains largely unexplored. Recent work (Oota et al., 2023) shows that speech models uniquely capture activity in early auditory regions, while text-based models better explain late language regions—suggesting that their combination could yield richer insights into neural language processing.

A second direction is to use nonlinear mappings between model features and brain activity. Cortical and language computations are inherently nonlinear, with interactions across features, time, and modalities (Deco et al., 2008; Beniaguev et al., 2021; Tuller et al., 2011; McGettigan et al., 2012). As Ivanova et al. (2022) argue, nonlinear mappings are often more appropriate for key neuroscientific goals such as in-silico experimentation, testing feature relevance, and understanding overall feature-set contributions. By relaxing the linearity assumption, these models can reveal functional organization patterns that remain hidden under linear readouts and can substantially improve prediction accuracy—both of which are critical for robust in-silico testing (Jain et al., 2024).

Despite this motivation, nonlinear encoding remains rare in speech fMRI. Existing nonlinear work in language has largely been restricted to simplified paradigms with isolated words or short phrases (Bingel et al., 2016; Oota et al., 2018), or to *unimodal* features (Moussa et al., 2024; Vattikonda et al., 2025), rather than naturalistic continuous speech with multimodal inputs. In contrast, nonlinear models have become standard in vision (Scotti et al., 2024; Yang et al., 2023; Chen et al., 2023), leveraging them to probe representational nonlinearities in visual cortex (Yang et al., 2023; Chen et al., 2023; Scotti et al., 2024). Nonlinear approaches also face unique challenges in speech encoding compared to vision (Appendix N) Unlike visual models, which operate over approximately 15k cortical voxels, speech models must predict substantially larger neural activation patterns—on the order of 80k–90k voxels—while simultaneously capturing rapid, fine-grained temporal dynamics inherent to continuous speech (LeBel et al., 2023), rather than block-wise visual paradigms (Allen et al., 2022). As a result, most speech encoding work has remained linear, even though analogous nonlinear interactions—especially across modalities—are likely to be equally important for language.

In this study, we address these gaps by introducing a nonlinear, multimodal encoding model that integrates audio and semantic features extracted from advanced models such as Whisper and Llama. Our contributions are as follows:

- **We demonstrate, for the first time, that nonlinear *multimodal* encoding is feasible for naturalistic speech and that it can uncover subtle cortical patterns traditional methods miss.** With only PCA and a single-hidden-layer MLP, our nonlinear multimodal encoder improves prediction accuracy by 17.2% (unnormalized) and 17.9% (normalized) over the standard semantic linear baseline (Antonello et al., 2024), and outperforms prior state-of-the-art linear ensembles by 7.7% and 14.4%. Such unusually large improvements for fMRI speech encoding (Appendix N.2) suggest that current linear, unimodal practice leaves a substantial amount of structured, explainable variance—particularly nonlinear multimodal interactions—on the table.
- **Through systematic comparisons, we show nonlinear multimodal interactions drives these improvements.** Linear models fail to capture the complex interactions between audio and language information in LLM embeddings, whereas our nonlinear encoders model these interactions more effectively and with fewer parameters. This demonstrates that incorporating both **nonlinearity** *and* **multimodality** is crucial for accurately modeling the brain's speech processing mechanisms.
- **We introduce a RED-based clustering analysis that tracks neural responses over both space and time.** Nonlinear models achieve superior functional clustering compared to linear encoders and standard connectivity analysis, revealing previously hidden patterns of brain organization and the spatiotemporal dynamics of language processing.
- **Variance partitioning and prediction accuracy analysis show that multimodal integration is essential for speech encoding.** Most regions rely on overlapping audio–semantic information, with unique contributions varying hierarchically from sensory to higher-order areas. This results extends neurolinguistic theories (Liberman et al., 1967; Damasio, 1989; Davis & Yee, 2021) by revealing how different brain regions jointly engage multiple aspects of speech input.

## 2 METHOD

### 2.1 MRI DATA

We used a public fMRI dataset (LeBel et al., 2023) of three subjects listening to 20 hours of English podcast. Training data included 95 stories across 20 scanning sessions (33,000 time points). Testing

used three held-out stories: one averaged across ten repetitions and two across five repetitions each, with no session containing repeated stimuli. Voxels were normalized to zero mean and unit variance, as in Antonello et al. (2024).

## 2.2 FEATURE EXTRACTION

A brain encoding model predicts voxel-wise fMRI responses from stimulus features, providing a framework to study how the brain represents language. In our study, the encoding model takes as input semantic features from LLaMA and audio features from Whisper, enabling us to test how linguistic and acoustic information jointly explain cortical activity.

We extracted semantic features from LLaMA models (LLaMA-1: 7B–65B (Touvron et al., 2023a); LLaMA-2: 7B (Touvron et al., 2023b); LLaMA-3: 8B (Dubey et al., 2024)) and audio features from Whisper models (Tiny–Large, including v2/v3; (Radford et al., 2023)). All models were obtained from Hugging Face (Wolf, 2019) and run in half-precision (float16). LLaMA features were obtained using a dynamically sized context window, while Whisper features were extracted from the encoder using a 16s sliding window with 0.1s stride, ensuring audio-specific representations. Refer to Antonello et al. (2024) for further details.

Following prior work (Antonello et al., 2024; Tang et al., 2023), we temporally aligned the hidden states from the $l^{\text{th}}$ layer of the language or audio models with fMRI acquisition times using Lanczos interpolation. To account for neural response delays, we concatenated representations from the four preceding timepoints (2, 4, 6, and 8 seconds prior) for each TR,

## 2.3 REPRESENTATIONS FOR FMRI DATA

The encoding model's outputs correspond directly to voxel-level fMRI activity. We tested both full-voxel prediction and dimensionality reduction, adopting PCA (512 components) for most analyses to prevent overfitting, reduce redundancy, and maintain interpretability. Direct full-voxel mapping is computationally prohibitive (e.g., 1.3B parameters for S1 vs. 8.4M with PCA) and redundant, as many voxels are highly correlated and can be masked with minimal loss (Jabakhanji et al., 2022; Lin et al., 2022). PCA also enables reconstruction of predicted responses back into voxel space, preserving neuroscientific interpretability. Formally, PCA was applied to the aggregate response matrix $Y_{\text{org}} \in \mathbb{R}^{N_{\text{TR}} \times N_{\text{voxels}}}$ to obtain $Y_{\text{PCA}} \in \mathbb{R}^{N_{\text{TR}} \times 512}$, and predictions $\hat{Y}_{\text{PCA}}^{\text{test}}$ were inverse-projected to voxel space for evaluation against ground-truth $Y^{\text{test}}$. Further details are provided in Appendix B.4.

## 2.4 ENCODING MODEL

Going beyond linear approaches (Tang et al., 2023; Huth et al., 2016; de Heer et al., 2017; LeBel et al., 2021; Jain & Huth, 2018; Schrimpf et al., 2021) we systematically investigate a range of encoding models varying in complexity and input modality to better capture complex relationships between stimuli and neural responses. We explored combinations of different stimulus representations, encoder architectures, and response representations (see Table 1). The following encoder architectures were used to assess the impact of complexity and nonlinearity (see Appendix B.5):

- *Linear Regression (Linear):* Following Antonello et al. (2024), we used ridge regression.
- *Multi-Layer Perceptron (MLP):* MLP with a single hidden layer of 256 units.
- *Multi-Layer Linear (MLLinear):* MLP but without dropout, batch normalization, and with the identity activation function. This model serves as a reduced-rank linear regression, helping to isolate the effects of dimensionality reduction from nonlinearity.
- *Delayed Interaction MLP (DIMLP):* Used for multimodal cases, this MLP variant processes each modality through separate 256-unit hidden layers before concatenation and final linear projection. This allows nonlinear processing within each modality while limiting cross-modal interaction to be linear, revealing the effects of nonlinear fusion of modalities.

## 2.5 NORMALIZED CORRELATION COEFFICIENT AND RELATIVE ERROR DIFFERENCE (RED)

Because fMRI data are inherently noisy, there exists a theoretical upper bound on explainable variance, known as the noise ceiling. We estimated this ceiling ($CC_{\text{max}}$) for each voxel using the method

Table 1: Performance of encoding models across modalities and architectures. We report average voxelwise $r^2$ and normalized correlation coefficient ($CC_{norm}$) for models using *text* inputs (from LLaMA-1), *audio* inputs (from Whisper-v1 Large), or *multimodal* inputs, paired with different encoder architectures (*Linear, MLLinear, DIMLP, MLP*). *MLLinear* is a linearized version of *MLP*, while *DIMLP* applies nonlinear processing within each modality but combines modalities linearly. The baseline is the semantic linear model in Antonello et al. (2024). Notably, *MLP* encoders consistently achieve the best performance with fewer parameters, underscoring the importance of nonlinearity and multimodal integration for accurate fMRI prediction. $r^2$ is computed as $|r| \cdot r$, and statistical significance analysis can be found in Appendix C

| modality 1 | modality 2 | encoder | response | Avg $r^2$ | Avg $CC_{norm}$ | #param |
|---|---|---|---|---|---|---|
| text | audio | MLP | PCA | **4.29% (+17.2%)** | **34.32% (+17.9%)** | 5.64M |
| text | audio | DIMLP | PCA | 4.18% (+14.2%) | 32.59% (+11.9%) | 5.77M |
| text | audio | MLLinear | PCA | 4.10% (+12.0%) | 32.41% (+11.3%) | 5.64M |
| text | audio | Linear | all voxels | 4.10% (+12.0%) | 31.36% (+7.7%) | 1.72B |
| text | audio | Linear | PCA | 3.87% (+5.7%) | 28.92% (-0.7%) | 11.01M |
| text | audio | MLP | all voxels | 3.83% (+4.6%) | 31.11% (+6.8%) | 26.07M |
| text | - | MLP | PCA | 3.79% (+3.6%) | 30.89% (+6.1%) | 4.33M |
| text | - | MLLinear | PCA | 3.67% (+0.3%) | 29.95% (+2.8%) | 4.33M |
| text | - | Linear | all voxels | 3.66% (Baseline) | 29.12% (Baseline) | 1.31B |
| text | - | Linear | PCA | 3.56% (-2.7%) | 26.88% (-7.7%) | 8.39M |
| text | - | MLP | all voxels | 3.36% (-8.2%) | 27.45% (-5.7%) | 24.75M |
| audio | - | MLP | PCA | 3.01% (-17.8%) | 29.01% (-0.4%) | 1.44M |
| audio | - | MLP | all voxels | 2.89% (-21.0%) | 28.21% (-3.1%) | 21.87M |
| audio | - | MLLinear | PCA | 2.89% (-21.0%) | 27.50% (-5.6%) | 1.44M |
| audio | - | Linear | PCA | 2.81% (-23.2%) | 26.71% (-8.3%) | 2.62M |
| audio | - | Linear | all voxels | 2.77% (-24.3%) | 25.20% (-13.5%) | 409.68M |

of Schoppe et al. (2016) applied to ten repeated responses to the same test story (Appendix B.2). Model performance was then normalized by dividing the absolute correlation coefficient ($CC_{abs}$, correlation between predicted and observed fMRI signals) by $CC_{max}$, yielding the normalized correlation coefficient ($CC_{norm}$). With 80,000 voxels, random noise can occasionally produce $CC_{abs}$ <$CC_{max}$, resulting in $CC_{norm}$ >1; to mitigate this, voxels with $CC_{max}$ <0.25 were regularized to 0.25 during computation.

To complement correlation-based metrics, we introduce the Relative Error Difference (RED), which quantifies the temporal advantage of one feature set over another. For each voxel $v$ at time $t$: $RED(v, t) = |f_1(v, t) - y(v, t)| - |f_2(v, t) - y(v, t)|$ where $f_1(v, t)$ and $f_2(v, t)$ are predictions from two feature sets (e.g., LLaMA vs. Whisper) and $y(v, t)$ is the ground-truth fMRI signal. Positive RED values indicate better prediction by feature set 2. Unlike traditional voxel-wise analyses that focus on spatial patterns ($f(v)$), RED preserves temporal dynamics ($f(v, t)$), enabling the joint analysis of spatial and temporal organization of brain responses. We leverage RED in Section 3.1.2 to cluster regions of interest based on semantic and audio processing dynamics.

## 3 RESULTS

We conduct experiments to evaluate the contributions of multimodality and nonlinearity in fMRI speech encoding. The primary objective is to determine whether nonlinear integration of audio and language representations provides measurable improvements over both the baseline (Antonello et al., 2024) and alternative encoding architectures. Model performance is assessed using variance explained ($r^2$) and normalized correlation coefficient ($CC_{norm}$), as in prior works.

Table 1 summarizes the overall comparison. The nonlinear multimodal MLP encoder achieves the highest performance, with 4.29% average $r^2$ and 34.32% $CC_{norm}$, corresponding to relative gains of 17.2% and and 17.9% over the baseline semantic linear model (Antonello et al., 2024). Notably, these improvements substantially exceed the incremental advances typically reported in fMRI speech encoding (Appendix N.2), despite using far fewer parameters (5.64M vs. 1.31B). The results suggest that additive linear fusion fails to capture complex audio–language interactions, underscor-

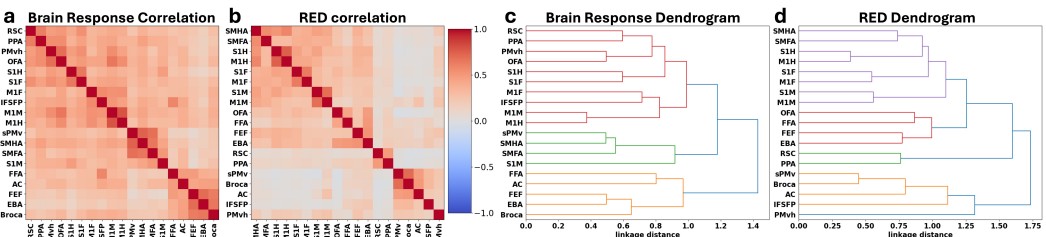

Figure 1: Spatio-temporal clustering analysis: **(a,b)** functional connectivity matrix and hierarchical clustering dendrogram from raw fMRI correlations. **(c,d)** Correlation matrices and dendrograms from Relative Error Difference (RED) between semantic and audio MLP encoders. Matrix values indicate regional similarity. Hierarchical clustering reveals brain region organization by response profiles. The nonlinear models **(d)** show clearer functional groupings than both linear models (modularity Q : 0.155 vs. 0.145) (Figure 23 **e**) and standard connectivity **(c)** (0.068). See Appendix A for full abbreviation names.

ing the value of nonlinear multimodal modeling. In the following subsections, we analyze the specific contributions of nonlinearity (Section 3.1), investigate how nonlinear multimodal combination drives improvements (Section 3.2) and demonstrate the benefits of multimodal fusion (Section 3.3).

## 3.1 NONLINEAR ENCODERS

### 3.1.1 NONLINEARITY IS THE KEY DRIVER OF SUPERIOR ENCODING PERFORMANCE

We found that the MLP consistently outperformed linear models, indicating that nonlinear transformations more effectively capture the mapping between neural activity and linguistic or acoustic features. To disentangle the role of nonlinearity from dimensionality reduction, we compared the MLP with two controls: *Linear* (linear regression on PCA-reduced data) and *MLLinear* (an MLP without nonlinear activations). Both performed similarly to or worse than the nonlinear MLP (Table 1), confirming that performance gains are driven by nonlinearity rather than reduced dimensionality.

Moreover, MLPs provided a clear and consistent advantage over linear encoders across all layers of both language and audio models (Figure 16). This layer-wise robustness underscores that nonlinear mapping captures meaningful representational structure regardless of depth. PCA preprocessing was nonetheless essential: MLPs trained directly on raw voxels performed substantially worse, likely due to overfitting (80–90k voxels vs. 512 PCA components). Together, these results demonstrate that while dimensionality reduction enables tractable modeling, it is nonlinearity that fundamentally drives superior encoding performance.

### 3.1.2 NONLINEARITY ENHANCES BRAIN-WIDE PREDICTIONS AND FUNCTIONAL CLUSTERING

Nonlinear MLP models capture complex relationships in brain activity during speech comprehension more effectively than linear models. As shown in Figure 1, MLP encoders outperform linear encoders across the cortex, with pronounced gains in semantic and auditory regions such as the precuneus (PrCu) and lateral temporal cortex (LTC). Brain maps in Appendix J.2 and J.3 further confirm these improvements, underscoring the critical role of nonlinear interactions in modeling brain activity, particularly in higher-order language processing areas.

Hierarchical clustering analysis using RED between Whisper and LLaMA encoding models (Figure 1, Appendix J.4) reinforces this advantage. Compared to linear models and traditional functional connectivity, nonlinear encoders achieve superior grouping (modularity $Q$: nonlinear 0.155, linear 0.145, FC 0.068). The MLP-based clustering (Figure 1 **d**) reveals coherent functional organization: motor and somatosensory regions cluster by body part before merging into broader networks; visual regions organize by function (OFA/FFA for faces; PPA/RSC for scenes); and speech-related areas (sPMv, Broca's area, AC) align with the dorsal stream pathway. These results show that nonlinear

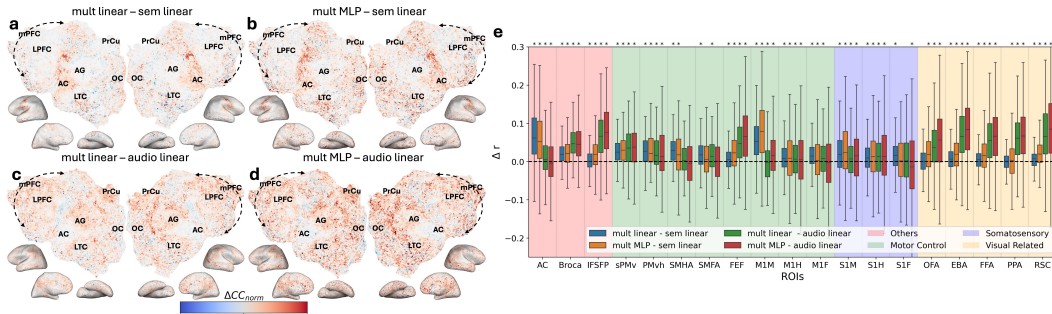

Figure 2: Multimodality improves encoding performance. Panels **(a–d)** show voxelwise $\Delta CC_{norm}$ for one subject, with warmer colors indicating regions where multimodal models outperform unimodal linear models. **(a)** Multimodal linear – Semantic linear: impact of adding audio features. **(b)** Multimodal MLP – Semantic MLP: impact of adding audio features with nonlinearity. **(c)** Multimodal linear – Audio linear: impact of adding semantic features. **(d)** Multimodal MLP – Audio MLP: impact of adding semantic features with nonlinearity. **(e)** ROI-level $\Delta r$ across all subjects, with significant improvements marked by asterisks (*, $p < 0.05$, FDR-corrected). Overall, multimodality yields widespread benefits across voxels and ROIs, with only a small minority showing reduced predictions.

models capture structured spatiotemporal relationships in brain responses, consistent with established principles of cortical organization.

## 3.2 NONLINEAR AND MULTIMODAL ENCODERS

### 3.2.1 NONLINEAR INTERACTIONS BETWEEN MODALITIES ENHANCE fMRI PREDICTIONS

To assess the role of nonlinear cross-modal interactions, we test Delayed Interaction MLP (DIMLP), which processes audio and semantic features separately before a final linear fusion stage. This contrasts with MLP, which allows full nonlinear interactions across modalities, enabling the comparison of within-modality nonlinearity (DIMLP) and cross-modal nonlinear interactions (MLP). As shown in Table 1, both DIMLP and MLP outperform linear models. DIMLP, incorporating only within-modality nonlinearity, yields a 2.0% gain over the linear model (from 4.10% average r² to 4.18%). But the standard MLP, allowing full nonlinear interactions, achieves a further 2.6% gain (from 4.18% to 4.29%). These results suggest that both forms of nonlinearity enhance encoding performance, but cross-modal nonlinear interactions contribute most significantly.

This conclusion is further supported by voxelwise analysis (Appendix L). While DIMLP improves prediction accuracy across brain regions compared to linear models, standard MLP leads to further, cortex-wide enhancements. This suggests nonlinear interactions between audio and semantic features are essential for modeling neural representations underlying speech comprehension.

ROI-wise analysis (Figure 32) shows regional variation in nonlinearity's benefits. Multimodal MLP consistently matches or outperforms DIMLP and often surpasses linear models. Motor (e.g., M1M) and somatosensory regions (e.g., S1M) benefit most from nonlinear cross-modal interactions, highlighting their role in complex multimodal processing during speech comprehension.

## 3.3 MULTIMODAL ENCODERS

### 3.3.1 MULTIMODALITY REVEALS WIDESPREAD CORTICAL INTEGRATION

Our analysis shows that multimodality not only increases prediction accuracy across the cortex but also explains brain activity more effectively through joint audio-semantic processing. Improvements are brain-wide and extend well beyond modality-specific regions. Figure 2 **(a,b)** shows that adding audio features enhances predictions not only in auditory areas but also in primary motor and somatosensory regions, as well as the paracentral lobule between mPFC and Precuneus (PrCu), and parts of occipital cortex (OC). These effects highlight the widespread impact of auditory informa-

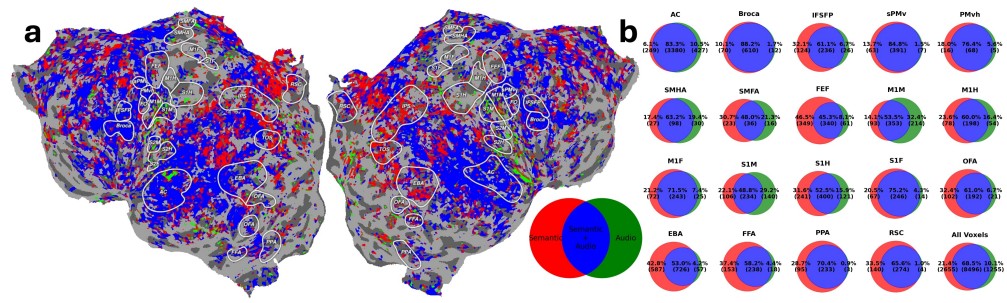

Figure 3: Visualization of most dominant feature type in brain activity predictions from variance partitioning analysis. (a) Voxel-wise plots from a single subject (S1) and (b) ROI-wise Venn diagrams showing which feature type (semantic: red, audio: green, joint: blue) explains the largest variance for each significantly predicted voxel ($q(\text{FDR}) < 0.01$) using MLP encoders. ROI results are aggregated across subjects with numbers indicating voxel percentages and counts.

tion. Conversely, Figure 2 **(c,d)** shows that adding semantic features improves predictions across most cortical regions, with the exception of some auditory cortex (AC) areas. This suggests that semantic processing exerts broad influence on neural activity, extending well beyond classical language regions.

These widespread improvements are further amplified by nonlinearity. Comparing Figure 2 **(b)** with **(a)**, and **(d)** with **(c)**, shows that MLP models not only strengthen effects seen with linear encoders but also unlock hidden gains in higher-order regions such as the LTC, mPFC, and OC. Variance partitioning analysis (Appendix M.2) reveals that most explained variance arises from joint audio–semantic contributions, while unique effects are dominated by semantic features, with audio contributing less across most regions. These results suggest that nonlinearity enables multimodal models to capture richer cross-modal integration, with semantics providing the primary source of unique information in brain-wide processing.

To further characterize representational dominance, we assigned each voxel to its most predictive modality. Joint audio–semantic features dominated cortical representations (Figure 3 **a**, shown for subject S1, with all subjects in Appendix M.3). This pattern is consistent across subjects: Rol-wise analysis (Figure 3 **b**) shows that semantic, audio, and joint features accounted for 21.4%, 10.1%, and 68.5% of significantly predicted voxels, respectively (subject-wise results in Appendix M.4).

Our findings both align with and extend prior multimodal language studies. Unlike Antonello et al. (2024), who reported localized auditory-driven improvements in AC and M1M, we observed cortex-wide gains. Methodological differences may explain this discrepancy: they used multiple Whisper layers, potentially introducing redundancy, and employed linear stacked regression, which limits modality interaction; in contrast, our approach leverages the final layer and direct concatenation, enabling richer integration (Appendix D). Our results also refine understanding of modality-specific contributions. Consistent with Oota et al. (2023), semantic models capture information beyond low-level acoustic features. Yet we find that audio models—though contributing less overall—provide meaningful complementary signals across multiple regions. This is evidenced by improved prediction accuracy and nonzero unique variance in our voxel-wise analyses, which likely capture fine-grained audio contributions that may be averaged out in the ROI-level analyses of Oota et al. (2023). Taken together, these patterns highlight distributed joint processing across the cortex, consistent with the Convergence-Divergence Zone theory (Damasio, 1989), which posits that semantic information is integrated from multiple modalities across widespread cortical regions.

### 3.3.2 MULTIMODAL FUSION IS CONSISTENT WITH NEUROLINGUISTIC THEORIES

Building on the brain-wide improvements observed, regions of interest (ROI) analyses reveal how multimodal integration aligns with and extends established neurolinguistic theories.

**Speech related regions (AC, Broca, sPMv, M1M)**

Our results highlight a systematic organization of speech processing along the auditory dorsal pathway, a core component of the dual-stream model of language processing (Hickok & Poeppel, 2007). This pathway, extending from the auditory cortex (AC) through Broca's area and the superior ventral premotor speech area (sPMv) to the primary motor cortex, shows distinct patterns of multimodal integration at successive stages.

In early AC, voxel-wise variance partitioning shows that unique contributions from audio features dominate (Figure 3), reinforcing its role in processing low-level acoustic information. However, processing in broader AC regions shows a shift to joint audio-semantic representations, with 83.3% of significantly predicted voxels showing joint audio-semantic representation. The improved performance from adding auditory features (Figures 2 **a, b**) supports this hierarchical pattern, with earlier AC areas showing greater gains.

Moving along the dorsal pathway to Broca's area and sPMv, we find predominant joint feature attribution (88.2% and 84.8% of voxels respectively) with improved predictions from the addition of either modality. This multimodal integration aligns with these regions' role in speech planning and articulatory control—processes that require integrating acoustic targets with semantic content and motor programs (Gough et al., 2005; Nixon et al., 2004; de Heer et al., 2017; Glanz et al., 2018).

At the terminus of the dorsal pathway, the mouth region in primary motor cortex (M1M) shows a strong contribution from auditory features, exceeding even AC, consistent with its role in executing speech articulation (32.4% of voxels) (Figure 3 **b**). This strong auditory presence in motor areas is further supported by substantial performance improvements when adding auditory features, reinforcing previous findings from Wu et al. (2014) that highlight the coupling between auditory and motor processes in speech production.

These findings extend our understanding of speech model representations. Our variance partitioning results align with previous findings that semantic models primarily predict AC activity by capturing low-level speech features (Oota et al., 2023). Our analysis also reveal some voxels show unique semantic contributions, and audio models capture distinct brain features beyond the typical scope of language models. The observed semantic contribution in AC, sPMv and Broca's area aligns with prior findings (de Heer et al., 2017) and may be a general mechanism for language processing.

**Motor and somatosensory areas: embodied speech processing**

The addition of audio or semantic features improved predictions in motor control (green) and somatosensory processing (blue) ROIs (Figure 2 **e**). Improvements vary: some ROIs benefit from semantic features (e.g., frontal eye field (FEF)), others from audio features (e.g., primary mouth motor cortex (M1M)), and some from both. Furthermore, variance partitioning analysis reveals that motor and somatosensory regions show unique contributions from both modalities in M1M, audio features uniquely explain 32.4% of the variance while semantic features explain 14.1%, with 53.5% jointly explained. Similar patterns emerge across motor areas (SMHA, SMFA, FEF, M1H, M1F) and somatosensory regions (S1M, S1H, S1F), suggesting these regions process unique auditory and semantic information absent from their overlapping features.

These findings align with the Motor Theory of Speech Perception (Liberman et al., 1967; 1952; Poeppel & Assaneo, 2020), which posits that motor regions simulate articulatory movements necessary for speech production, aiding comprehension. In particular, improvements from the addition of and the unique contribution from auditory features align with research showing tight coupling between auditory and motor-sensory processing (Skipper et al., 2005; Wu et al., 2014; Wilson et al., 2004).

These findings suggest semantic information shapes activity within somatosensory regions, indicating broader involvement in speech comprehension than previously recognized. One possible interpretation is embodied semantic memory theory, where concept understanding is grounded in sensorimotor experiences (Binder & Desai, 2011). Our results match Nagata et al. (2022)'s evidence that sensorimotor cortex processes both concrete and abstract word semantics. However, an alternative possibility is that the observed effects reflect quasi-semantic factors such as lexical frequency, predictability, or articulatory demands rather than concept-specific embodied simulation; our current design cannot distinguish between these explanations, and future work will be required to disentangle these multimodal interactions that our model captures.

The enhancements in these motor and sensory areas are more pronounced with MLP models, underscoring nonlinear interactions between auditory and semantic information. We explore this further in Section 3.2.

**Higher-order visual areas: multimodal semantic representations**

Adding semantic features enhances fMRI prediction accuracy in high-level visual areas like OFA (Pitcher et al., 2011), EBA (Downing et al., 2001), FFA (Kanwisher et al., 1997), PPA (Epstein & Kanwisher, 1998), and RSC (Vann et al., 2009) (Figure 2 **e**). Variance partitioning (Figure 3 **b**) shows these ROIs have largest contributions from semantic and joint features, suggesting text-derived semantics provide substantial predictive information for visual regions beyond audio features alone.

This finding matches studies showing visual and linguistic stimuli with similar semantic content elicit similar brain responses (Huth et al., 2012; 2016; Tang et al., 2024; Deniz et al., 2019; Devereux et al., 2013; Fairhall & Caramazza, 2013; Popham et al., 2021). In particular, Popham et al. (Popham et al., 2021) showed that semantic maps derived from movies and narrative stories in the same participants are aligned along the border of visual cortex; our results extend this line of work by demonstrating that multimodal text–audio encoding models also predict activity in overlapping high-level visual ROIs during purely auditory naturalistic listening. These results are consistent with the convergence-divergence-zone theory (Popham et al., 2021; Damasio et al., 1996; 2004; Damasio, 1989), which posits semantic information from multiple modalities integrates across the cortex, forming unified representations. This suggests the brain constructs modality-independent semantics using information from vision, language, and other senses (Tang et al., 2023; Binder & Desai, 2011; Tang et al., 2024; Martin, 2016).

Our study also provides novel evidence for auditory modality's contribution to this unified semantic representation. Variance partitioning (Figure 3 **b**) shows auditory information accounts for 5% of voxels in higher visual area ROIs. Adding audio features resulted in significant performance increases in these ROIs (Figure 2 **e**), suggesting auditory information, such as tone of voice and environmental sounds, may provide unique semantic context not fully captured by visual or linguistic features alone.

The consistent observation that multimodal fusion, particularly with nonlinear models, enhances prediction accuracy emphasizes the brain's use of complex, nonlinear computations to combine information from different modalities for a holistic understanding of language. Subject-wise ROI prediction differences are visualized in Figure 29 (Appendix K.3).

## 4 DISCUSSION AND CONCLUSION

This study underscores the transformative potential of nonlinear multimodal approaches to speech encoding for advancing our understanding of speech comprehension in the brain. While nonlinear approaches have become standard in vision encoding models (Yang et al., 2023; Scotti et al., 2024; Chen et al., 2023), their application to language has faced unique challenges due to the dynamic, cortex-wide nature of speech comprehension. Our approach overcomes these challenges, achieving a 14.4% increase in mean normalized correlation compared to previous state-of-the-art models (Antonello et al., 2024), while more importantly revealing previously hidden functional organization patterns.

A key finding is that nonlinear models provide more nuanced insights into neural activity, outperforming linear approaches across all network layers, with gains driven by nonlinearity rather than dimensionality reduction alone. The benefits of nonlinear encoding are showcased in our RED analysis, which reveals improved hierarchical clustering of brain regions, with higher modularity (0.155) than linear models (0.145) and traditional connectivity measures (0.068).

Our second key finding illustrates how multimodal encoding approaches expose aspects of neural computation that may be overlooked in unimodal models. By systematically comparing unimodal and multimodal predictions across the cortex, we discovered widespread cross-modal integration patterns. Through ROI-wise analyses of both variance partitioning and performance improvements, we show alignment with key neurolinguistic theories including the Motor Theory of Speech Perception (Liberman et al., 1967), Convergence-Divergence Zone model (Damasio, 1989), and embodied

semantics (Davis & Yee, 2021), and dorsal aspect of the dual stream hypothesis (Hickok & Poeppel, 2007) highlighting the brain's reliance on distributed multimodal fusion.

Our nonlinear encoding approach has two main limitations. First, insufficient dataset size currently constrains model complexity, leading to overfitting when adding hidden layers or using RNNs and Transformers (Appendix E). Given data scaling benefits in linear encoders (Antonello et al., 2024) and how a large dataset such as the Natural Scenes Dataset (Allen et al., 2022) enabled deep learning breakthroughs in visual encoding and decoding (Adeli et al., 2023; Scotti et al., 2024), larger language fMRI datasets are needed to fully harness the potential of deep learning and drive further advancements. Second, while nonlinear encoders offer strong performance gains, they create new interpretability challenges. While variance partitioning and RED-based clustering offer preliminary insights, further innovations such as RSA (Kriegeskorte et al., 2008) and novel feature attribution (Oota et al., 2023) are necessary. Moreover, nonlinear models offer unique interpretative possibilities, as shown by (Yang et al., 2023) in memory vision encoding.

Taken together, these results suggest that nonlinear encoders should not replace linear models, but rather complement them. Linear models remain preferable when the brain–model relationship is approximately linear and the primary goal is fine-grained attribution of feature weights (e.g., separating semantic, syntactic, or discourse dimensions and mapping them to specific regions). However, if cortical language and multimodal processing depend on strong nonlinear interactions, enforcing a purely linear mapping can misattribute variance or underestimate the contribution of such interactions. In this regime, a relatively simple nonlinear mapping like our PCA + single-hidden-layer MLP can provide a more faithful approximation of the underlying computation, while linear analyses remain essential for detailed feature-level interpretation.

In conclusion, our study demonstrates that while linear and unimodal approaches have provided valuable insights in speech encoding research, nonlinear multimodal encoding models reveal important aspects of neural speech processing that complement these established methods. Addressing dataset size and model interpretability limitations will be key to advancing brain aligned AI, enabling models that better reflect the hierarchical and distributed nature of neural processing.

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

## Appendix table of contents

# A  ABBREVIATIONS OF BRAIN AREAS AND REGIONS OF INTEREST (ROIs)

Brain Areas are abbreviated as follows :

- **AC**: Auditory Cortex
- **AG**: Angular Gyrus
- **LPFC**: Lateral Prefrontal Cortex
- **LTC**: Lateral Temporal Cortex
- **mPFC**: Medial Prefrontal Cortex
- **OC**: Occipital Cortex
- **PrCu**: Precuneus

The ROIs are abbreviated as follows :

- **AC**: Auditory Cortex
- **AG**: Angular Gyrus
- **Broca**: Broca's Area
- **EBA**: Extrastriate Body Area
- **FFA**: Fusiform Face Area
- **FEF**: Frontal Eye Field
- **IFSFP**: Inferior Frontal Sulcus Face Patch
- **LPFC**: Lateral Prefrontal Cortex
- **LTC**: Lateral Temporal Cortex
- **M1F**: Primary Motor Cortex - Foot
- **M1H**: Primary Motor Cortex - Hand
- **M1M**: Primary Motor Cortex - Mouth
- **mPFC**: Medial Prefrontal Cortex
- **OC**: Occipital Cortex
- **OFA**: Occipital Face Area
- **PMvh**: Ventral Premotor Hand Area
- **PPA**: Parahippocampal Place Area
- **PrCu**: Precuneus
- **RSC**: Retrosplenial Cortex
- **S1F**: Primary Somatosensory Cortex - Foot
- **S1H**: Primary Somatosensory Cortex - Hand
- **S1M**: Primary Somatosensory Cortex - Mouth
- **sPMv**: Superior Ventral Premotor Speech Area
- **SMFA**: Supplementary Motor Foot Area
- **SMHA**: Supplementary Motor Hand Area

# B  DETAILS OF IMPLEMENTATION

## B.1  LLAMA FEATURE EXTRACTION STRATEGY

Llama feature extraction was done in a dynamical window size manner for efficiency. Initially, the context window grew incrementally as tokens were added, up to a maximum of 512 tokens, after which the window was reset to a new context of 256 tokens. This approach avoided memory overheads associated with processing the entire tokenized text while maintaining sufficient contextual information for accurate semantic representation.

## B.2 Noise ceiling ($CC_{max}$) and nocmalized correlation ($CC_{norm}$) calculation

For each voxel, the maximum correlation coefficient is estimated as $CC_{max} = (\sqrt{1 + \frac{NP}{SP \times N}})^{-1}$, where $N$ is the number of repeats (10 in our case), $NP$ is the noise power or unexplainable variance, and $SP$ is the amount of variance that could be explained by an ideal predictive model.

## B.3 Resampling the hidden state of LLMs to fMRI time points

After giving the language/audio model the same input as the subject, we temporally aligned the hidden states of its $l^{th}$ layer corresponding to a given $i^{th}$ token (last token of the $i^{th}$ word for language models), $H_l^i(S_{\{k|k \leq i\}}) \in \mathbb{R}^{d_{model}^l}$ (aggregate shape of $\mathbb{R}^{N_{token} \times d_{model}^l}$ for the whole story where $N_{token}$ is the number of tokens/words), to the fMRI acquisition times (TR times) using Lanczos interpolation, obtaining an extracted feature of size $\mathbb{R}^{N_{TR} \times d_{model}^l}$, where $N_{TR}$ is the number of tokens (or number of words for language models) for each story and $d_{model}^l$ is the dimension of the $l^{th}$ hidden layer. We constructed the feature corresponding to a given $n^{th}$ TR ($2n$ seconds in physical time) by concatenating the representations from four previous TRs (2, 4, 6, 8 seconds before $t$ in physical time) to get a vector of shape $\mathbb{R}^{4d_{model}^l}$ for every $n^{th}$ TR, which we denote as $H_l'^n(S_{\{t|t \leq 2n\}})$. $H'$ denotes the additional resampling and concatenation done after applying the model, $H$. We used four previous time delays (2, 4, 6, 8 seconds) to account for the delay between the stimuli and brain response and to provide past stimuli information to the model. (Our process is identical to that of Antonello et al. (2024), ensuring that the same input was given, ensuring fair comparison.

## B.4 Representations for fMRI response using PCA

To an aggregate fMRI response, $Y_{org} \in \mathbb{R}^{N_{TR} \times N_{voxels}}$, we applied PCA with 8192 maximum components along the voxel dimension using scikit-learn (Pedregosa et al., 2011), yielding an approximate projection matrix, $W \in \mathbb{R}^{N_{voxels} \times N_{8192}}$. Given $N_{PCA}$ number of principal components to consider, we take the top $N_{PCA}$ components to get $W_{PCA} \in \mathbb{R}^{N_{voxels} \times N_{PCA}}$, and train the encoding model to predict the reduced dimension PCA projection of the data, $Y_{PCA} = Y_{org}W_{PCA} \in \mathbb{R}^{N_{TR} \times N_{PCA}}$. During evaluation, the trained model outputs a reduced dimension representation of the data, $\hat{Y}_{PCA}^{test} \in \mathbb{R}^{N_{TR\text{-}test} \times N_{PCA}}$, where $N_{TR\text{-}test}$ denotes the number of timepoints (TRs) in the test story. This is reconstructed back the the original voxel space by applying an inverse of the projection matrix, $\hat{Y}^{test} = \hat{Y}_{PCA}^{test}W_{PCA}^T \in \mathbb{R}^{N_{TR\text{-}test} \times N_{voxels}}$, which is later compared with the ground truth, $Y^{test} \in \mathbb{R}^{N_{TR\text{-}test} \times N_{voxels}}$.

It should be noted that due to the high dimensionality of the data, incremental PCA was used, in place of regular PCA.

## B.5 Details of encoding models

The encoding model architecture is as follows:

- *Linear Regression (Linear):* Ridge regression. Following Antonello et al. (2024), ridge regression with bootstrapping ($n = 3$) was used to estimate the optimal regularization parameters (alphas) for each voxel. The training data was divided into chunks of length 20, with 25% used for held-out validation in each bootstrap iteration. The best alpha values were averaged across iterations, and the final model was trained on the full training dataset using these alphas.

- *Multi-Layer Perceptron (MLP):* MLP with a single hidden layer of 256 units, applying batch normalization and dropout to prevent overfitting. The hyperbolic tangent ($\tanh$) was used as the activation function.

- *Multi-Layer Linear (MLLinear):* MLP but without dropout, batch normalization, and with the identity activation function.

- *Delayed Interaction MLP (DIMLP):* MLP variant processes. Each modality through separate 256-unit hidden layers before concatenation and final linear projection.

We implemented encoding models using PyTorch. We employed the AdamW optimizer (Loshchilov, 2017) with a batch size of 128 and Mean Absolute Error (MAE) as the loss function to mitigate excessively penalizing random signal fluctuations. Our training regime consisted of 200 epochs with early stopping (patience = 10) based on validation loss, and we applied batch normalization with a momentum of 0.1. For robust evaluation, we implemented 5-fold cross-validation, averaging predictions across the five models for our final results. Hyperparameter optimization was conducted using Optuna (Akiba et al., 2019), which performed 70 trials to determine optimal values for the dropout rate (0.1 to 0.3), learning rate ($10^{-5}$ to $10^{-1}$), and weight decay ($5 \times 10^{-5}$ to $10^{-1}$).

Ridge regression was performed using a CPU node with 96 cores (Intel(R) Xeon(R) Gold 6240R CPU @ 2.40GHz) and 512 GB of RAM. Running the audio and language models and training encoding models was done using a GPU node with 8 H100 80GB GPUs.

## C  STATISTICAL TESTING

Tables 2 and 3 report voxelwise encoding performance for all models and subjects. For each subject and model, we first compute the prediction score ($r^2$ or $CC_{norm}$) for every voxel and summarize these values as mean $\pm$ standard error of the mean (SEM) across voxels. Thus, the SEM reflects the variability of model performance over voxels within a subject, treating voxels as repeated measurements of the same underlying model. Overall performance level can vary substantially across subjects due to idiosyncratic factors. As a result, the variance of subject-averaged scores across subjects is dominated by these random effects and is not very informative as an uncertainty measure. We therefore do not report an across-subject SEM; instead, the "Across subjects" column gives only the arithmetic mean of the subject-wise means, and the percentages in parentheses indicate the relative change with respect to the text-linear (all voxels) baseline model.

Using the same voxelwise scores, Figures 4 and 5 perform pairwise significance testing between models. For each pair of models $(A, B)$ and each subject, we compute the distribution across voxels of the difference in prediction scores and form a t-statistic for the null hypothesis that the mean difference is zero. The resulting t-statistics, after Bonferroni correction over all model pairs, are visualized as heatmaps, where red (blue) cells indicate that Model A performs significantly better (worse) than Model B, and non-significant comparisons are masked in white.

Taken together, these results indicate that the main conclusions of the paper are robust to formal statistical testing. First, both Tables 2 and 3 show that multimodal text+audio models consistently outperform text-only and audio-only models for every subject, with the text+audio MLP (PCA) variant achieving the highest across-subject scores on both $r^2$ and $CC_{norm}$ (approximately +17% relative to the text-linear baseline). Second, Figures 4 and 5 confirm that these gains are statistically reliable: the top multimodal models significantly outperform the text-linear baseline and all audio-only models for nearly all subjects, even under conservative Bonferroni correction. Finally, the pooled voxelwise analyses (panels (d)) demonstrate that these effects are not driven by any single subject but are stable across the dataset. Overall, the statistical tests support our claims that (i) incorporating audio provides a genuine benefit over text alone, and (ii) non-linear encoders with appropriate dimensionality reduction (MLP + PCA) extract substantially more predictive information from the stimuli than linear baselines.

Table 2: Per-subject voxelwise encoding performance measured by average $r^2$ (mean $\pm$ SEM, in %). For each model and subject, the mean and SEM are computed across voxels; the across-subject value is the mean of the subject-wise means. Percentages in parentheses indicate the across-subject change relative to the text-linear (all voxels) baseline.

| modality 1 | modality 2 | encoder | response | S1 ($\pm$SEM) | S2 ($\pm$SEM) | S3 ($\pm$SEM) | Across subjects |
|---|---|---|---|---|---|---|---|
| text | audio | MLP | PCA | 2.58$\pm$0.02 | 4.49$\pm$0.02 | 5.79$\pm$0.03 | **4.29 (+17.2)** |
| text | audio | DIMLP | PCA | 2.54$\pm$0.02 | 4.40$\pm$0.02 | 5.61$\pm$0.03 | 4.18 (+14.2) |
| text | audio | MLLinear | PCA | 2.43$\pm$0.02 | 4.33$\pm$0.02 | 5.54$\pm$0.03 | 4.10 (+12.0) |
| text | audio | Linear | all voxels | 2.48$\pm$0.02 | 4.26$\pm$0.02 | 5.57$\pm$0.03 | 4.10 (+12.0) |
| text | audio | Linear | PCA | 1.70$\pm$0.01 | 4.24$\pm$0.02 | 5.66$\pm$0.03 | 3.87 (+5.7) |
| text | audio | MLP | all voxels | 2.46$\pm$0.02 | 4.01$\pm$0.02 | 5.00$\pm$0.02 | 3.83 (+4.6) |
| text | – | MLP | PCA | 2.10$\pm$0.01 | 4.14$\pm$0.02 | 5.13$\pm$0.02 | 3.79 (+3.6) |
| text | – | MLLinear | PCA | 1.97$\pm$0.01 | 4.07$\pm$0.02 | 4.97$\pm$0.02 | 3.67 (+0.3) |
| text | – | Linear | all voxels | 2.02$\pm$0.01 | 4.01$\pm$0.02 | 4.96$\pm$0.02 | 3.66 (Baseline) |
| text | – | Linear | PCA | 1.59$\pm$0.01 | 4.01$\pm$0.02 | 5.08$\pm$0.02 | 3.56 (–2.7) |
| text | – | MLP | all voxels | 2.02$\pm$0.01 | 3.55$\pm$0.02 | 4.51$\pm$0.02 | 3.36 (–8.2) |
| audio | – | MLP | PCA | 2.04$\pm$0.01 | 2.83$\pm$0.02 | 4.15$\pm$0.02 | 3.01 (–17.8) |
| audio | – | MLP | all voxels | 1.94$\pm$0.01 | 2.70$\pm$0.01 | 4.04$\pm$0.02 | 2.89 (–21.0) |
| audio | – | MLLinear | PCA | 1.95$\pm$0.01 | 2.73$\pm$0.02 | 3.99$\pm$0.02 | 2.89 (–21.0) |
| audio | – | Linear | PCA | 1.73$\pm$0.01 | 2.61$\pm$0.01 | 4.07$\pm$0.02 | 2.81 (–23.2) |
| audio | – | Linear | all voxels | 1.86$\pm$0.01 | 2.63$\pm$0.02 | 3.83$\pm$0.02 | 2.77 (–24.3) |

Table 3: Per-subject voxelwise encoding performance measured by normalized correlation coefficient ($CC_{norm}$, mean $\pm$ SEM, in %). For each model and subject, the mean and SEM are computed across voxels; the across-subject value is the mean of the subject-wise means. Percentages in parentheses indicate the across-subject change relative to the text-linear (all voxels) baseline.

| modality 1 | modality 2 | encoder | response | S1 ($\pm$SEM) | S2 ($\pm$SEM) | S3 ($\pm$SEM) | Across subjects |
|---|---|---|---|---|---|---|---|
| text | audio | MLP | PCA | 36.74$\pm$0.11 | 27.23$\pm$0.10 | 39.01$\pm$0.09 | **34.32 (+17.9)** |
| text | audio | DIMLP | PCA | 34.88$\pm$0.11 | 25.08$\pm$0.10 | 37.81$\pm$0.10 | 32.59 (+11.9) |
| text | audio | MLLinear | PCA | 33.85$\pm$0.11 | 26.53$\pm$0.10 | 36.85$\pm$0.10 | 32.41 (+11.3) |
| text | audio | Linear | all voxels | 31.98$\pm$0.12 | 26.41$\pm$0.10 | 35.72$\pm$0.10 | 31.37 (+7.7) |
| text | audio | MLP | all voxels | 32.45$\pm$0.11 | 25.35$\pm$0.09 | 35.53$\pm$0.09 | 31.11 (+6.8) |
| text | audio | Linear | PCA | 22.66$\pm$0.11 | 27.12$\pm$0.09 | 36.98$\pm$0.10 | 28.92 (–0.7) |
| text | – | MLP | PCA | 32.40$\pm$0.11 | 24.68$\pm$0.10 | 35.61$\pm$0.09 | 30.89 (+6.1) |
| text | – | MLLinear | PCA | 30.50$\pm$0.11 | 25.60$\pm$0.10 | 33.77$\pm$0.10 | 29.95 (+2.8) |
| text | – | Linear | all voxels | 28.37$\pm$0.12 | 25.82$\pm$0.10 | 33.36$\pm$0.10 | 29.12 (Baseline) |
| text | – | Linear | PCA | 22.03$\pm$0.11 | 25.12$\pm$0.10 | 33.49$\pm$0.10 | 26.88 (–7.7) |
| text | – | MLP | all voxels | 27.84$\pm$0.11 | 22.49$\pm$0.09 | 32.01$\pm$0.09 | 27.45 (–5.7) |
| audio | – | MLP | PCA | 30.59$\pm$0.11 | 21.70$\pm$0.09 | 34.73$\pm$0.09 | 29.01 (–0.4) |
| audio | – | MLP | all voxels | 28.85$\pm$0.11 | 22.29$\pm$0.08 | 33.50$\pm$0.09 | 28.21 (–3.1) |
| audio | – | MLLinear | PCA | 28.71$\pm$0.11 | 21.08$\pm$0.09 | 32.70$\pm$0.09 | 27.50 (–5.6) |
| audio | – | Linear | PCA | 24.62$\pm$0.10 | 22.02$\pm$0.08 | 33.50$\pm$0.09 | 26.71 (–8.3) |
| audio | – | Linear | all voxels | 25.67$\pm$0.11 | 19.94$\pm$0.08 | 29.86$\pm$0.09 | 25.16 (–13.5) |

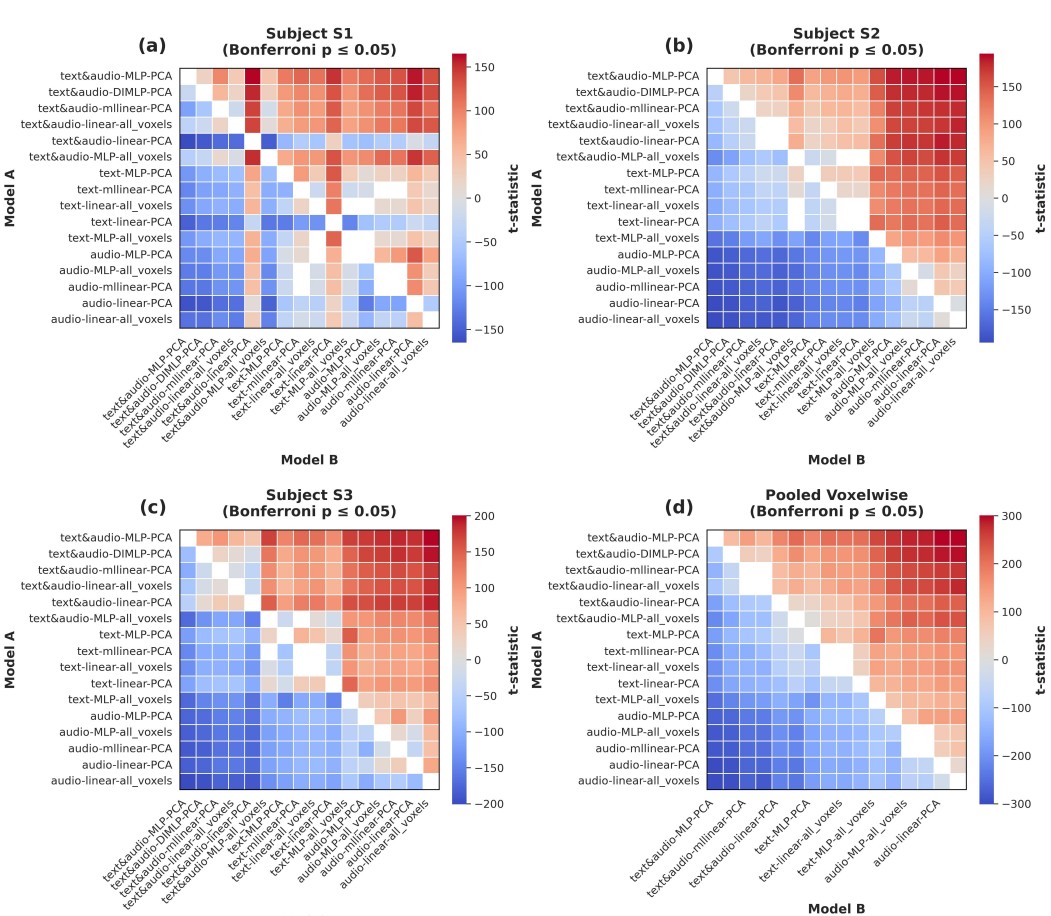

Figure 4: Pairwise comparison of encoding model performance ($r^2$) across subjects. Heatmaps display the t-statistics for pairwise comparisons of voxelwise prediction performance between models. Each cell $(i, j)$ represents the t-statistic for the hypothesis that Model A (row $i$) outperforms Model B (column $j$). Red indicates that Model A performs significantly better than Model B, while blue indicates that Model B performs significantly better than Model A. Comparisons that are not statistically significant after Bonferroni correction ($p > 0.05$) are masked in white. Panels **(a)-(c)** show results for individual subjects (S1, S2, S3), and panel **(d)** shows the pooled results across all subjects.

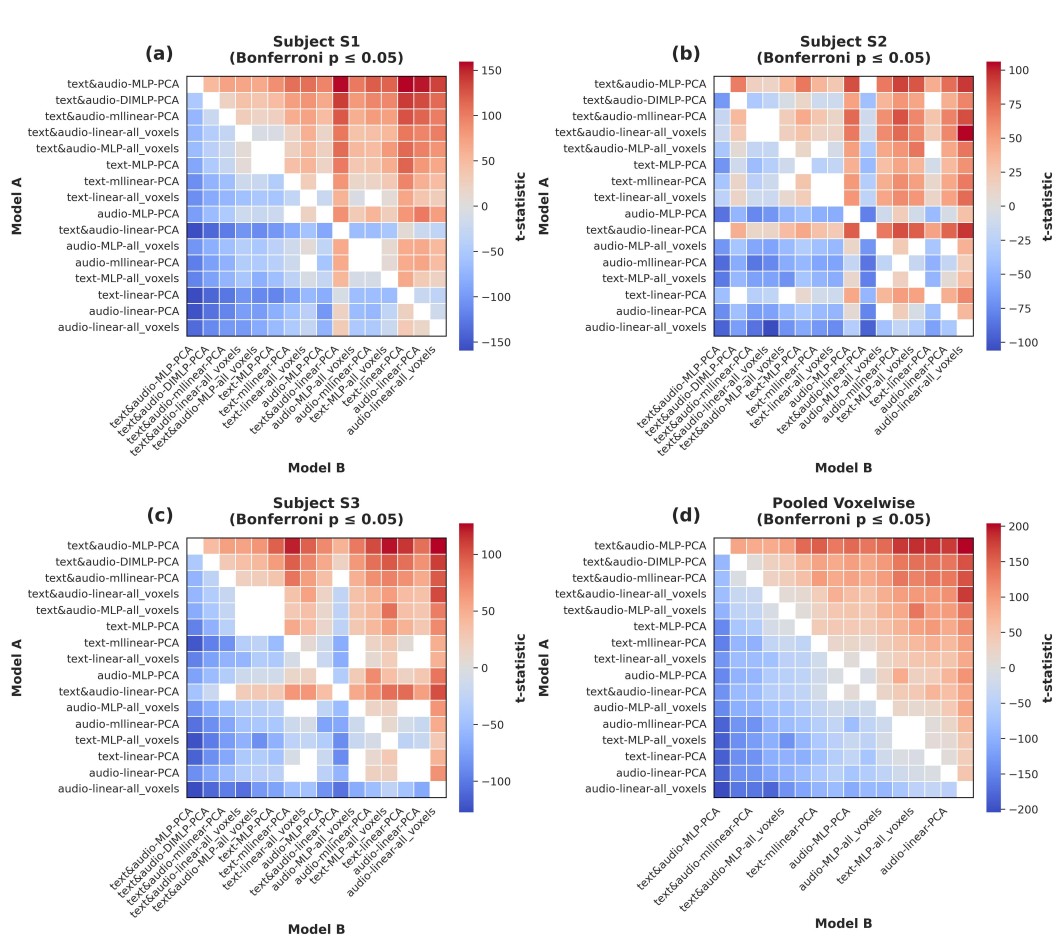

Figure 5: Pairwise comparison of encoding model performance ($CC_{\text{norm}}$) across subjects. The rest are identical to Table 4

# D COMPARISON WITH STACKED REGRESSION MODEL OF ANTONELLO ET AL. (2024)

Table 4: Comparing encoding performance across different models using the single test story evaluation protocol. Values show normalized correlation coefficient ($CC_{norm}$) and story-specific $r^2$ (**Avg $r^2$ (story)**)(distinguishing from Table 1's three-story evaluation (**Avg $r^2$**)). SR refers to the previous state-of-the-art stacked regression model (Antonello et al., 2024), which combines LLM and audio predictions through weighted averaging. Two masking approaches are used: 1) "$mask_A$" - their pre-computed validation-based voxel selection mask, and 2) "mask" - our computed masks that retain voxels showing validation improvements. For "mask", Linear+Mask indicates creating and applying a mask based on multimodal linear vs semantic linear performance, while MLP+Mask does the same using MLP models. $semantic_A$ denotes features from LLAMA-30B's 18th layer used in SR, while our models uses features from the 12th layer of LLAMA-7B. All approaches are evaluated using identical test data for fair comparison and $r^2$ is computed as $|r| * r$.

| modality 1 | modality 2 | encoder | response | Avg $r^2$ (single story) | Avg $CC_{norm}$ |
|---|---|---|---|---|---|
| semantic | audio | MLP | PCA | **5.13%** (+7.7%) | **34.32%** (+14.4%) |
| semantic | audio | MLP + mask | PCA | 5.02% (+5.5%) | 33.33% (+11.0%) |
| semantic | audio | DIMLP | PCA | 4.93% (+3.6%) | 32.59% (+8.6%) |
| semantic | audio | MLLinear | PCA | 5.00% (+5.1%) | 32.41% (+8.0%) |
| semantic | audio | MLP + $mask_A$ | PCA | 4.77% (+0.2%) | 31.70% (+5.6%) |
| semantic | audio | Linear | all voxels | 4.92% (+3.4%) | 31.36% (+4.5%) |
| semantic | audio | MLP | all voxels | 4.54% (-4.5%) | 31.11% (+3.6%) |
| semantic | audio | Linear + mask | all voxels | 4.90% (+2.9%) | 31.09% (+3.6%) |
| $semantic_A$ | audio | SR + $mask_A$ | all voxels | 4.76% (Baseline) | 30.02% (Baseline) |
| semantic | audio | Linear | PCA | 4.48% (-5.8%) | 28.92% (-3.7%) |
| semantic | - | MLP | PCA | 4.58% (-3.7%) | 30.89% (+2.9%) |
| semantic | - | MLLinear | PCA | 4.59% (-3.6%) | 29.95% (-0.2%) |
| $semantic_A$ | - | Linear | all voxels | 4.60% (-3.3%) | 29.84% (-0.6%) |
| semantic | - | Linear | all voxels | 4.50% (-5.4%) | 29.12% (-3.0%) |
| semantic | - | MLP | all voxels | 3.97% (-16.6%) | 27.45% (-8.6%) |
| semantic | - | Linear | PCA | 4.15% (-12.8%) | 26.88% (-10.4%) |
| audio | - | MLP | PCA | 3.83% (-19.6%) | 29.01% (-3.4%) |
| audio | - | MLP | all voxels | 3.67% (-22.8%) | 28.21% (-6.0%) |
| audio | - | MLLinear | PCA | 3.66% (-23.1%) | 27.50% (-8.4%) |
| audio | - | Linear | PCA | 3.54% (-25.6%) | 26.71% (-11.0%) |
| audio | - | Linear | all voxels | 3.46% (-27.3%) | 25.20% (-16.0%) |

To establish the effectiveness of our nonlinear multimodal approach, we conduct a detailed comparison with the current state-of-the-art stacked regression model (Antonello et al., 2024). Their method combines semantic and audio predictions through stacked regression followed by voxel-selection, where they decide what model to use (stacked regression or semantic linear) for each voxel based on a validation dataset. Their results are compared here and not in Table 1 due to their use of only parts of the test stories as validation, barring computation of the "**Avg $r^2$**" value in Table 1. For accurate comparison, we obtain and use their published model weights and features.

The evaluation protocols differ specifically for the stacked regression (SR) model: while all models (including those in Antonello et al. (2024)) primarily report performance using three test stories (Table 1), SR uniquely requires using two of these test stories for validation-based voxel selection and only using the story "wheretheressmoke" for final testing.

Also, following the identification of an error in the original evaluation protocol through community feedback, we corrected the methodology for fair comparison. Note that $CC_{norm}$ values remain consistent with Table 1 as they were originally computed using only the "wheretheressmoke" story due to the unavailability of test repeats for the other two stories.

To ensure fair comparison with SR, we additionally evaluate all models using their single-story protocol in Table 4, reporting both $CC_{norm}$ and a story-specific **Avg $r^2$ (single story)** metric to distinguish from our three-story evaluation. We found $CC_{norm}$ provides more stable comparisons than $r^2$ in this context, as the reduced number of timepoints (251 versus 790) makes $r^2$ more susceptible

to noisy voxels compared to $CC_{norm}$ that accounts for these noisy voxels. This stability is reflected in the closer alignment between $CC_{norm}$ and $r^2$ rankings in Table 1 compared to Table 4. Therefore, we sort Table 4 with respect to the $CC_{norm}$.

Also, while their approach uses LLAMA-30B's 18th layer (denoted as semantic$_A$), we demonstrate competitive performance using LLAMA-7B features, consistent with our finding that encoding performance roughly plateaus beyond 7B parameters (Appendix G). For comprehensive comparison, we implement both their pre-computed validation-based voxel selection mask ("mask$_A$", created using an unspecified significance threshold) and our simpler approach ("mask") that retains voxels showing any validation set improvement.

Table 4 demonstrates several key results about our multimodal nonlinear approach. Our multimodal MLP achieves 34.32% $CC_{norm}$ without masking, representing a 14.4% improvement over the baseline stacked regression model, though the Avg $r^2$ (story) improvement is more modest at 7.7%.

Our multimodal linear encoder also outperforms stacked regression by 4.5%, supporting our hypothesis that direct concatenation enables more effective modality interaction compared to weighted averaging of unimodal predictions. The performance hierarchy (MLP > Linear > SR) suggests that both architectural choices - direct multimodal fusion and nonlinearity - contribute independently to improved predictions.

Interestingly, validation-based masking did not improve performance for either our linear or MLP models, regardless of whether using our mask or the precomputed mask$_A$ from previous work. This suggests our models learn effective feature selection implicitly, determining when to leverage or ignore audio features for specific voxels without explicit masking. The benefit of removing masking also likely stems from our models' ability to learn voxel-specific feature importance through direct access to input data, combined with the inherent noise in validation masks due to the limited number of timepoints.

These results demonstrate that enabling direct interaction between modalities through concatenation, combined with nonlinear processing, provides a more robust approach than previous methods relying on weighted averaging and explicit feature selection.

# E    RESULTS OF MORE COMPLEX NONLINEAR MODELS

Table 5: Encoding performance of various nonlinear semantic encoders compared to other models. The table presents the average $r^2$ and normalized correlation coefficients ($CC_{norm}$) along with percentage changes relative to the baseline Linear model. Deep MLP refers to an MLP with two hidden layers, while MLP is an MLP with one hidden layer.

| modality 1 | modality 2 | encoder | response | Avg $r^2$ | Avg $CC_{norm}$ |
|---|---|---|---|---|---|
| semantic | - | MLP | PCA | 3.79% (+3.6%) | 30.89% (+6.1%) |
| **semantic** | **-** | **Linear** | **all voxels** | **3.66% (Baseline)** | **29.12% (Baseline)** |
| semantic | - | LSTM | PCA | 3.33% (-9.0%) | 26.95% (-7.46%) |
| semantic | - | GRU | PCA | 3.21% (-12.3%) | 26.15% (-10.2%) |
| semantic | - | DeepMLP | PCA | 3.05% (-16.7%) | 27.45% (-5.73%) |
| semantic | - | RNN | PCA | 2.99% (-18.0%) | 25.42% (-12.7%) |
| semantic | - | Transformer | PCA | 2.82% (-23.0%) | 27.97% (-3.95%) |

We explored a range of more complex nonlinear models, as detailed in Table 5. Specifically, we evaluated LSTM, GRU, RNN, and Transformer architectures, each configured with a single layer. The hidden dimensions for these models were determined by experimenting with sizes of 256, 512, 768, and 1024, selecting the dimension that yielded the best performance.

All models received inputs consisting of four timepoints, consistent with the MLP model, which concatenates these timepoints. For the recurrent models (LSTM, GRU, RNN), the final predictions were generated by applying a linear projection to a weighted pooling of the outputs corresponding to the four input timepoints. In the case of the Transformer model, we utilized learnable positional embeddings along with full self-attention mechanisms, and the final prediction was obtained by linearly projecting the output of the last token.

Additionally, we examined the DeepMLP model, an extension of the standard MLP with two hidden layers instead of one.

Our results indicate that while the MLP with a single hidden layer outperforms linear models, introducing greater complexity—such as recurrent models or additional hidden layers—leads to overfitting and decreased performance.

## F  PERFORMANCE OF MULTIMODAL MLP MODEL WHEN MIXING DIFFERENT LAYERS

We observe in Figure 6 that integrating the best performing layers from each modality results in the best performing multimodal model.

Figure 6: Heatmap showing average $r^2$ values for different combinations of LLAMA and Whisper layer depths using an MLP encoder. Darker colors represent higher performance, with the best results obtained when the best layers in the respective uni-modal encoding models were used.

## G  SCALING LLM AND AUDIO MODELS DOES NOT NECESSARILY LEAD TO BETTER ENCODERS

Previous research by Antonello et al. (2024) found that increasing the size of large language models (LLMs) and audio models, such as scaling OPT from 125M to 175B parameters or Whisper from 8M to 637M parameters, enhanced encoding performance. However, performance gains plateaued for larger models like LLAMA-33B and OPT-175B, which they attributed to overfitting from larger hidden sizes.

Building on these findings, our study delves deeper into the scaling trends and offers a refined perspective on their implications for brain encoding models. For audio models, we confirm a positive

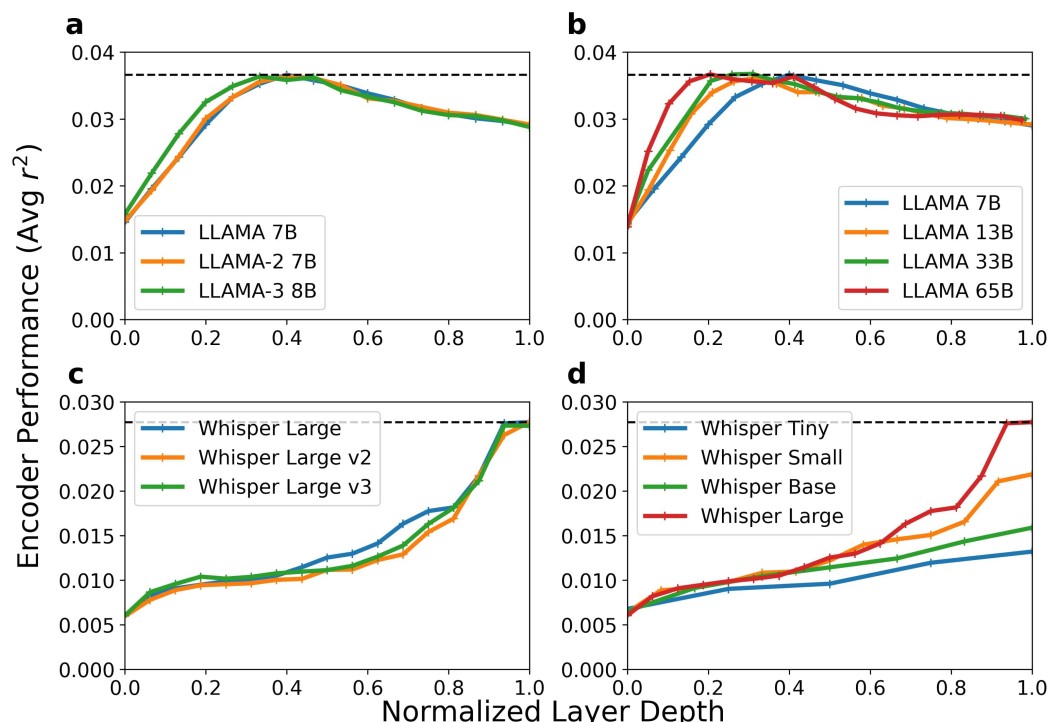

Figure 7: Encoder performance across different LLAMA and Whisper model variants, using linear regression applied to the full set of voxels. Panel (a) compares LLAMA models of various architectures (LLAMA-2 and LLAMA-3) with 7B and 8B parameters. Panel (b) presents performance across different LLAMA models of increasing sizes, from 7B to 65B. Panels (c) and (d) show the performance for different Whisper model variants, including comparisons between Whisper Large versions (c) and different model sizes (d), from Whisper Tiny to Whisper Large. Performance is measured in terms of average $r^2$, plotted against normalized layer depth.

correlation between model size and performance, as shown in Figure 7 (d). However, this scaling effect does not hold for language models. Specifically, LLAMA-7B, LLAMA-13B, LLAMA-33B, and LLAMA-65B exhibit comparable encoding performance, as shown in Figure 7 (b). This suggests diminishing returns beyond 7 billion parameters, a finding consistent with prior work by Bonnasse-Gahot & Pallier (2024), which reported performance plateaus for LLMs larger than 3 billion parameters.

We also evaluated the impact of scaling training data by examining newer versions of LLAMA and Whisper (e.g., LLAMA-1, LLAMA-2, LLAMA-3; Whisper v1, v2, v3). Despite larger datasets, newer versions did not yield significant performance improvements for either audio or semantic encoding models. This indicates that advancements in self-supervised learning (SSL) tasks, such as better next-token prediction, do not necessarily translate to more effective features for brain encoding models. In essence, SSL improvements do not directly enhance brain-aligned representations.

In conclusion, our findings highlight two key points: (1) scaling language models beyond 7 billion parameters does not substantially improve encoding performance, and (2) increasing training data or using newer model versions does not enhance brain encoding feature extractors. These results challenge the assumption that simply scaling feature extractors, as proposed by Antonello et al. (2024), will lead to better encoding models.

## H  CONTEXT SIZE SPEECH MODELS INFLUENCE ENCODER PERFORMANCE

Figure 8 illustrates the impact of varying the context size (window size) of the Whisper model on encoding performance when using linear encoders, as explored in Oota et al. (2023). The results indicate that a 16-second window size, which was used as the default throughout our study, delivers the best performance. This outcome aligns with expectations, as the selected window size is consistent with the recommendations from Antonello et al. (2024).

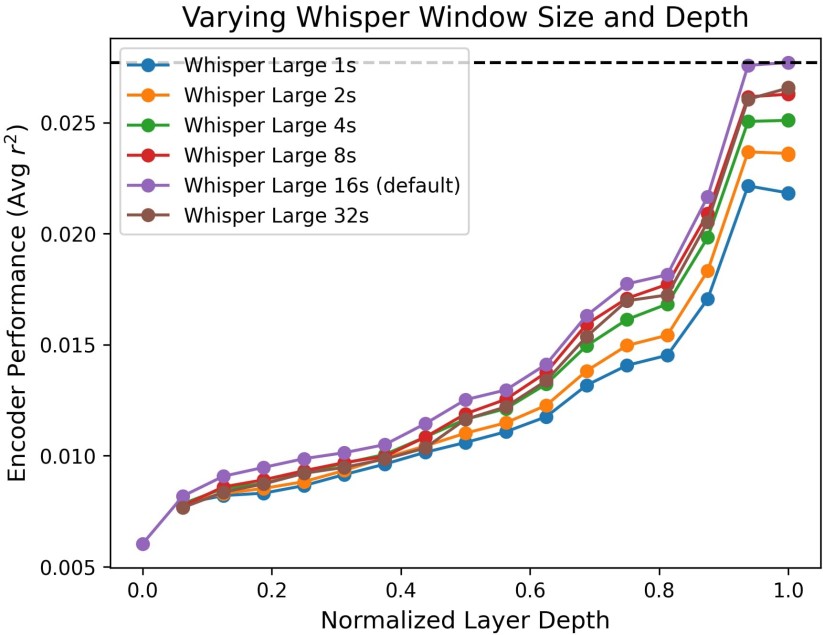

Figure 8: Encoder performance across different Whisper Large models with varying window size, using linear regression applied to the full set of voxels.

## I  PERFORMANCE OF VARIOUS ENCODING MODELS USING DIFFERENT INPUTS

### I.1  VOXELWISE $r$ VALUES FROM DIFFERENT ENCODING MDOELS AND STIMULI

Figures 9, 10, and 11 each represent the voxelwise correlation ($r$) values using various encoders and inputs for subjects S1, S2, and S3, respectively.

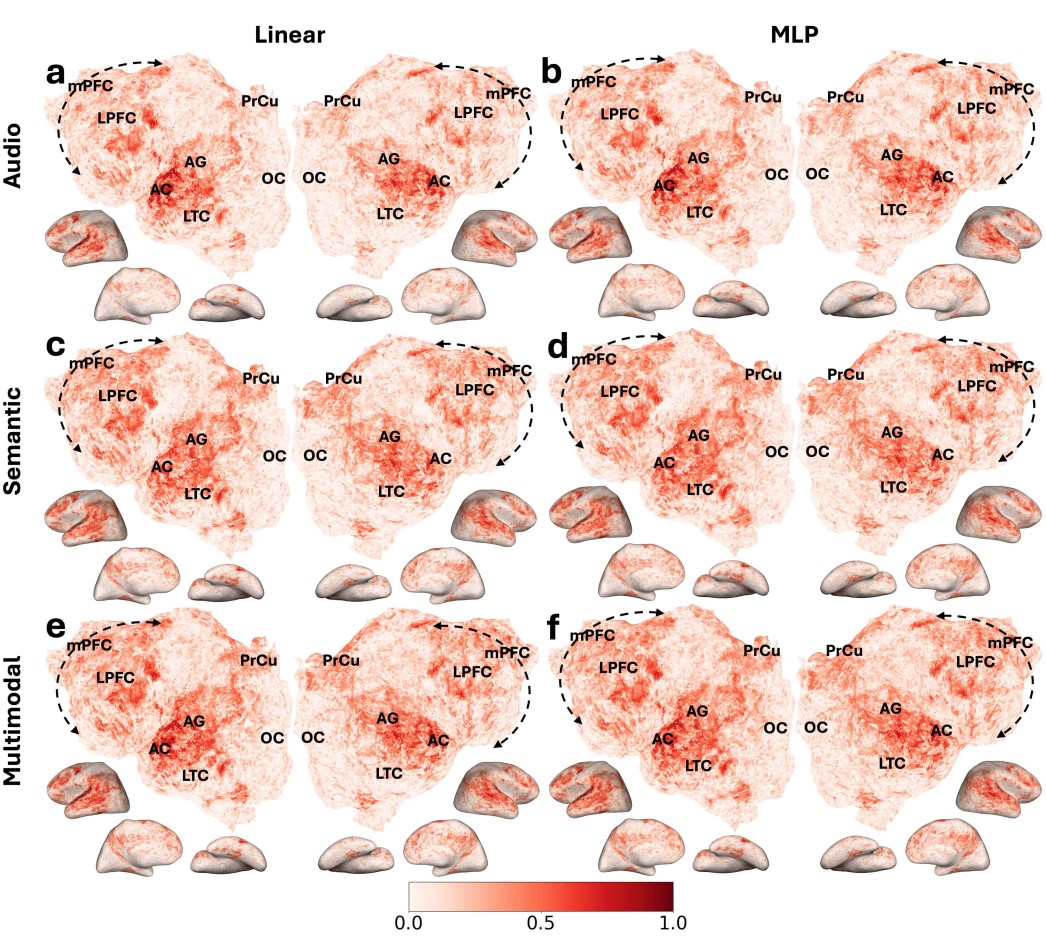

Figure 9: Voxelwise $r$ values for Subject S1 across different input modalities and encoding models. Rows show audio-only (a,b), semantic-only (c,d), and multimodal (e,f) inputs. Columns compare Linear (left) and MLP (right) encoders. Warmer colors indicate higher prediction accuracy.

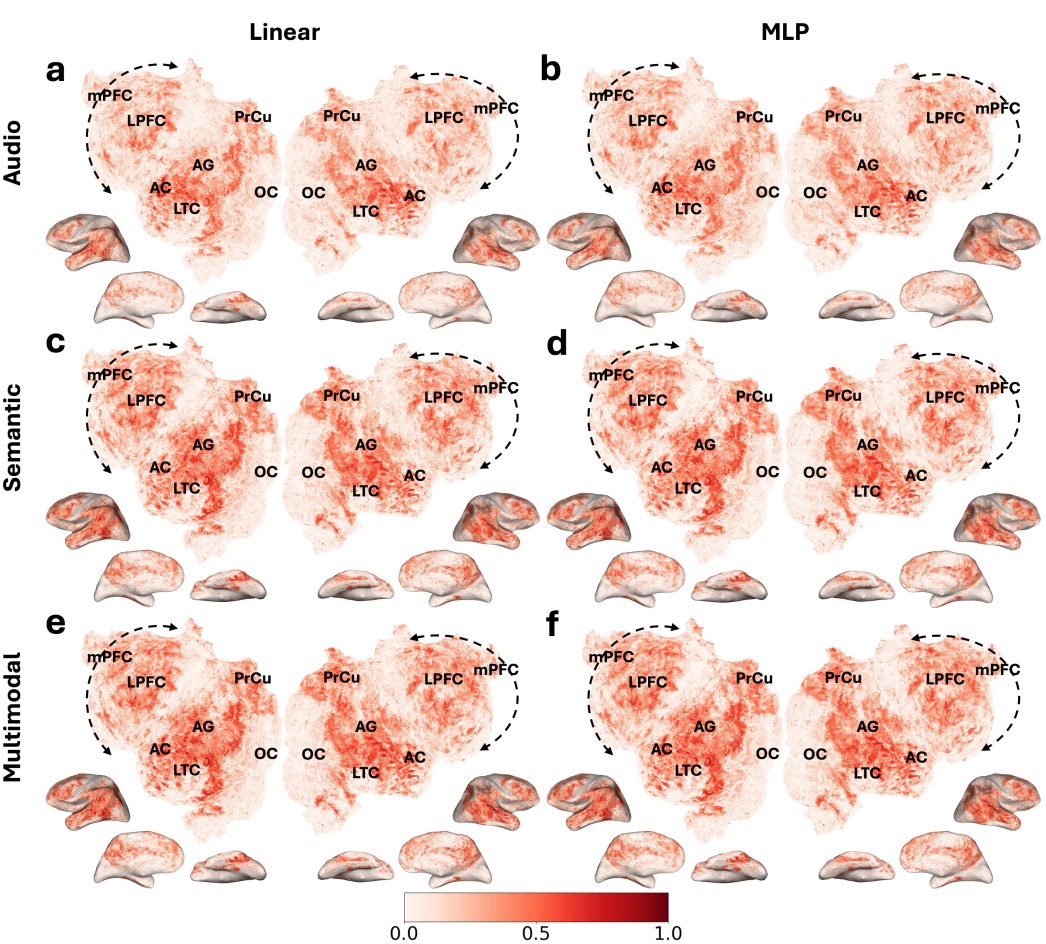

Figure 10: Voxelwise $r$ values for Subject S2 across different input modalities and encoding models. Rows show audio-only (a,b), semantic-only (c,d), and multimodal (e,f) inputs. Columns compare Linear (left) and MLP (right) encoders. Warmer colors indicate higher prediction accuracy.

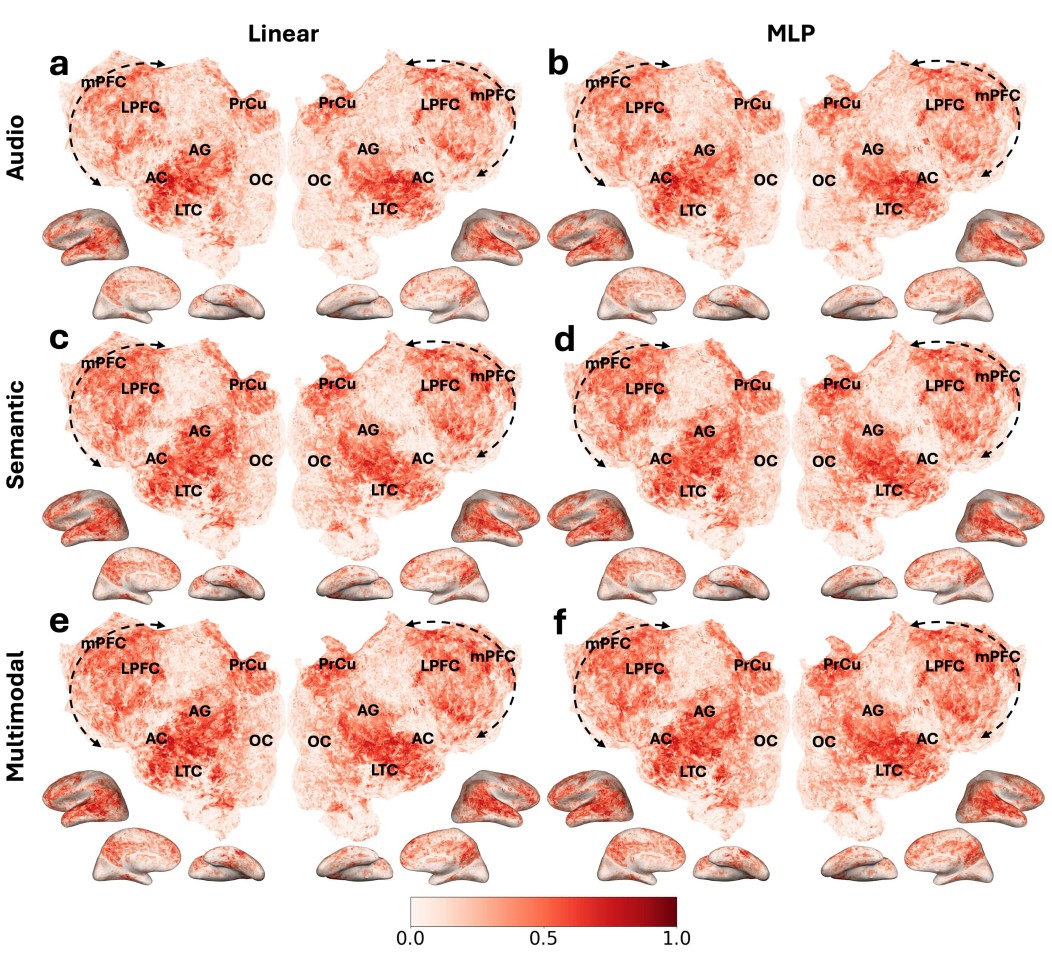

Figure 11: Voxelwise $r$ values for Subject S3 across different input modalities and encoding models. Rows show audio-only (a,b), semantic-only (c,d), and multimodal (e,f) inputs. Columns compare Linear (left) and MLP (right) encoders. Warmer colors indicate higher prediction accuracy.

## I.2 ROI-WISE $r$ VALUES FROM DIFFERENT ENCODING MODELS AND STIMULI

Figure 12 shows the $r$ value for different encoding models and stimuli averaged across subjects.

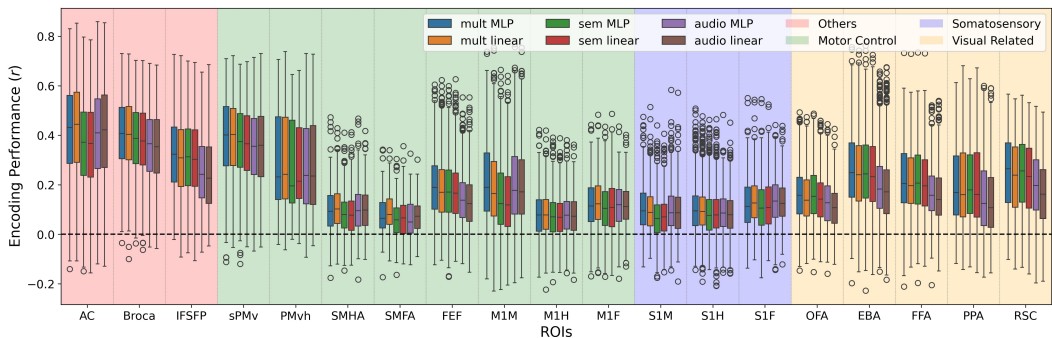

Figure 12: Box plot showing $r$ across different regions of interest (ROIs), where the $r$ values are aggregated over all subjects. *multi* refers to multimodal, and *sem* refers to semantic encoders. ROIs are grouped and color-coded by their functions.

### I.3 VOXELWISE $CC_{norm}$ VALUES FROM DIFFERENT ENCODING MDOELS AND STIMULI

Figures 13, 14, and 15 each represent the normalized voxelwise correlation ($CC_{norm}$) values using various encoders and inputs for subjects S1, S2, and S3, respectively.

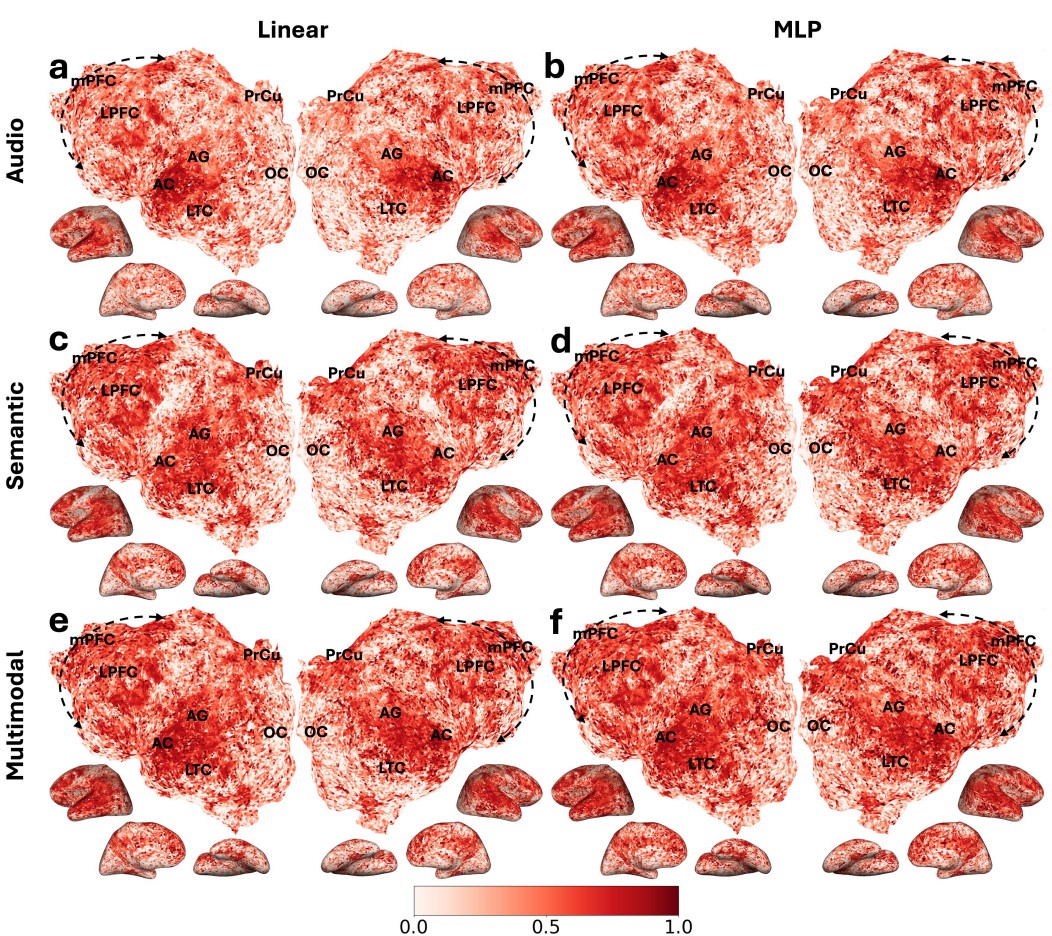

Figure 13: Voxelwise $CC_{norm}$ values for Subject S1 across different input modalities and encoding models. Rows show audio-only (a,b), semantic-only (c,d), and multimodal (e,f) inputs. Columns compare Linear (left) and MLP (right) encoders. Warmer colors indicate higher prediction accuracy.

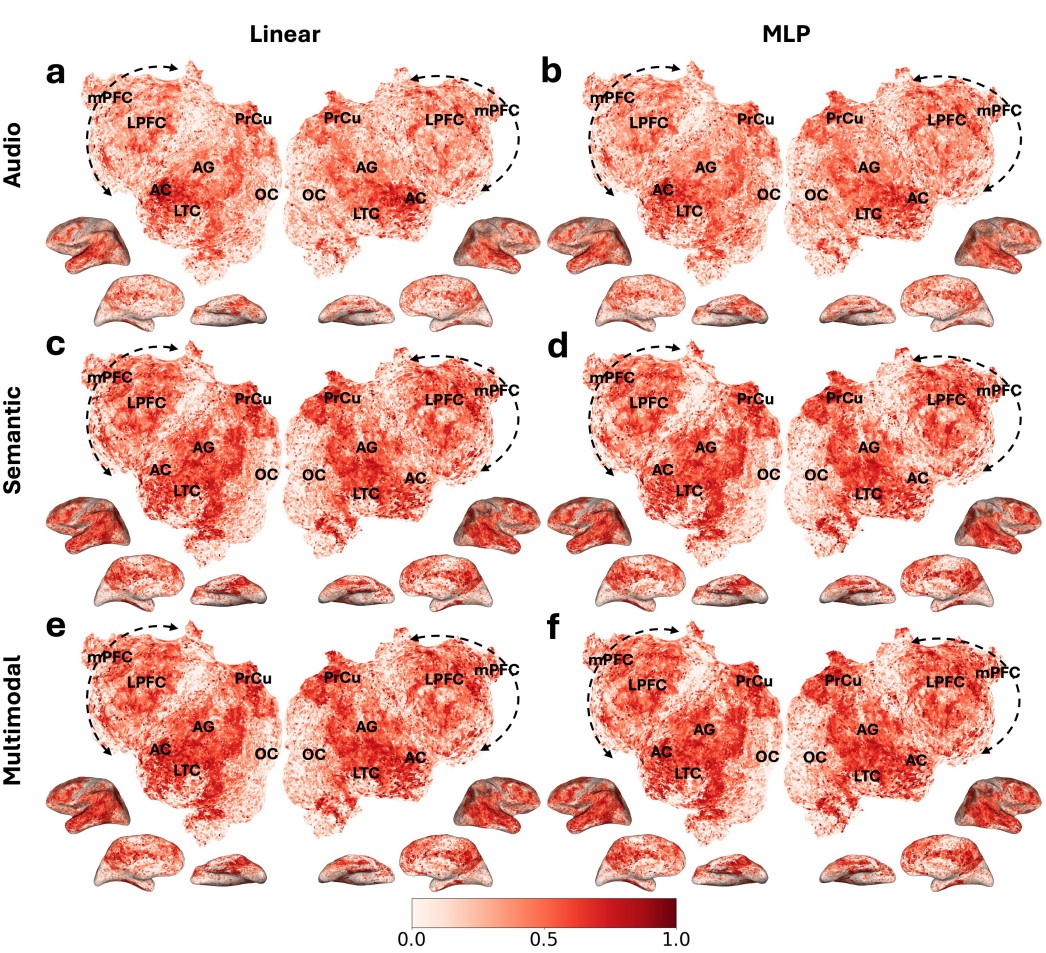

Figure 14: Voxelwise $CC_{norm}$ values for Subject S2 across different input modalities and encoding models. Rows show audio-only (a,b), semantic-only (c,d), and multimodal (e,f) inputs. Columns compare Linear (left) and MLP (right) encoders. Warmer colors indicate higher prediction accuracy.

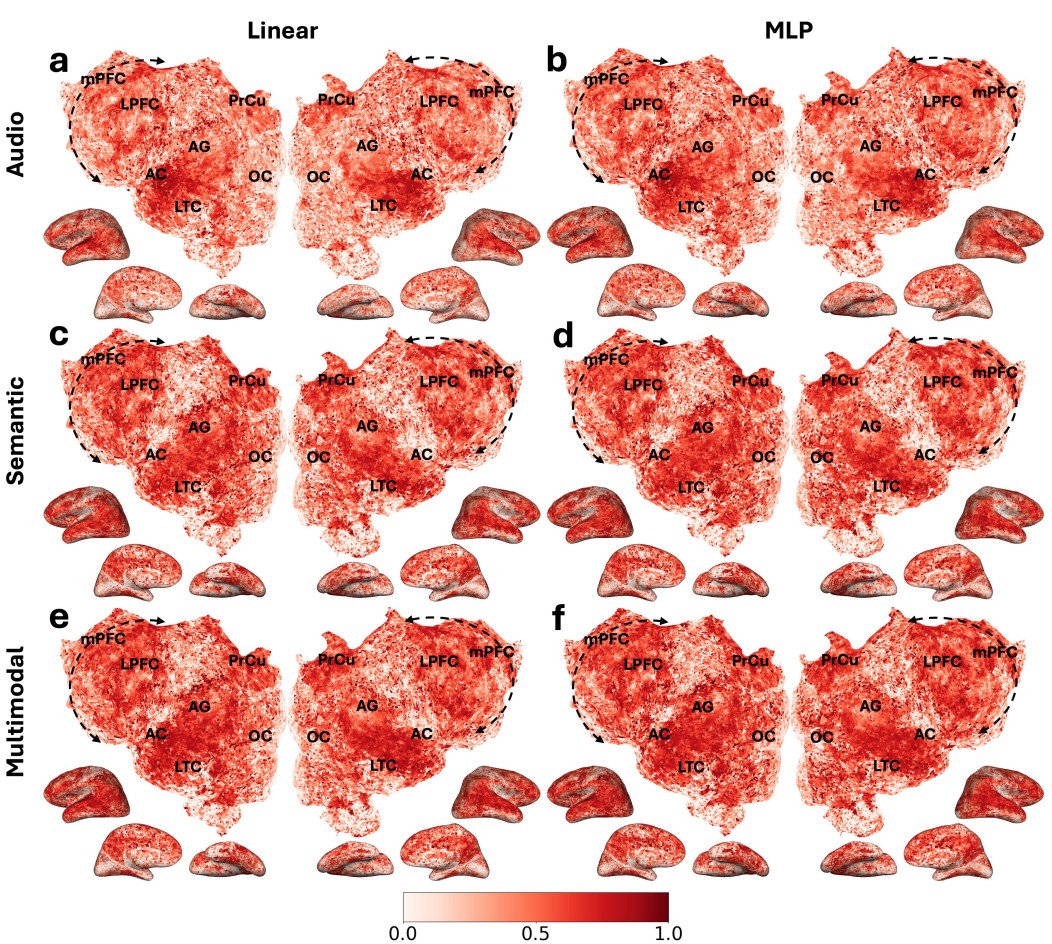

Figure 15: Voxelwise $CC_{norm}$ values for Subject S3 across different input modalities and encoding models. Rows show audio-only (a,b), semantic-only (c,d), and multimodal (e,f) inputs. Columns compare Linear (left) and MLP (right) encoders. Warmer colors indicate higher prediction accuracy.

## J IMPROVEMENTS FROM NONLINEARITY

### J.1 LAYERWISE PERFORMANCE INCREASES FROM MLP

Figure 16 shows that MLP improves encoding performance for both language and audio models, regardless of what layer is used for the MLP encoding model.

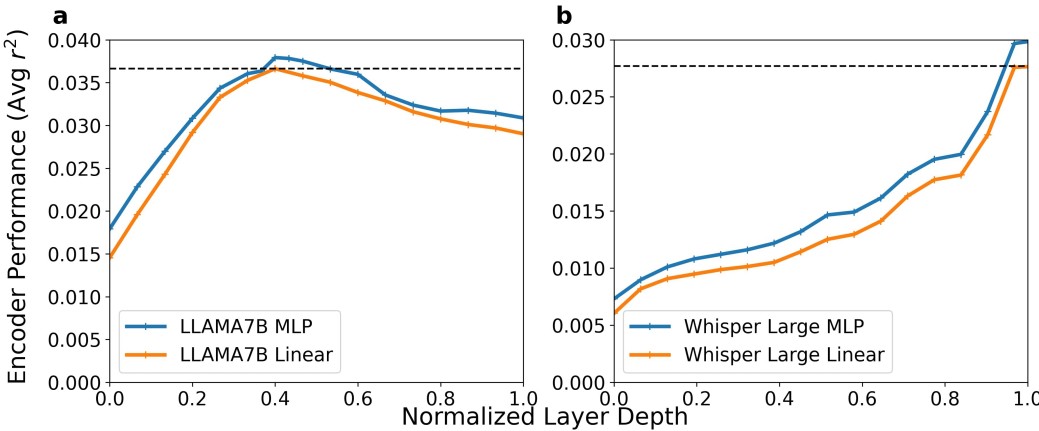

Figure 16: Average voxel-wise $r^2$ values, computed as the mean across three subjects, for each layer of the (a) language (LLAMA7B) and (b) audio (Whisper Large) models. Comparisons are shown between the MLP and linear encoders, and dashed black lines indicate the best performance for linear encoders

### J.2 VOXELWISE IMPROVEMENTS FROM MLP ($r$ ANALYSIS)

Figures 17, 18, and 19 each represent the performance improvements in voxelwise correlation values for semantic, audio, and multimodal inputs, respectively, for each subject.

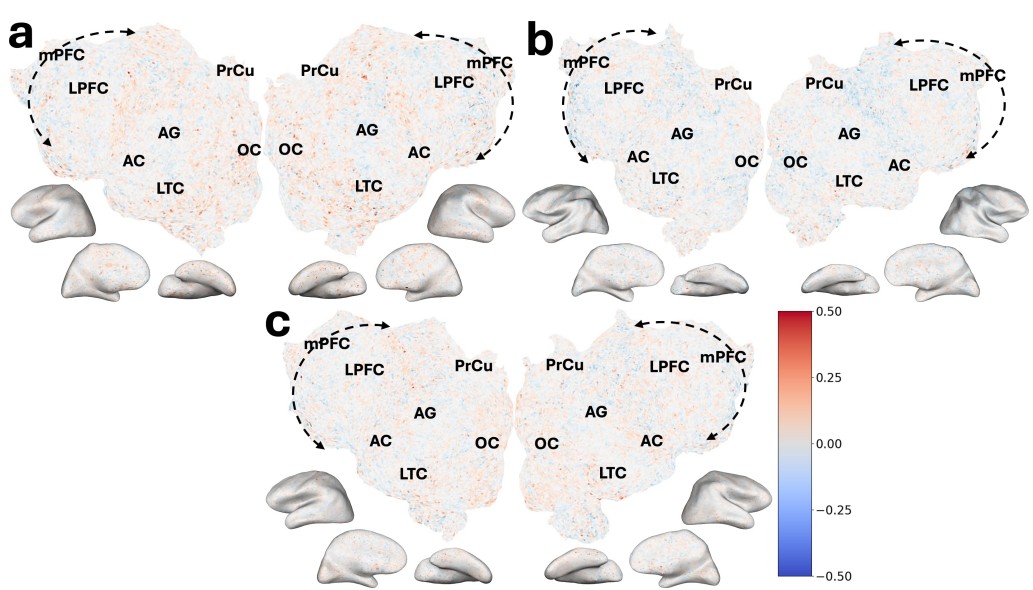

Figure 17: Encoding model performance improvements. (a-c) Voxelwise $\Delta r$ (MLP performance minus linear performance) for semantic input for subjects S1, S2, S3, respectively. Positive values indicate MLP outperformance.

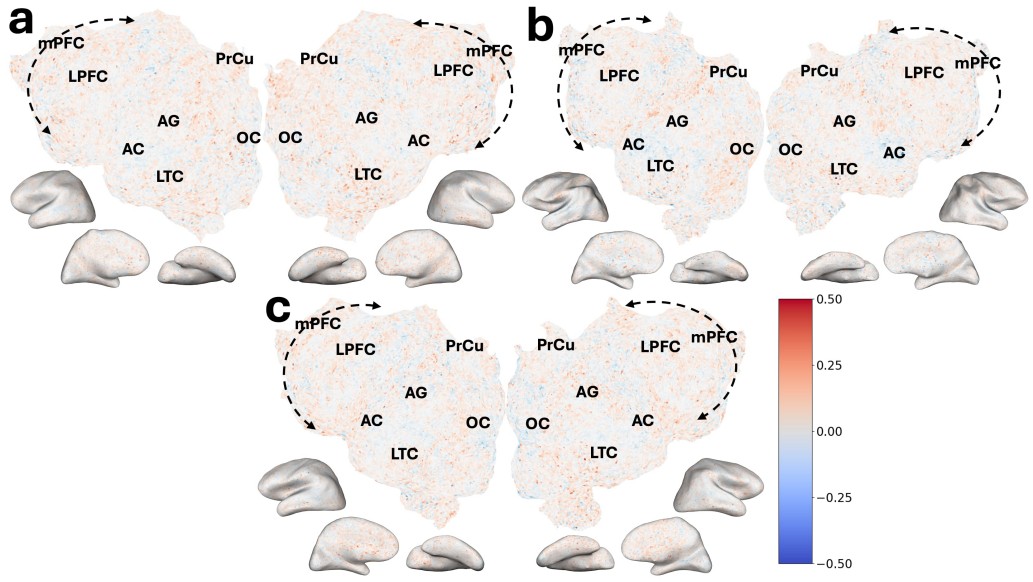

Figure 18: Encoding model performance improvements. (a-c) Voxelwise $\Delta r$ (MLP performance minus linear performance) for audio input for subjects S1, S2, S3, respectively. Positive values indicate MLP outperformance.

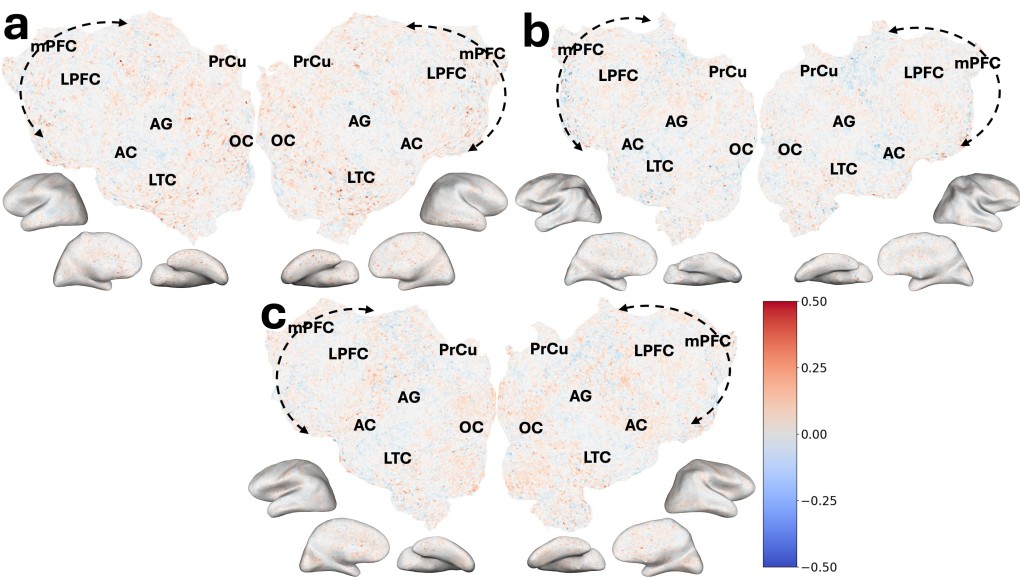

Figure 19: Encoding model performance improvements. (a-c) Voxelwise $\Delta r$ (MLP performance minus linear performance) for multimodal input for subjects S1, S2, S3, respectively. Positive values indicate MLP outperformance.

### J.3 VOXELWISE IMPROVEMENTS FROM MLP ($CC_{norm}$ ANALYSIS)

Figures 21, 20, and 22 each represent the performance improvements in voxelwise $CC_{norm}$ values for semantic, audio, and multimodal inputs, respectively, for each subject. The improvements are more pronounced with $CC_{norm}$ compared to $r$ as noise is taken into account.

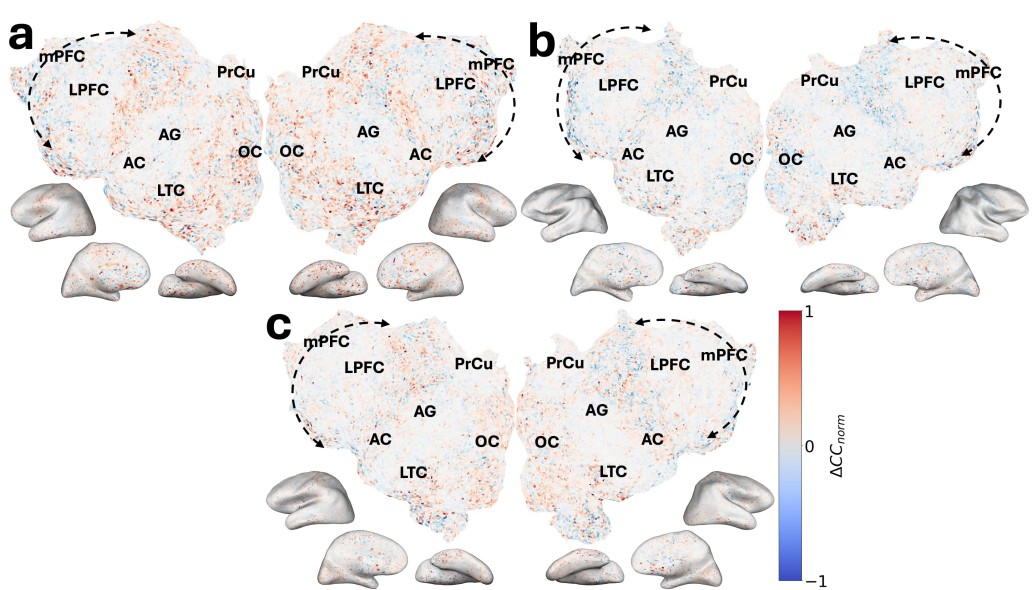

Figure 20: Encoding model performance improvements. (a-c) Voxelwise $\Delta CC_{norm}$ (MLP performance minus linear performance) for semantic input for subjects S1, S2, S3, respectively. Positive values indicate MLP outperformance.

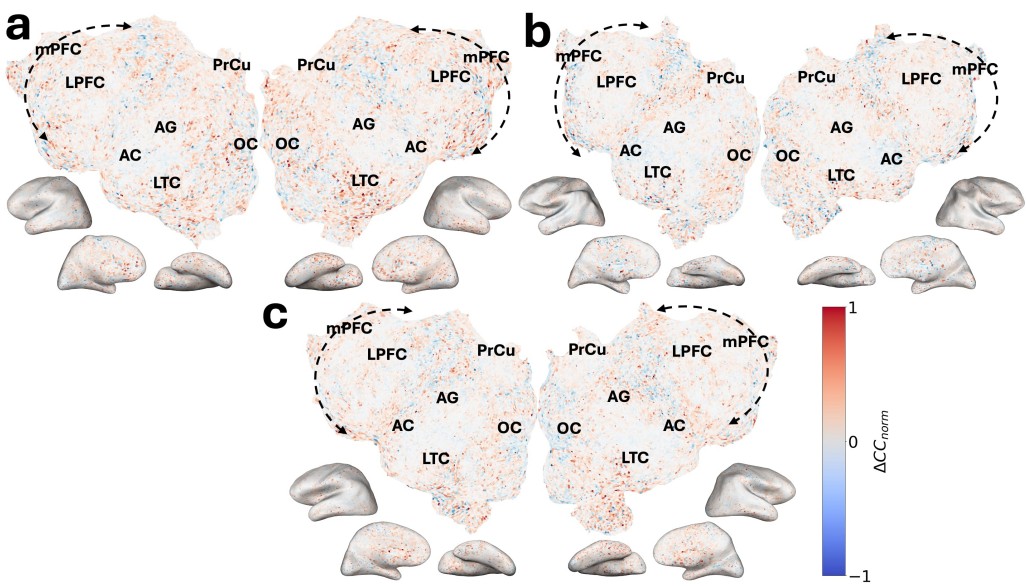

Figure 21: Encoding model performance improvements. (a-c) Voxelwise $\Delta CC_{norm}$ (MLP performance minus linear performance) for audio input for subjects S1, S2, S3, respectively. Positive values indicate MLP outperformance.

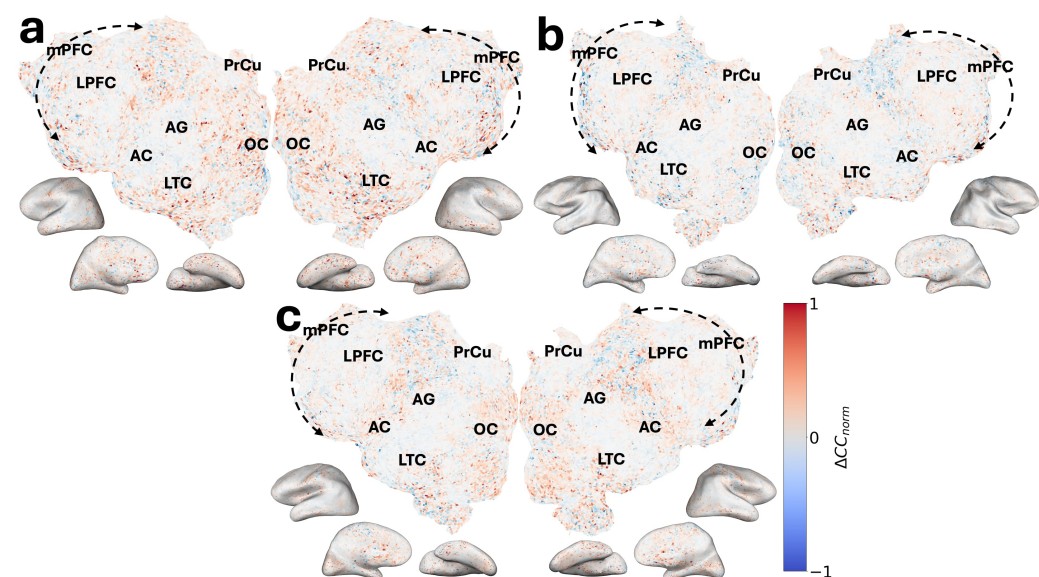

Figure 22: Encoding model performance improvements. (a-c) Voxelwise $\Delta CC_{norm}$ (MLP performance minus linear performance) for multimodal input for subjects S1, S2, S3, respectively. Positive values indicate MLP outperformance.

### J.4 BETTER SPATIO-TEMPORAL COMPARTMENTALIZATION OF BRAIN FUNCTION

To compare the performance between Whisper and LLAMA models, we define the Relative Error Difference (RED) for each voxel $v$ at time $t$ as:

$$\text{RED}(v,t) = |f_{\text{semantic}}(v,t) - y(v,t)| - |f_{\text{audio}}(v,t) - y(v,t)|$$

where $f_{\text{semantic}}(v,t)$ is the prediction from the semantic encoding model for voxel $v$ at time $t$, $f_{\text{audio}}(v,t)$ is the prediction from the audio encoding model for voxel $v$ at time $t$, and $y(v,t)$ represents the true value at voxel $v$ and time $t$. A positive RED value indicates that the audio model outperforms the semantic model at that specific voxel and time, while a negative value indicates that the semantic model performs better.

In this analysis, we computed the RED between Whisper and LLAMA models for each voxel $v$ at a given time $t$. For each region of interest (ROI), the average RED is calculated as:

$$\text{RED}_{\text{ROI}}(t) = \frac{1}{N} \sum_{v \in \text{ROI}} \text{RED}(v,t)$$

Where $N$ is the number of voxels in the ROI. The correlation matrices were then computed over these ROI time series for both linear and nonlinear (MLP) encoders (Figure 23 (b, c)). A high correlation between two ROIs indicates that their semantic/audio processing temporal dynamics are similar over time.

For comparison, functional connectivity (FC) was also computed using the average fMRI signal for each voxel (Figure 23 a). Hierarchical clustering was then performed on the correlation matrices, producing the dendrograms in panels (d-f).

As shown in Figure 23, panel (d) does not exhibit meaningful compartmentalization, indicating that the ROIs are not functionally clustered based on FC. However, the correlation matrices derived from RED (panels b, c) demonstrate clear block-diagonal structures, suggesting better functional compartmentalization. The dendrograms in panels (e, f) show that the ROIs cluster according to

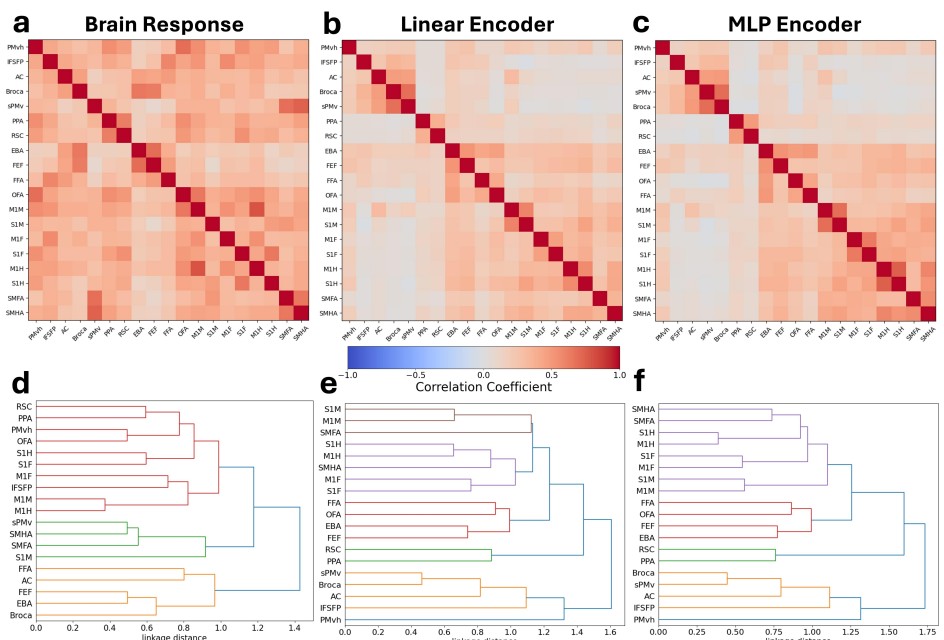

Figure 23: Spatio-temporal clustering based on Relative Error Difference (RED) between semantic and audio encoding models. Panels (a-c) display correlation matrices representing the temporal relationships between regions of interest (ROIs). For consistency, all the ROIs in (a,b,c) are ordered according to the most optimal ordering for (c). Panel (a) shows the functional connectivity (FC) matrix, calculated from the average fMRI signals. Panel (b) presents the correlation matrix from Relative Error Difference between Whisper and LLAMA using linear encoders, while panel (c) uses nonlinear (MLP) encoders, showing better functional compartmentalization with stronger block-diagonal structures. Panels (d-f) depict hierarchical clustering dendrograms derived from the correlation matrices in panels (a-c). Panel (d), based on FC, shows no clear compartmentalization of ROIs. Panel (e), based on linear encoders, show almost perfect functional clustering, though with inaccuracies (e.g., SMFA clustered with S1M/M1M). Panel (f), based on nonlinear (MLP) encoders, achieves better functional clustering, correctly grouping motor-related regions. The modularity Q values confirm this improvement: FC (a) scored 0.068, linear encoders (b) scored 0.145, and nonlinear encoders (c) scored 0.155, highlighting the advantage of nonlinear encoders for functional organization.

their functional roles, where the somatosensory and motor areas, visual areas, and auditory areas are grouped (even lower levels are grouped well (M1H/S1H, M1M/S1M, M1F/S1F, SMHA/SMFA, Broca/sPMv are grouped)) with nonlinear (MLP) models (f) achieving more accurate clustering than linear models (e). Specifically, panel (e) incorrectly clusters SMFA with S1M and M1M, whereas panel (f) correctly clusters SMHA and SMFA together before clustering them with other sensory and motor-related regions.

This study presents a novel approach, as it is the first to use fMRI speech encoding models to group ROIs based not only on spatial dynamics but also on their temporal processing dynamics. Traditionally, voxel-wise functional classification or grouping has been the norm in fMRI analysis, focusing solely on static (spatial) relationships. However, here with the help of fMRI encoders, we incorporate both spatial and temporal information, allowing for a more comprehensive, dynamic view of brain function, especially in the context of semantic and auditory encoding.

In summary, using nonlinear (MLP) models leads to better functional compartmentalization. In fact, modularity Q values further confirm this: FC (a) scored 0.068, linear encoders (b) scored 0.145, and nonlinear encoders (c) scored 0.155, highlighting the improved functional clustering achieved with better encoders.

## K IMPROVEMENTS FROM MULTIMODALITY

### K.1 VOXELWISE IMPROVEMENTS FROM MULTIMODALITY ($r$ ANALYSIS)

This section shows the subject-wise plots of voxelwise $\Delta r$ between multimodal linear/MLP and semantic/audio linear models (Figure 25, Figure 26). We observe consistent patterns of improvement when using multimodal models. For direct comparison with Figure 2 (which plots $\Delta CC_{norm}$), we provide here the provide the equivalent plot with $\Delta r$ in Figure 24.

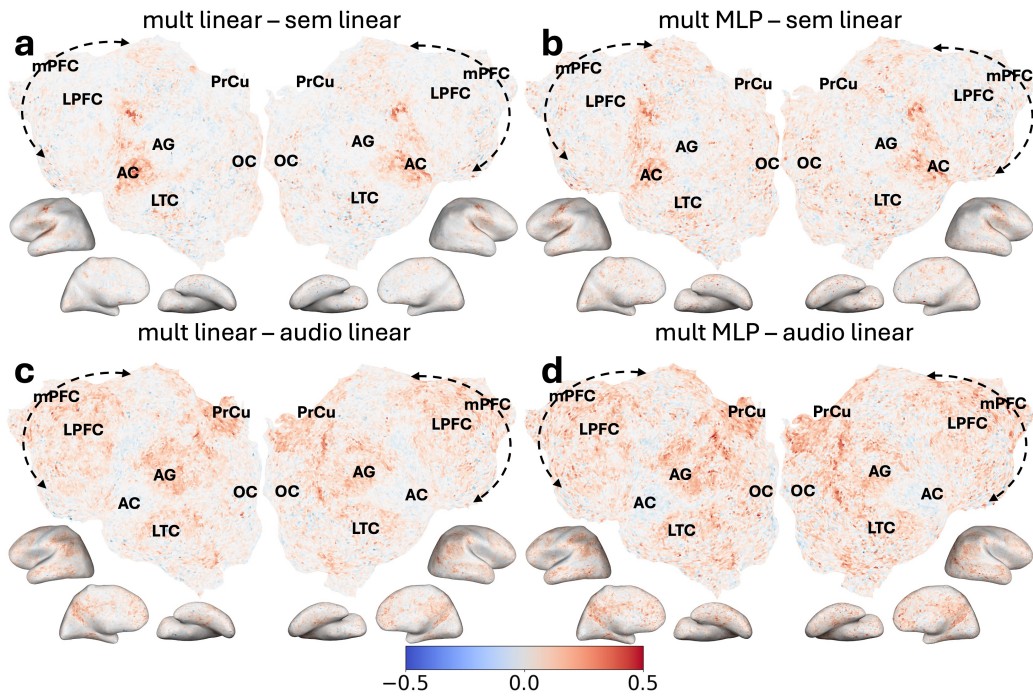

Figure 24: Multimodality improvement ($\Delta r$) in encoding models. Panels (a)-(d) display voxelwise $\Delta r$ values of a single subject (S1), with warmer colors indicating regions where multimodal models outperform linear models. *mut*, *sem*, each refer to multimodal and semantic encoders.

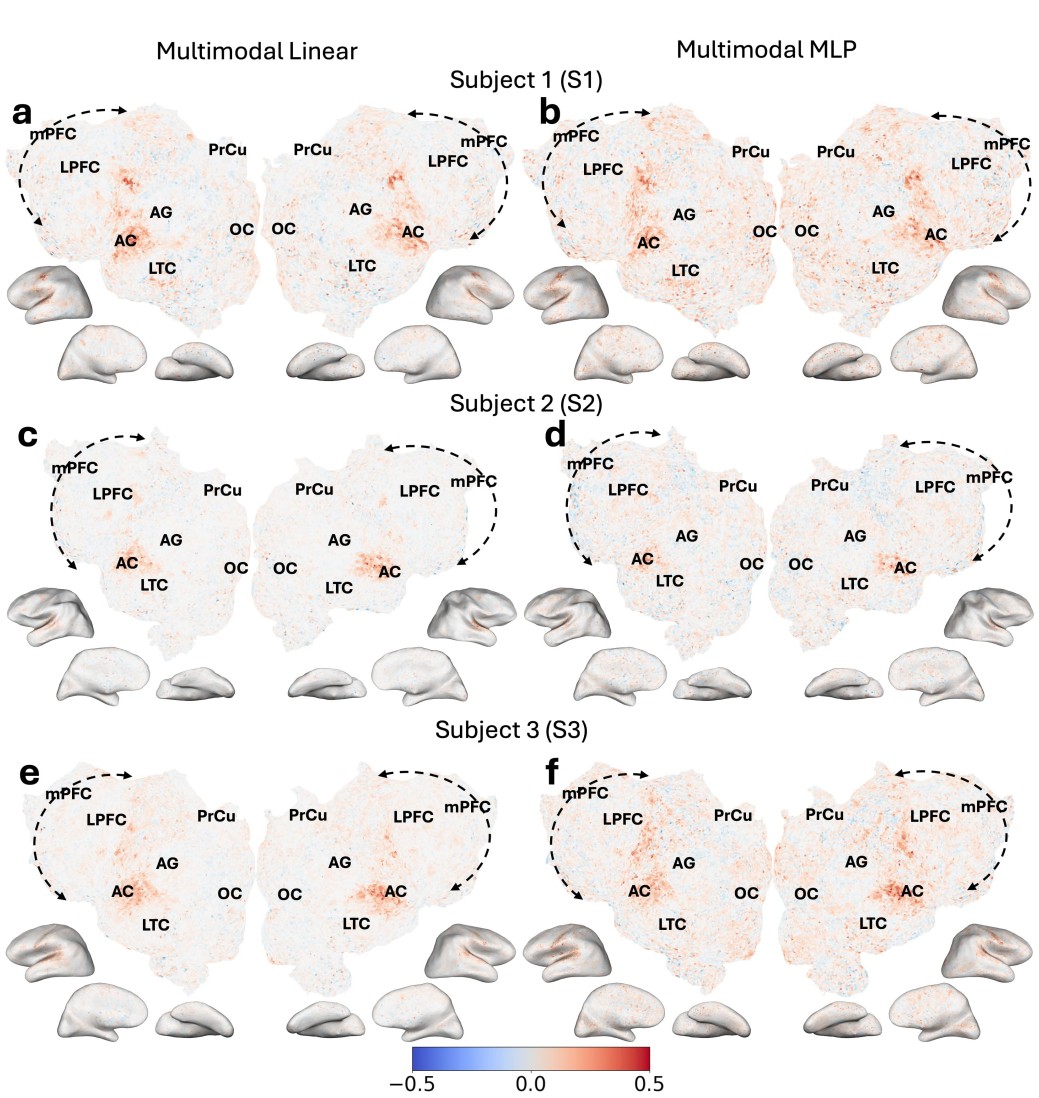

Figure 25: Subject-wise voxelwise $\Delta r$ plots of multimodal models compared to semantic models. Panels (a-f) display voxelwise $\Delta r$ values comparing multimodal and unimodal models across three subjects. Panels a, c, e show the difference between multimodal linear and semantic linear models, while panels b, d, f compare multimodal MLP and semantic linear models. Each row represents a different subject: Subject 1 (S1) in panels a-b, Subject 2 (S2) in panels c-d, and Subject 3 (S3) in panels e-f. Warmer colors indicate regions where the multimodal models outperform the unimodal linear models in prediction accuracy. The spatial patterns highlight enhanced encoding performance in key areas associated with semantic and auditory processing, such as the medial prefrontal cortex (mPFC), angular gyrus (AG), precuneus (PrCu), and lateral temporal cortex (LTC), emphasizing the benefits of multimodal models in capturing complex brain activity.

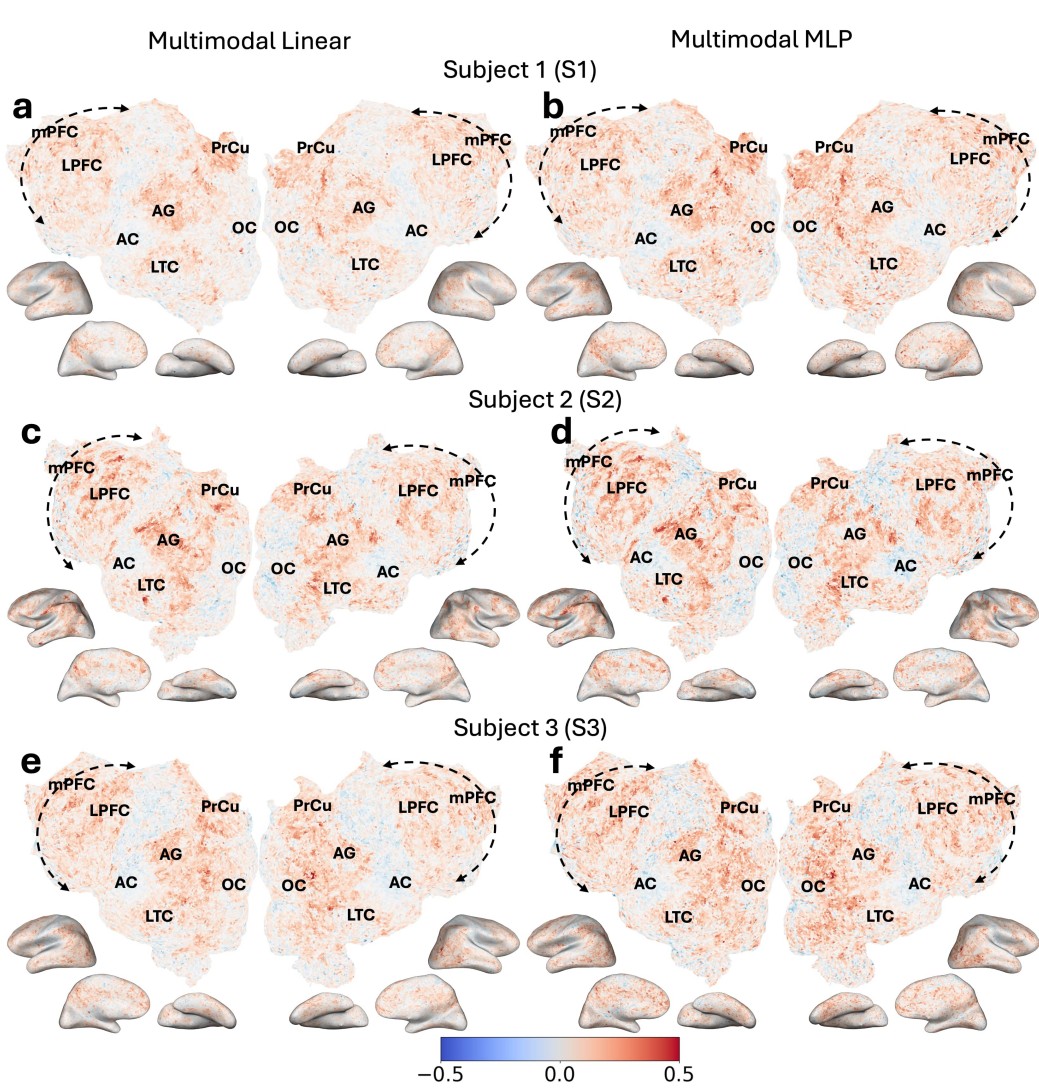

Figure 26: Subject-wise voxelwise $\Delta r$ plots of multimodal models compared to audio models. Panels (a-f) display voxelwise $\Delta r$ values comparing multimodal and unimodal models across three subjects. Panels a, c, e show the difference between multimodal linear and audio linear models, while panels b, d, f compare multimodal MLP and audio linear models. Each row represents a different subject: Subject 1 (S1) in panels a-b, Subject 2 (S2) in panels c-d, and Subject 3 (S3) in panels e-f. Warmer colors indicate regions where the multimodal models outperform the unimodal linear models in prediction accuracy.

## K.2 VOXELWISE IMPROVEMENTS FROM MULTIMODALITY ($CC_{norm}$ ANALYSIS)

This section shows the subject-wise plots of voxelwise $\Delta CC_{norm}$ between multimodal linear/MLP and semantic/audio linear models (Figure 28, Figure 28). We observe consistent patterns of improvement when using multimodal models. The improvements are more noticable with $CC_{norm}$ compared to $r$ as noise is taken into account.

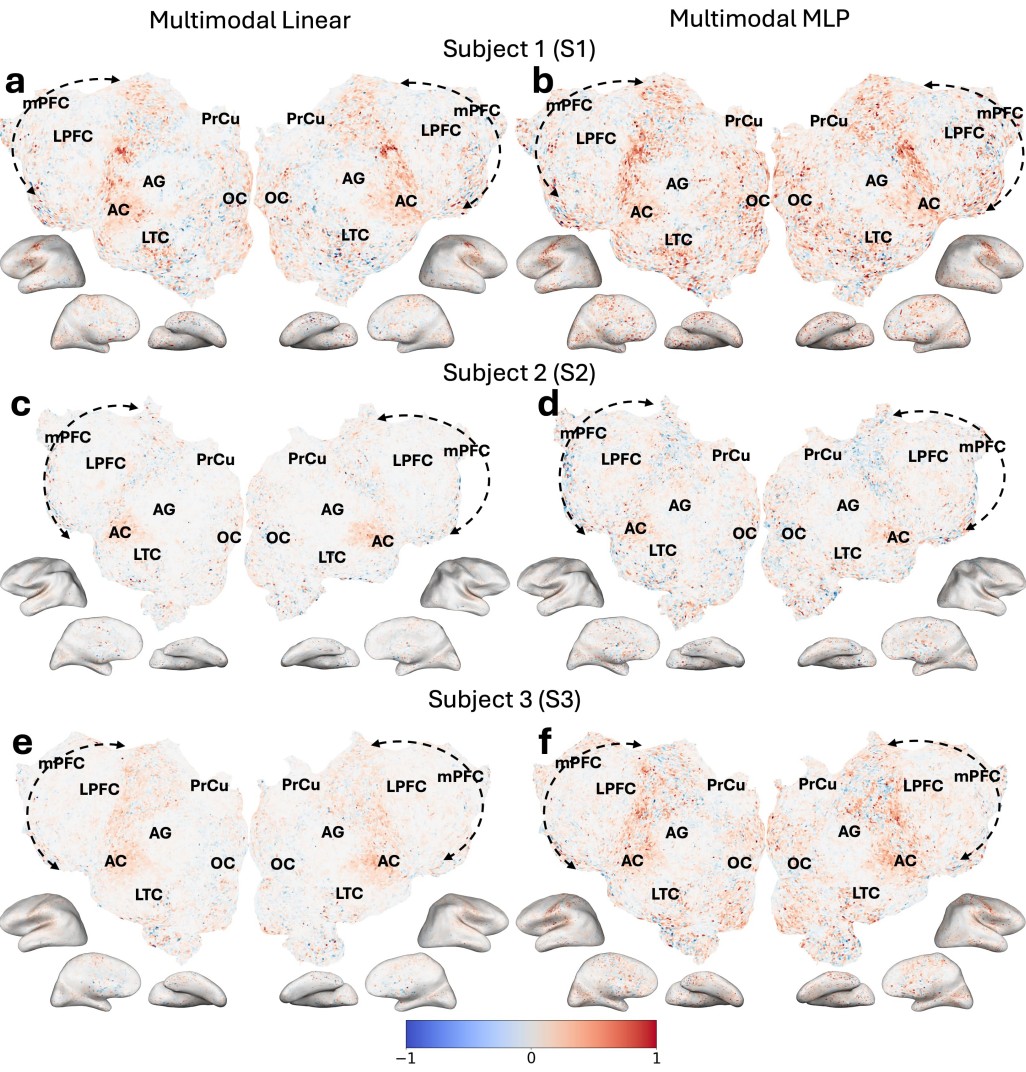

Figure 27: Subject-wise voxelwise $\Delta CC_{norm}$ plots of multimodal models compared to semantic models. Panels (a-f) display voxelwise $\Delta CC_{norm}$ values comparing multimodal and unimodal models across three subjects. Panels a, c, e show the difference between multimodal linear and semantic linear models, while panels b, d, f compare multimodal MLP and semantic linear models. Each row represents a different subject: Subject 1 (S1) in panels a-b, Subject 2 (S2) in panels c-d, and Subject 3 (S3) in panels e-f. Warmer colors indicate regions where the multimodal models outperform the unimodal linear models in prediction accuracy.

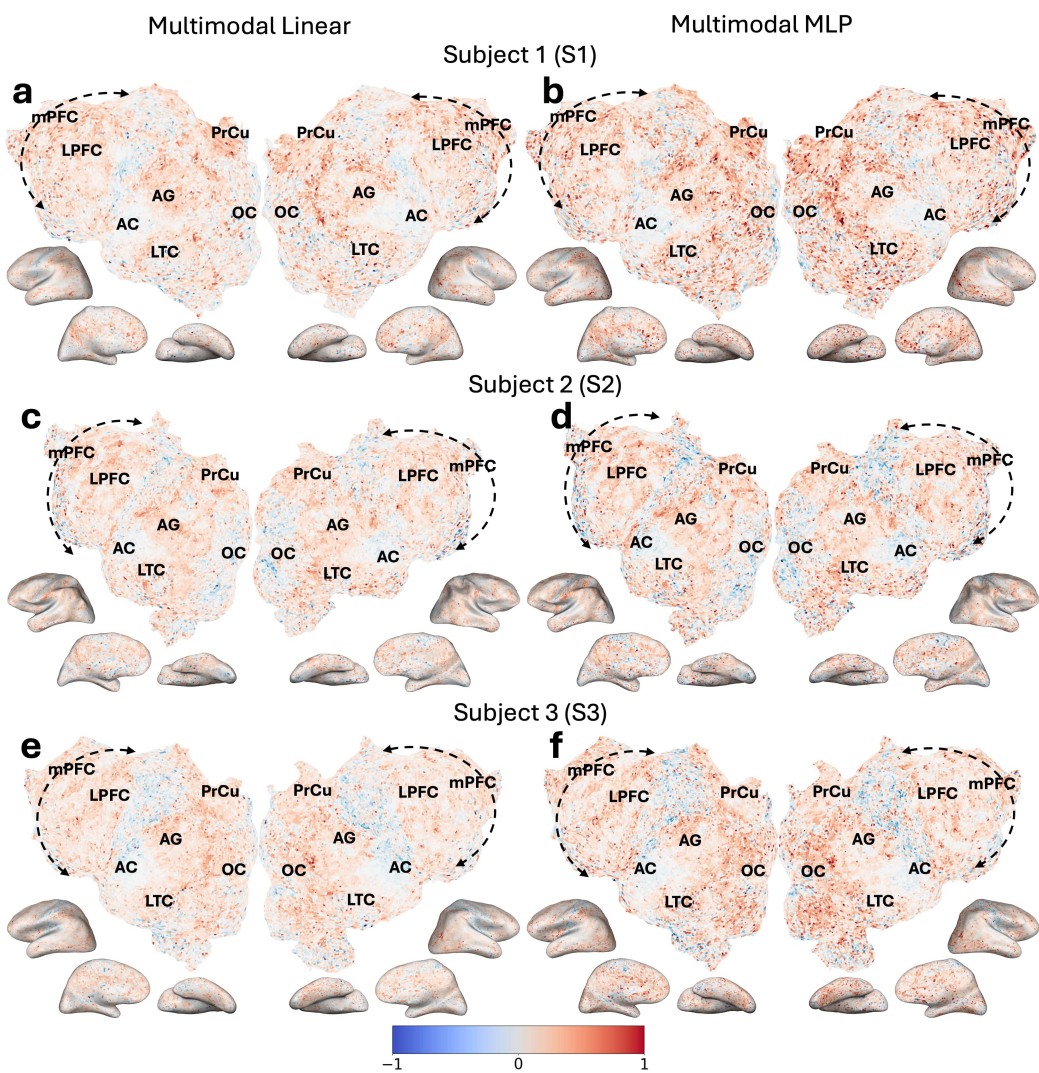

Figure 28: Subject-wise voxelwise $\Delta CC_{norm}$ plots of multimodal models compared to audio models. Panels (a-f) display voxelwise $\Delta CC_{norm}$ values comparing multimodal and unimodal models across three subjects. Panels a, c, e show the difference between multimodal linear and audio linear models, while panels b, d, f compare multimodal MLP and audio linear models. Each row represents a different subject: Subject 1 (S1) in panels a-b, Subject 2 (S2) in panels c-d, and Subject 3 (S3) in panels e-f. Warmer colors indicate regions where the multimodal models outperform the unimodal linear models in prediction accuracy.

## K.3 ROI PREDICTIONS IMPROVEMENTS FROM MULTIMODALITY

This section shows the ROI-wise improvements from using multimodal models (Figure 29)

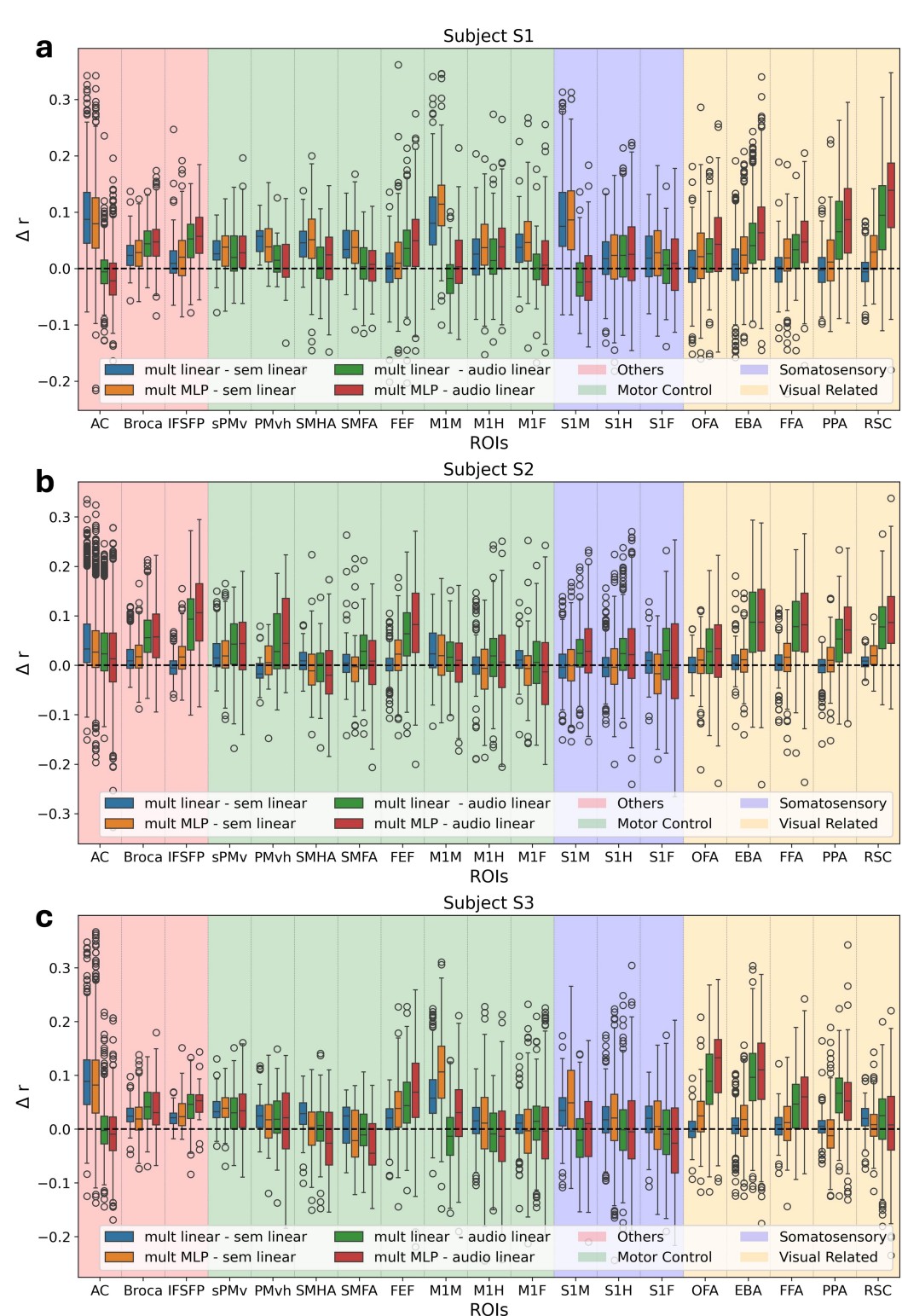

Figure 29: Subject-wise boxplots of performance differences ($\Delta r$) across different ROIs. The comparisons are made between different stimuli and encoding models: multimodal linear and multimodal MLP (mult MLP) models are compared against semantic (sem) and audio linear models. The ROIs are grouped into functional categories.

## L IMPROVEMENTS FROM NONLINEARITY AND MULTIMODALITY

### L.1 VOXELWISE IMPROVEMENTS FROM DIMLP, AND ADDITIONAL IMPROVEMENTS FROM MLP ($r$ ANALYSIS)

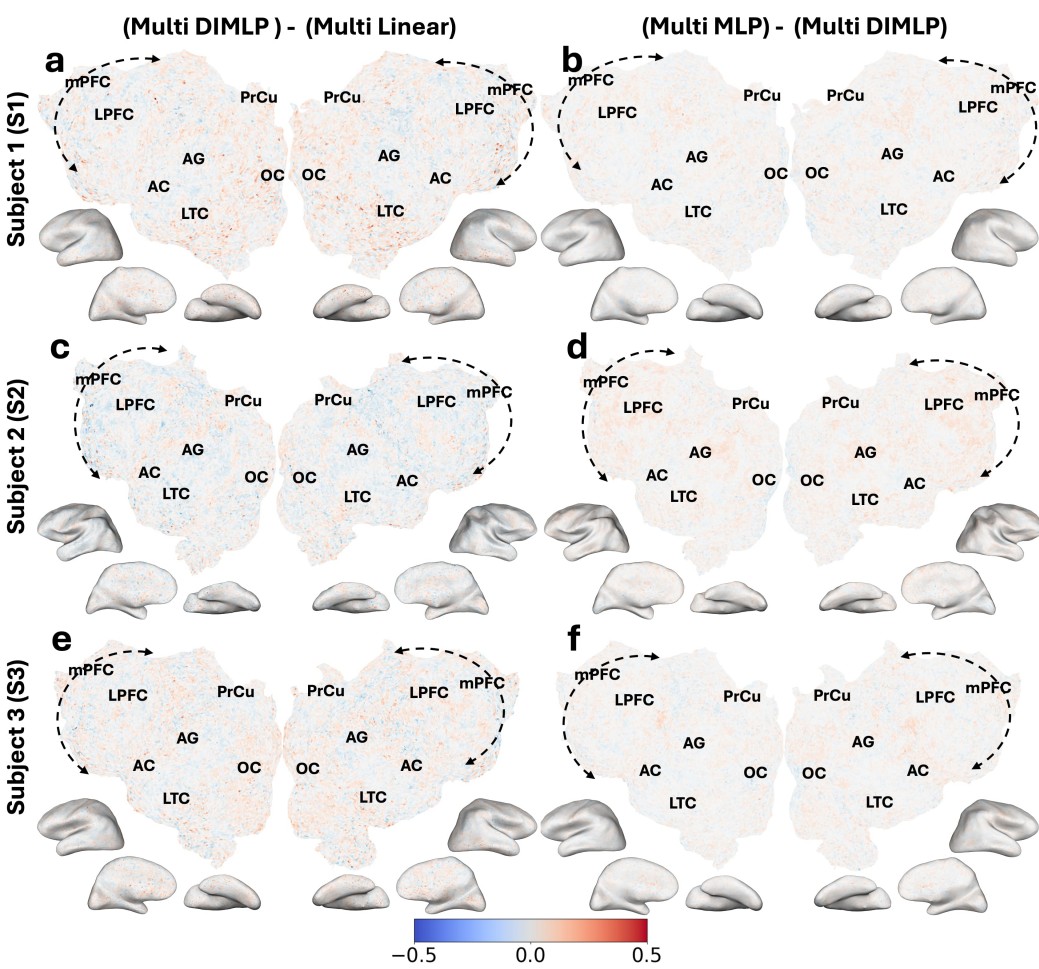

Figure 30: Nonlinearity Enhances Multimodal fMRI Predictions. Panels (a, c, e) show the voxelwise $\Delta r$ values (DIMLP minus linear model), illustrating the improvements achieved through nonlinear processing within each modality, while largely limiting cross-modal interactions. Panels (b, d, f) display voxelwise $\Delta r$ values (Multi MLP minus Multi DIMLP), highlighting the additional benefits of allowing nonlinear interactions between modalities ("Multi" denotes Multimodal). Each row represents the same subject: Subject 1 (S1) in panels a-b, Subject 2 (S2) in panels c-d, and Subject 3 (S3) in panels e-f. Warmer colors indicate regions where the nonlinear models outperform linear models.

### L.2 VOXELWISE IMPROVEMENTS FROM DIMLP, AND ADDITIONAL IMPROVEMENTS FROM MLP ($CC_{norm}$ ANALYSIS)

Figure 31 shows the voxel-wise performance improvements in voxelwise $CC_{norm}$ values when incorporating nonlinear interactions. The improvements are more pronounced with $CC_{norm}$ compared to $r$ as noise is taken into account.

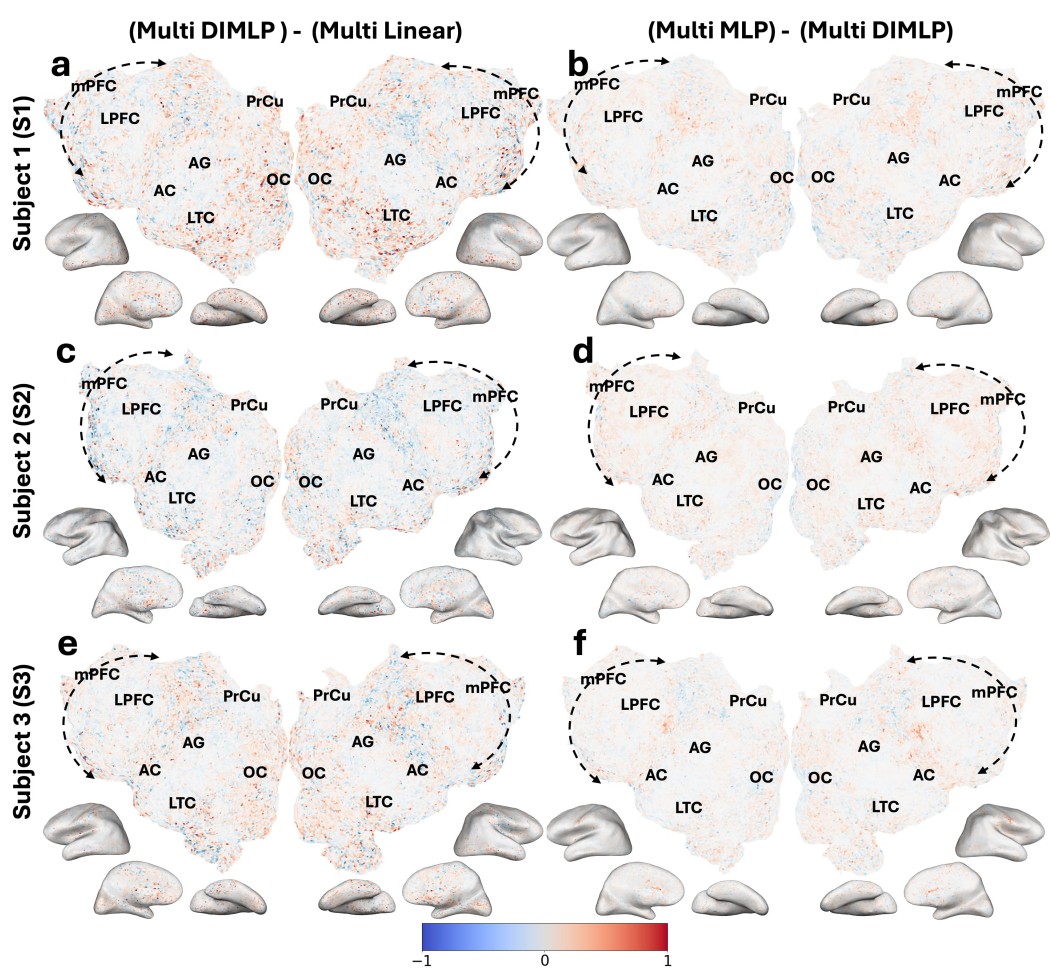

Figure 31: Nonlinearity Enhances Multimodal fMRI Predictions. Panels (a, c, e) show the voxelwise $\Delta CC_{norm}$ values (DIMLP minus linear model), illustrating the improvements achieved through nonlinear processing within each modality, while largely limiting cross-modal interactions. Panels (b, d, f) display voxelwise $\Delta CC_{norm}$ values (Multi MLP minus Multi DIMLP), highlighting the additional benefits of allowing nonlinear interactions between modalities ("Multi" denotes Multi-modal). Each row represents the same subject: Subject 1 (S1) in panels a-b, Subject 2 (S2) in panels c-d, and Subject 3 (S3) in panels e-f. Warmer colors indicate regions where the nonlinear models outperform linear models.

## L.3 ROI-WISE IMPROVEMENTS OF MULTIMODAL DIMLP AND MLP FROM MULTIMODAL LINEAR MODEL

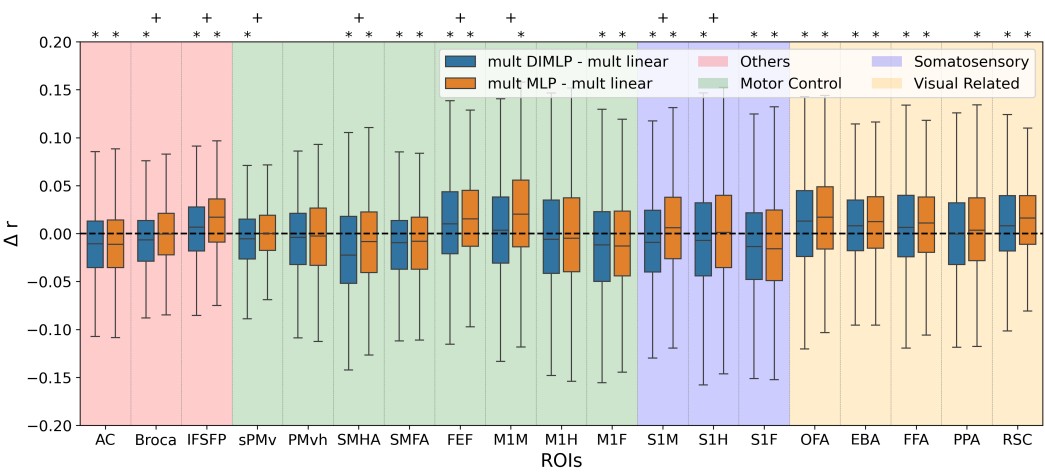

Figure 32: Box plot showing $\Delta r$ across ROIs, where the $\Delta r$ values are aggregated over all subjects. *multi* refers to multimodal, and *sem* refers to semantic encoders, and *DIMLP* refers to Delayed Interaction MLP, where only a *linear* interaction between modalities is allowed. The ROIs are color-coded by function. Regions where $\Delta r > 0$ with a p-value less than 0.05 are indicated by * symbols. Additionally, + symbols denote ROIs where there is a statistically significant difference (p-value < 0.05) between the two models based on a pairwise t-test. Voxelwise and ROI-wise plots for each subjects can be found in Figure 30 (Appendix), and Figure 33 (Appendix), respectively.

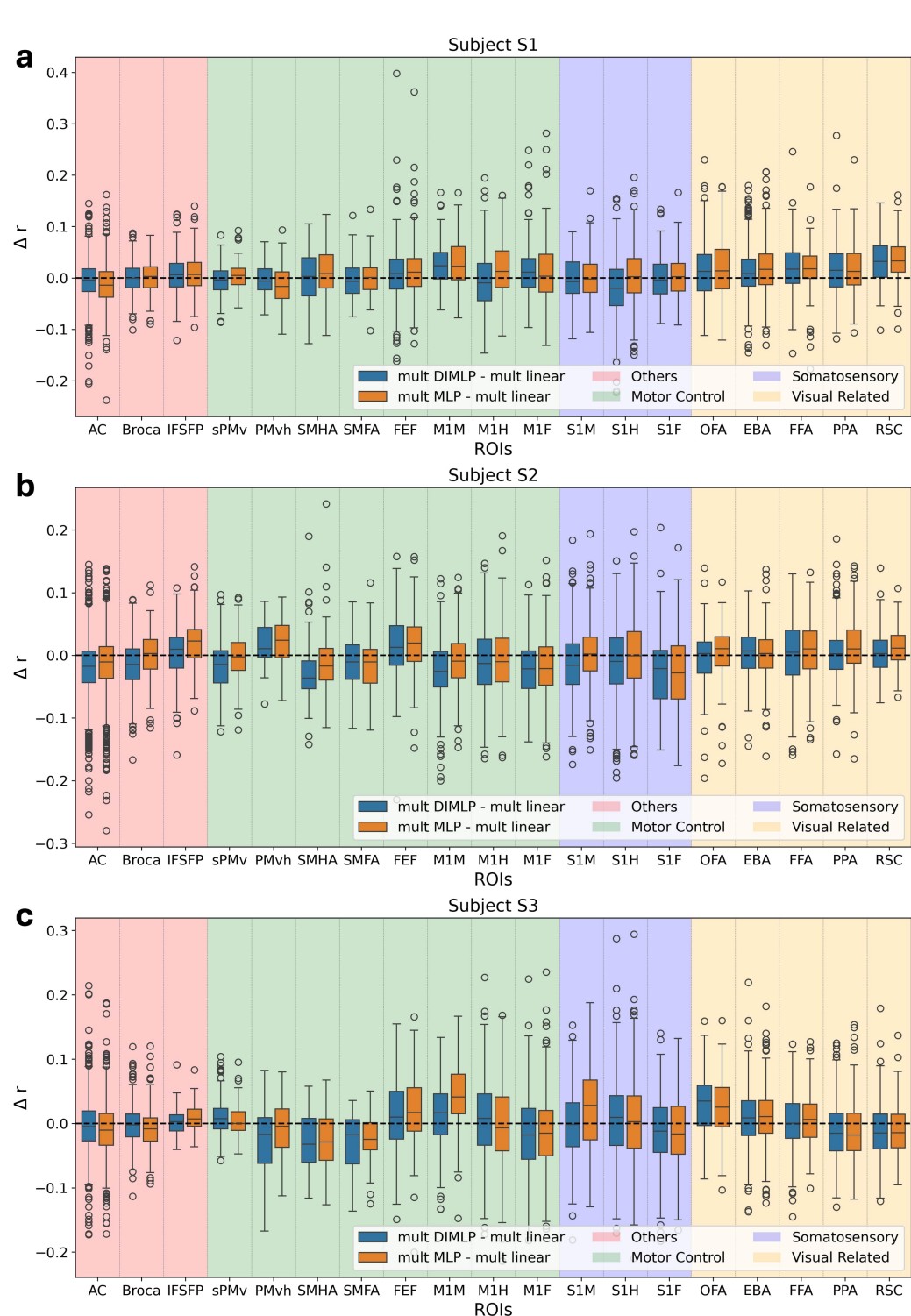

Figure 33: Subject-wise boxplots of voxel-wise differences ($\Delta r$) across different ROIs. The comparisons are made between different encoding models: multimodal MLP and multimodal DIMLP models are compared against multimodal linear models. The ROIs are grouped into functional categories.

## M  Variance partitioning analysis

To quantify the unique contributions of different feature spaces in our nonlinear multimodal encoding models, we employed a variance partitioning analysis similar to de Heer et al. (2017). This approach allowed us to determine how much variance could be uniquely explained by each feature versus that explained by a multiple features. We estimated both the fraction of variance explained by each feature space individually and the fraction that might be equally well explained by combinations of feature spaces.

We show our variance partitioning analysis results in three complementary ways: 1) voxel-wise variance partition results (Appendix M.2), 2) voxel-wise plots showing the largest variance partition for each voxel (Appendix M.3), and 3) ROI-wise Venn diagrams illustrating the distribution of variance explained across different brain regions (Appendix M.4).

For this analysis, we fit models with all possible combinations of feature spaces: two single-feature models (audio and semantic), one model combining both features (semantic-audio), and examined the distribution of variance explained within brain regions. This allowed us to decompose the total explained variance into three components: variance uniquely explained by audio features, variance uniquely explained by semantic features, and variance jointly explained by both feature spaces.

### M.1  Summary of variance partitioning results

Looking at the results of Appendix M.2, we observe that joint variance dominates across most cortical regions, contrasting with de Heer et al. (2017) where semantic only features showed greater dominance. This difference likely stems from our feature choices - whereas de Heer et al. (2017) used spectral and articulatory features that primarily contained information relevant mostly only to auditory cortex, our use of Whisper features provides richer auditory representations that enable better predictions beyond traditional auditory regions. This finding aligns with our earlier argument (Section 3.3.2) that multiple modalities jointly contribute to neural computations across the cortex rather than having one modality dominate.

The dominance pattern of joint variance is consistent both within and near AC, with a notable exception in early auditory regions where audio features show unique contributions. This hierarchical organization suggests that while early AC predominantly processes pure acoustic information, later AC regions integrate both semantic and auditory features for higher-level speech processing. The unique contribution of audio features in early AC is noteworthy as it suggests preservation of modality-specific processing at early sensory stages despite using rich Whisper features.

Also, Appendix M.3 reveals distinct spatial patterns in feature representation across cortical regions. The prefrontal cortex exhibits mixed dominance patterns, showing both joint semantic-audio representation and semantic-only areas. While early auditory cortex shows expected unique audio contributions, we also observe audio-specific representation in motor-sensory mouth areas (M1M, S1M), though this pattern varies across subjects.

The ROI-wise analysis in Appendix M.4 reveals that joint semantic-audio features dominate cortical representation, accounting for approximately 65% of significantly predicted voxels across the entire cortex. Core language-processing regions (AC, Broca's area, sPMv) show particularly strong joint representation (around 80 to 90%), supporting our hypothesis that speech comprehension relies on integrated multimodal processing. This integration is consistently observed across subjects, though some ROIs (e.g., PMvh in Subject S2 with only 14 voxels) have insufficient data for reliable interpretation. The transition from linear to MLP encoders increases the total number of significantly predicted voxels while maintaining similar representation patterns, indicating that nonlinear encoding primarily enhances prediction accuracy rather than fundamentally altering feature representation structure.

### M.2  Variance partitioning of various models

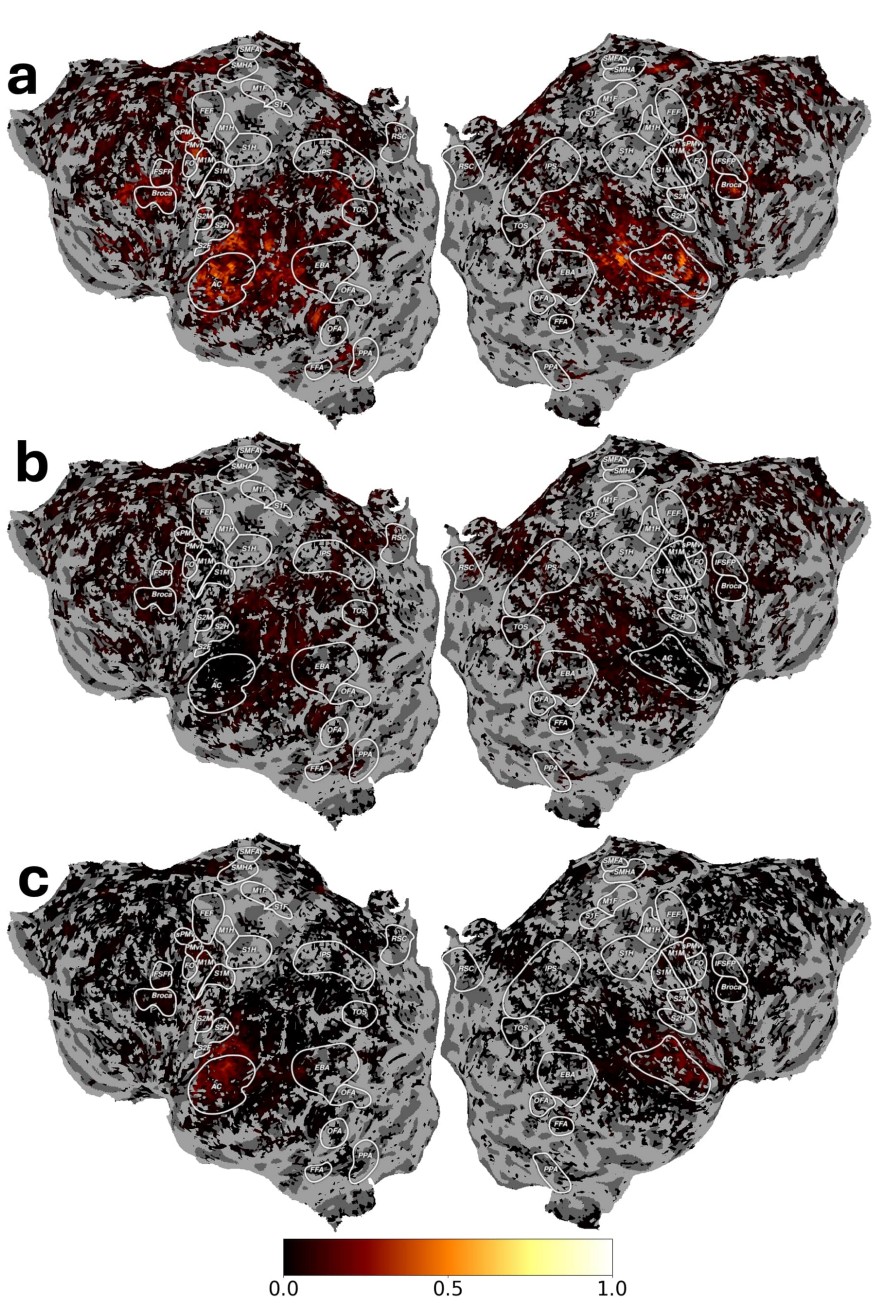

Figure 34: Voxelwise variance partitioning analysis showing the contributions of different feature types to prediction accuracy for a subject S1 using linear models. The flatmaps display (a) variance jointly explained by audio and semantic features, (b) variance uniquely explained by semantic features, and (c) variance uniquely explained by audio features. Values shown are normalized correlations ($CC_{norm}$) for voxels where the joint model achieved significant prediction ($q(\text{FDR}) < 0.01$).

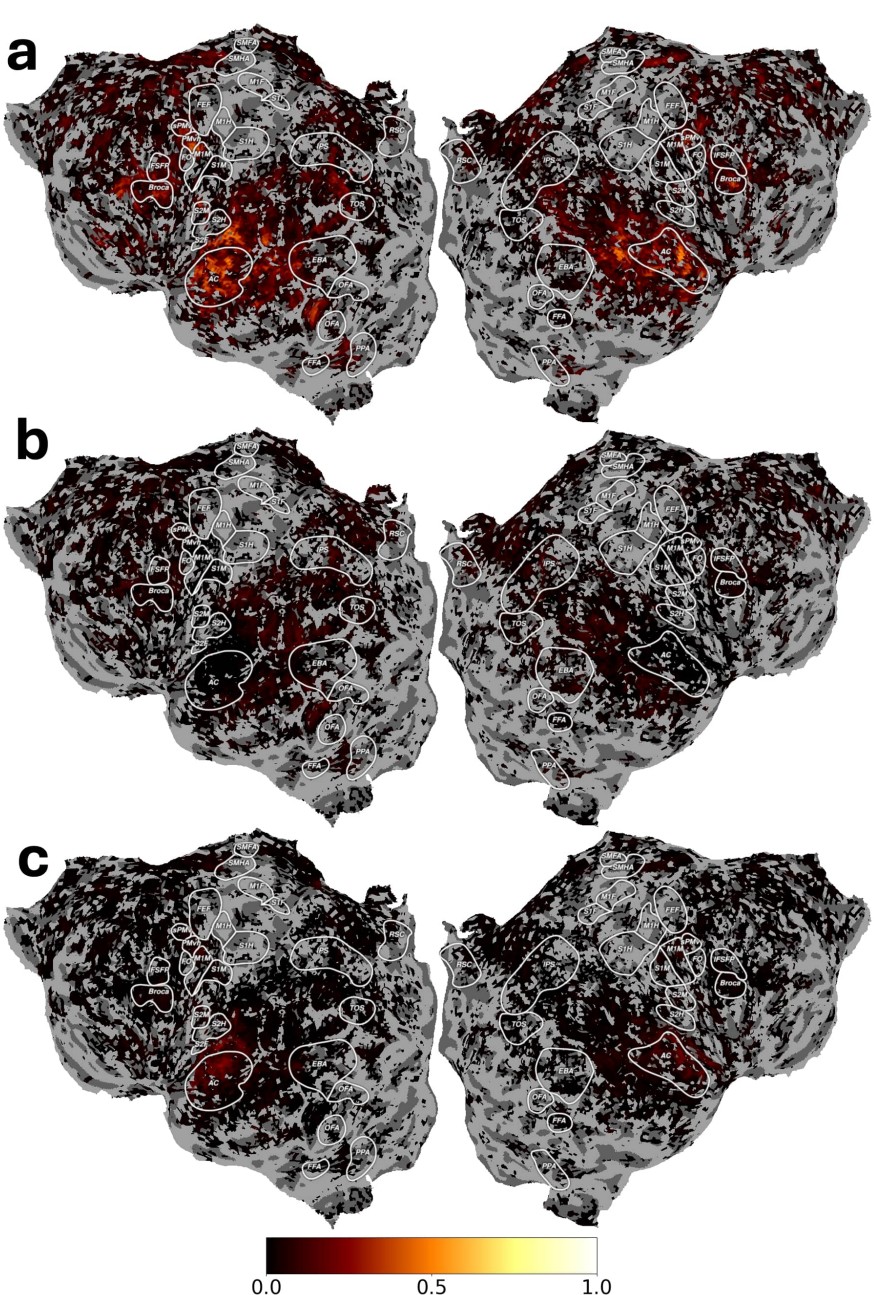

Figure 35: Voxelwise variance partitioning analysis showing the contributions of different feature types to prediction accuracy for a subject S1 using MLP models. The flatmaps display (a) variance jointly explained by audio and semantic features, (b) variance uniquely explained by semantic features, and (c) variance uniquely explained by audio features. Values shown are normalized correlations ($CC_{norm}$) for voxels where the joint model achieved significant prediction ($q(\text{FDR}) < 0.01$).

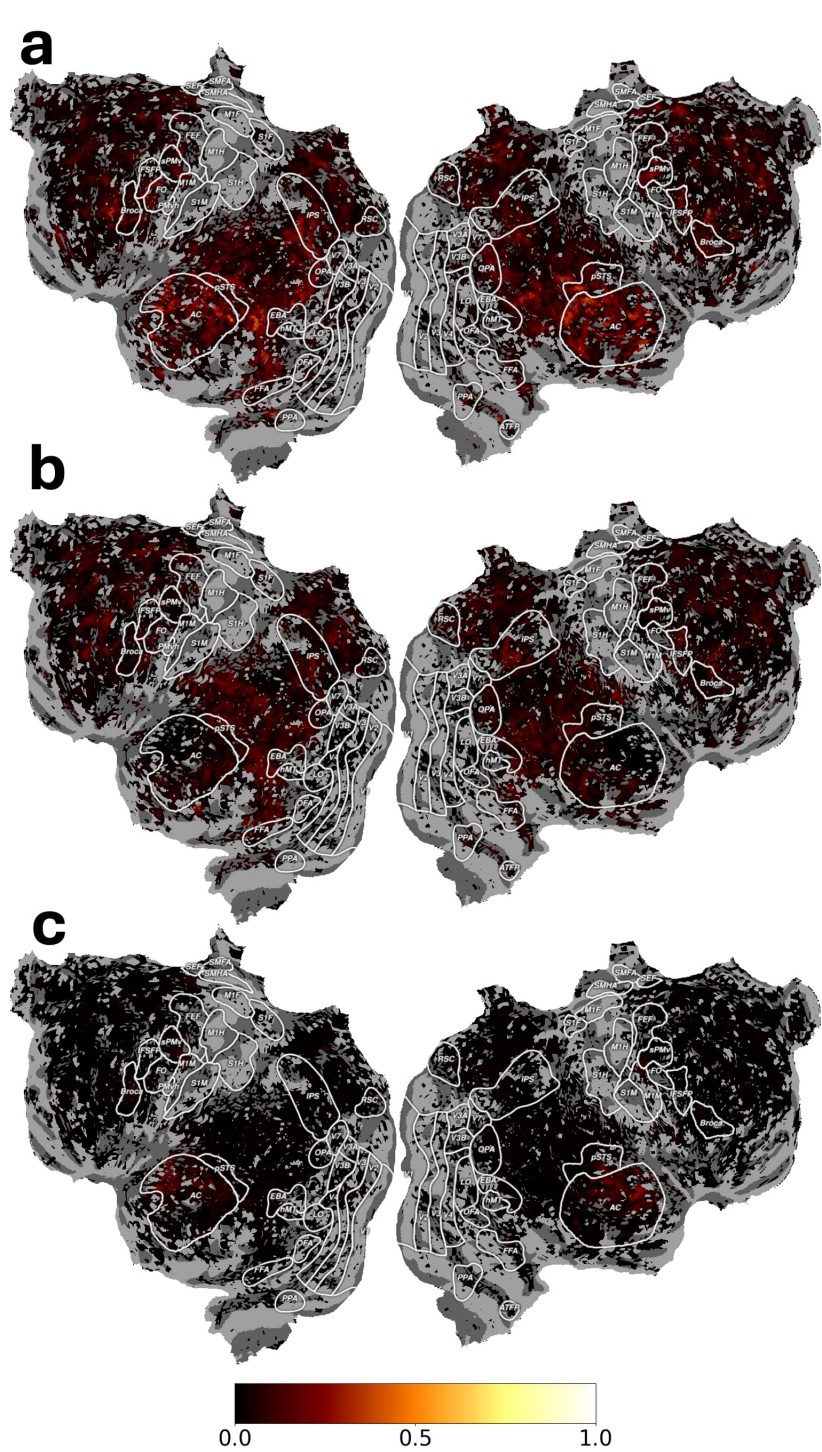

Figure 36: Voxelwise variance partitioning analysis showing the contributions of different feature types to prediction accuracy for a subject S2 using linear models. The flatmaps display (a) variance jointly explained by audio and semantic features, (b) variance uniquely explained by semantic features, and (c) variance uniquely explained by audio features. Values shown are normalized correlations ($CC_{norm}$) for voxels where the joint model achieved significant prediction ($q$(FDR) $< 0.01$).

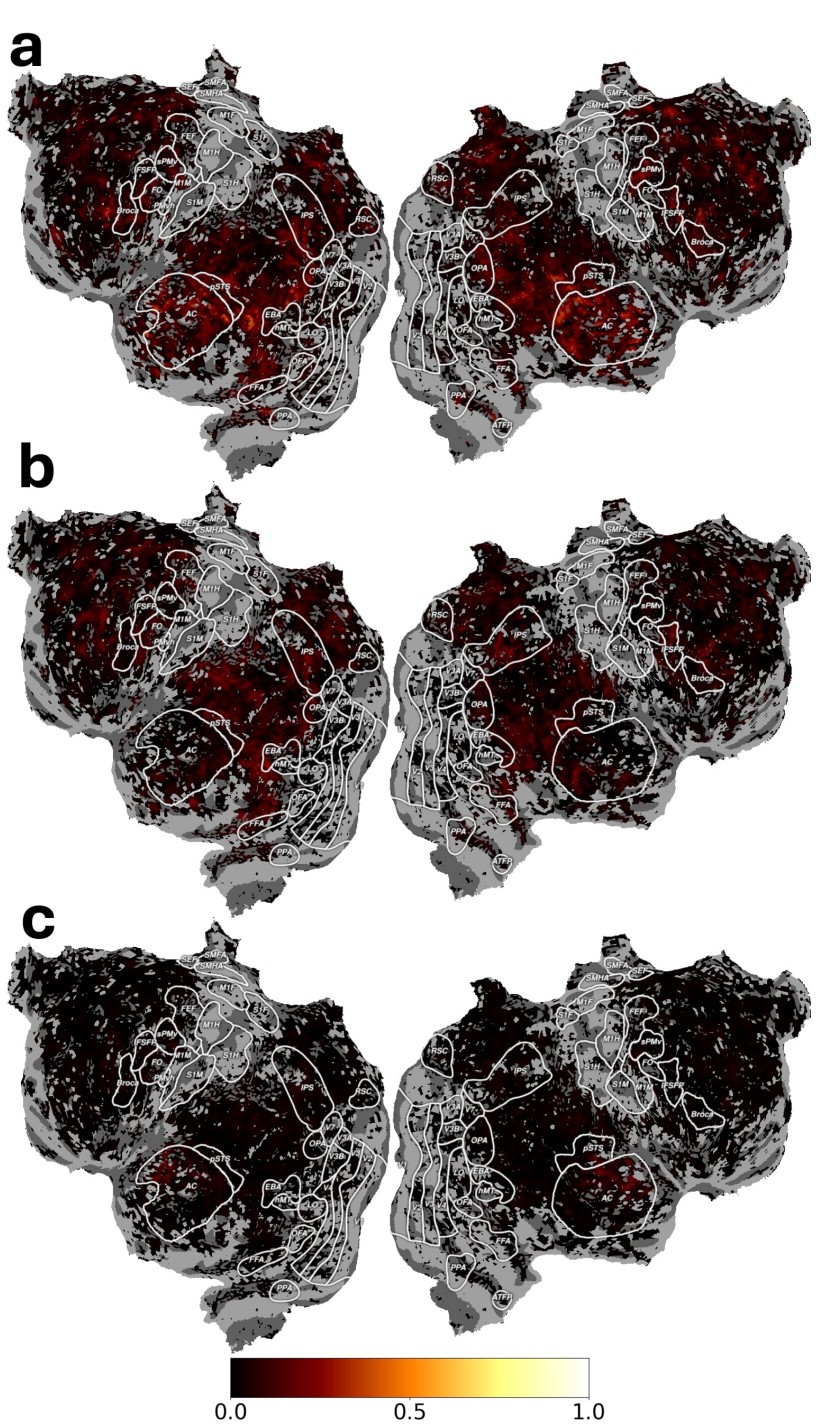

Figure 37: Voxelwise variance partitioning analysis showing the contributions of different feature types to prediction accuracy for a subject S2 using MLP models. The flatmaps display (a) variance jointly explained by audio and semantic features, (b) variance uniquely explained by semantic features, and (c) variance uniquely explained by audio features. Values shown are normalized correlations ($CC_{norm}$) for voxels where the joint model achieved significant prediction ($q$(FDR) $< 0.01$).

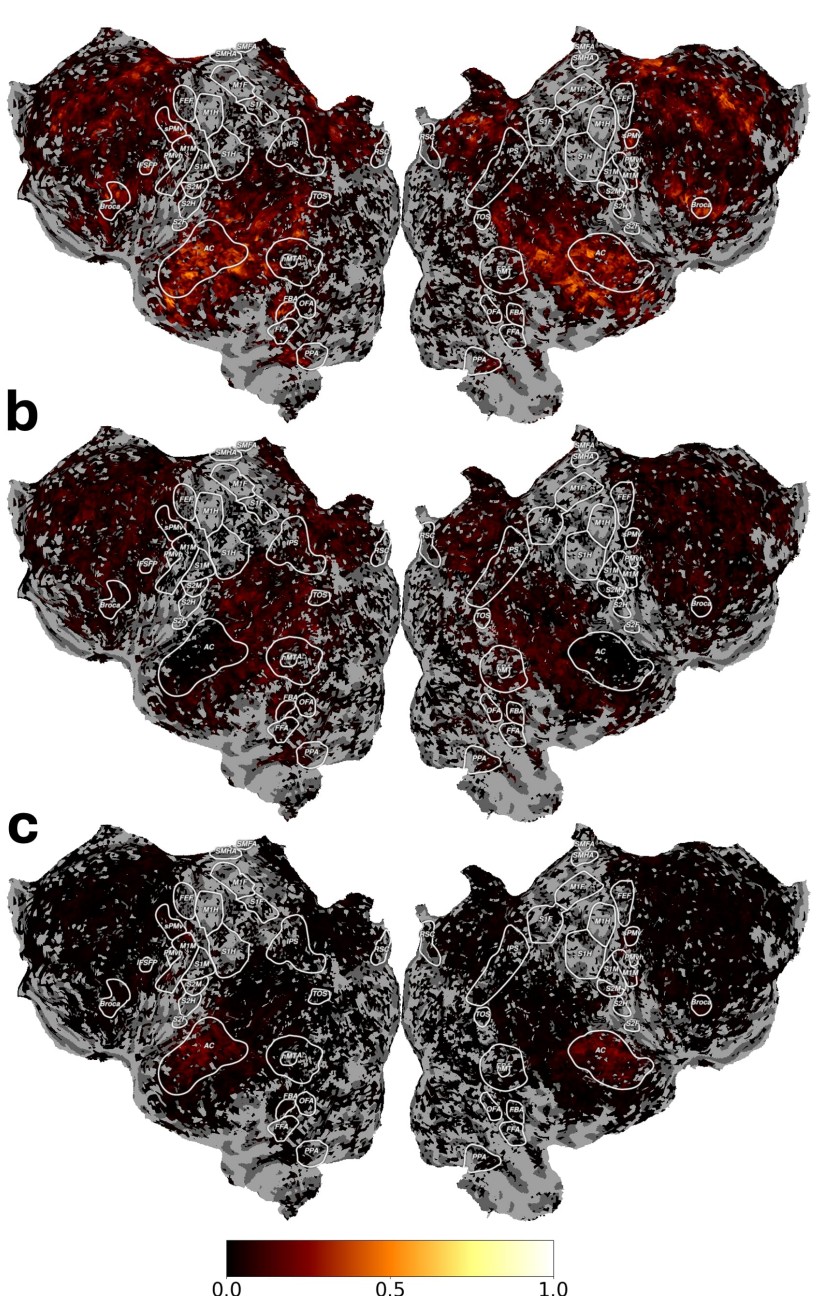

Figure 38: Voxelwise variance partitioning analysis showing the contributions of different feature types to prediction accuracy for a subject S3 using linear models. The flatmaps display (a) variance jointly explained by audio and semantic features, (b) variance uniquely explained by semantic features, and (c) variance uniquely explained by audio features. Values shown are normalized correlations ($CC_{norm}$) for voxels where the joint model achieved significant prediction ($q(\text{FDR}) < 0.01$).

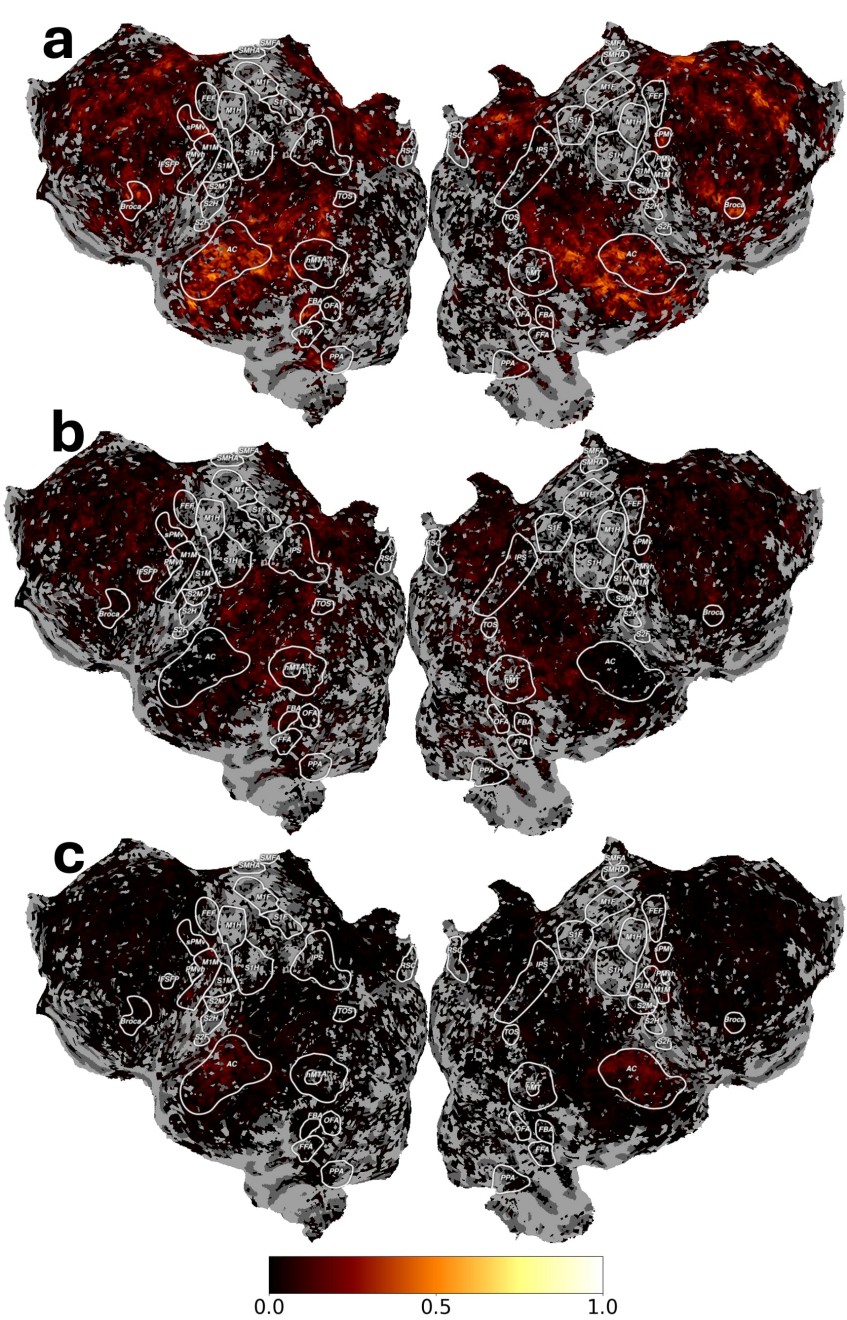

Figure 39: Voxelwise variance partitioning analysis showing the contributions of different feature types to prediction accuracy for a subject S3 using MLP models. The flatmaps display (a) variance jointly explained by audio and semantic features, (b) variance uniquely explained by semantic features, and (c) variance uniquely explained by audio features. Values shown are normalized correlations ($CC_{norm}$) for voxels where the joint model achieved significant prediction ($q$(FDR) $< 0.01$).

## M.3 LARGEST VARIANCE PARTITIONING FOR EACH VOXEL

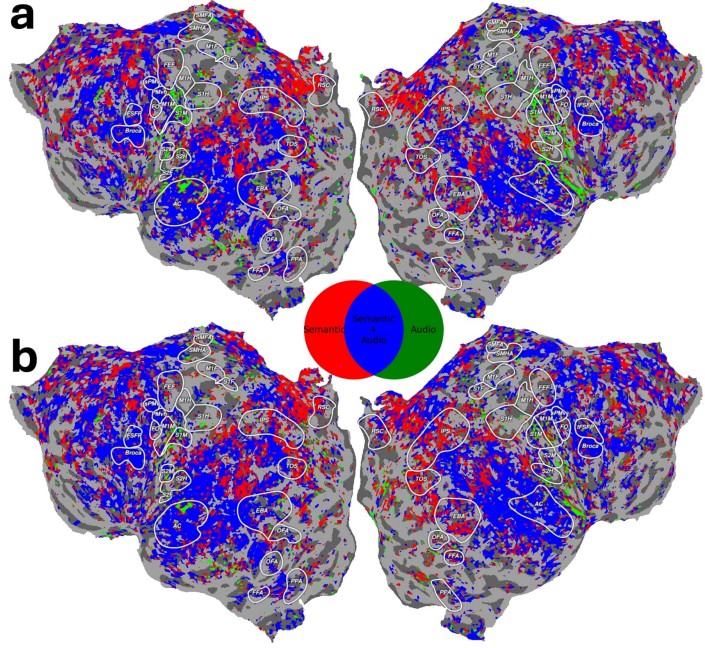

Figure 40: Voxelwise analysis showing the largest variance explained by each feature type for all significantly predicted voxels ($q(\text{FDR}) < 0.01$) for subject S1. The flatmaps display which feature partition (semantic in red, audio in green, or their combination in blue) best explains the variance in each cortical voxel using (a) linear and (b) MLP encoders, with outlined regions indicating key functional areas.

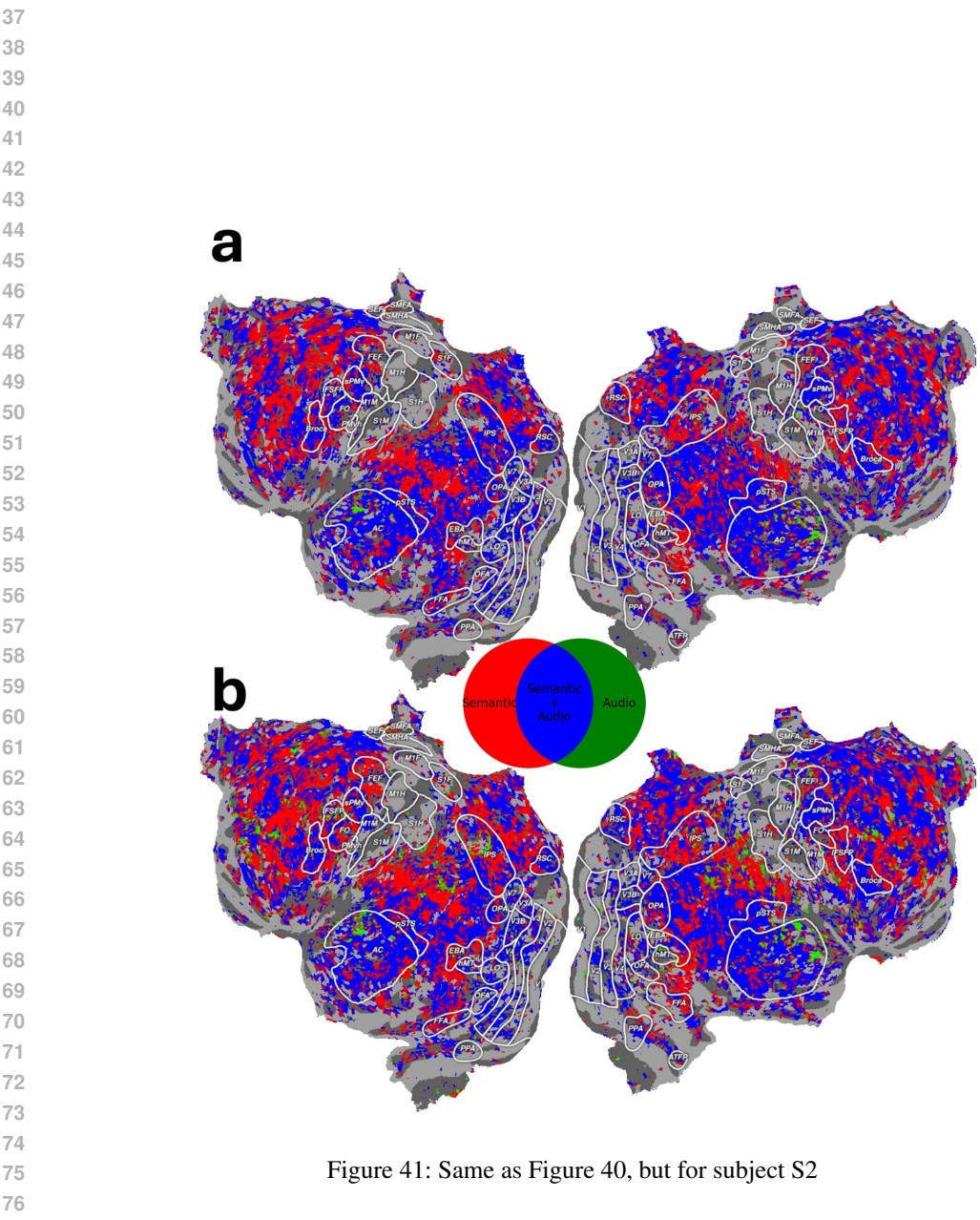

Figure 41: Same as Figure 40, but for subject S2

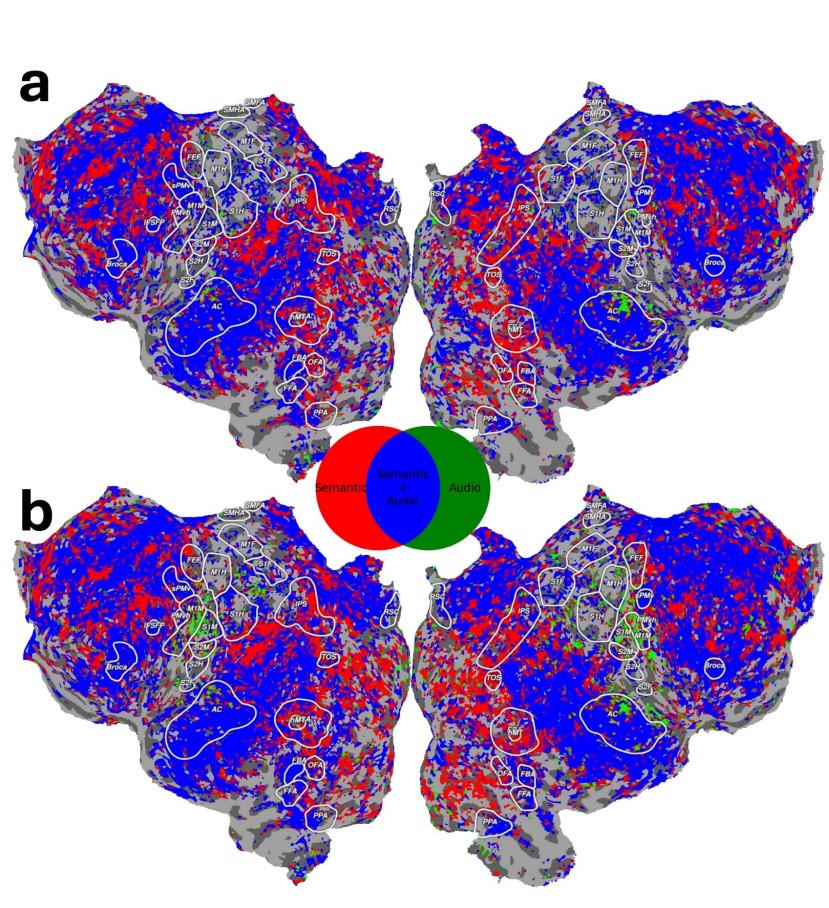

Figure 42: Same as Figure 40, but for subject S3

## M.4 VARIANCE PARTITIONING VENN DIAGRAM

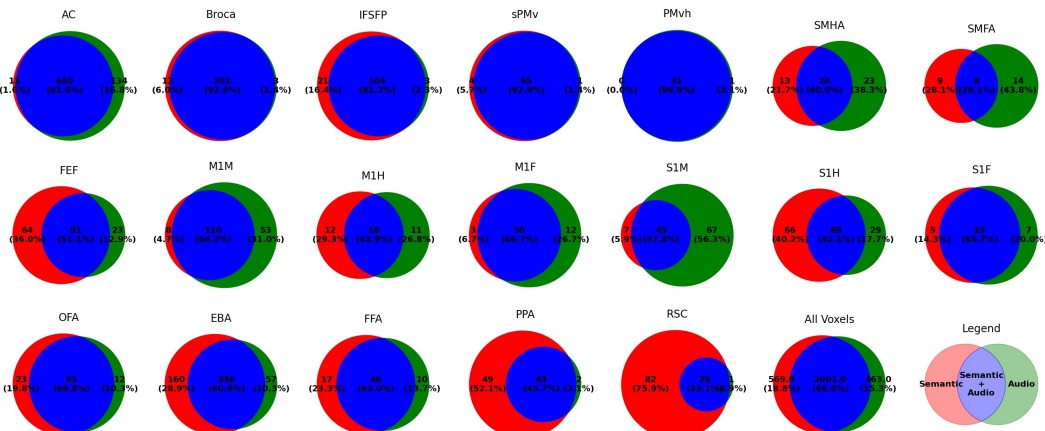

Figure 43: Venn diagrams showing the distribution of explained variance across different brain regions of interest (ROIs) for subject S1, using linear encoder. Each diagram displays the unique and shared variance explained by semantic features (red), audio features (green), and their overlap (blue). Values indicate the number of significantly predicted voxels and their percentages. Only the voxels that was predicted statistically significantly ($q(\text{FDR}) < 0.01$) was used in the analysis

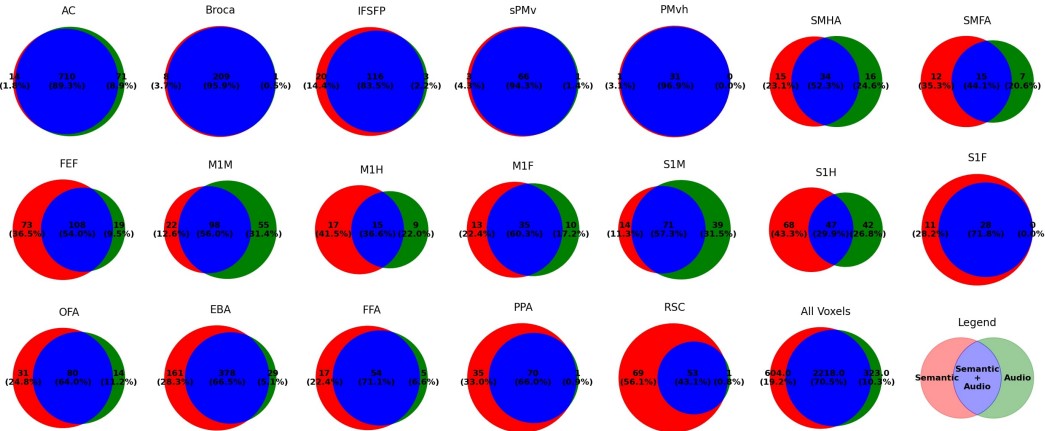

Figure 44: Venn diagrams showing the distribution of explained variance across different brain regions of interest (ROIs) for subject S1, using MLP encoder. Refer to Fig 43 for more detail.

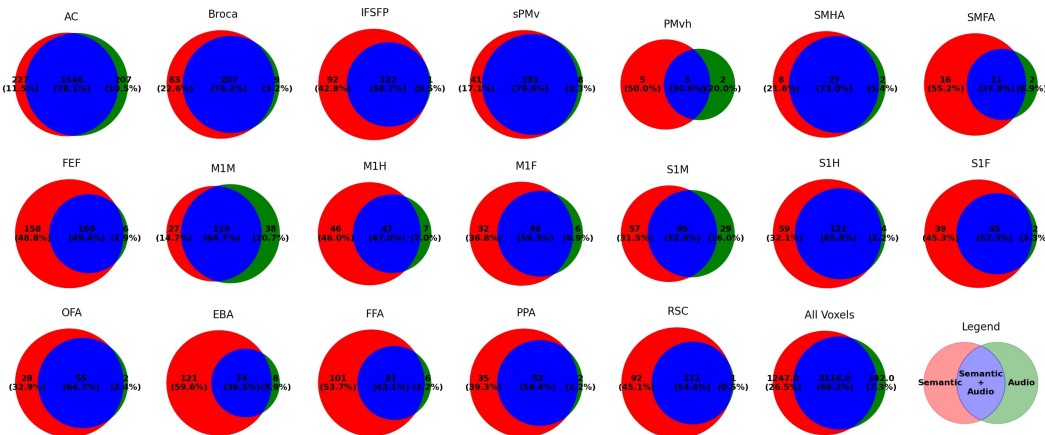

Figure 45: Venn diagrams showing the distribution of explained variance across different brain regions of interest (ROIs) for subject S2, using linear encoder. Refer to Fig 43 for more detail.

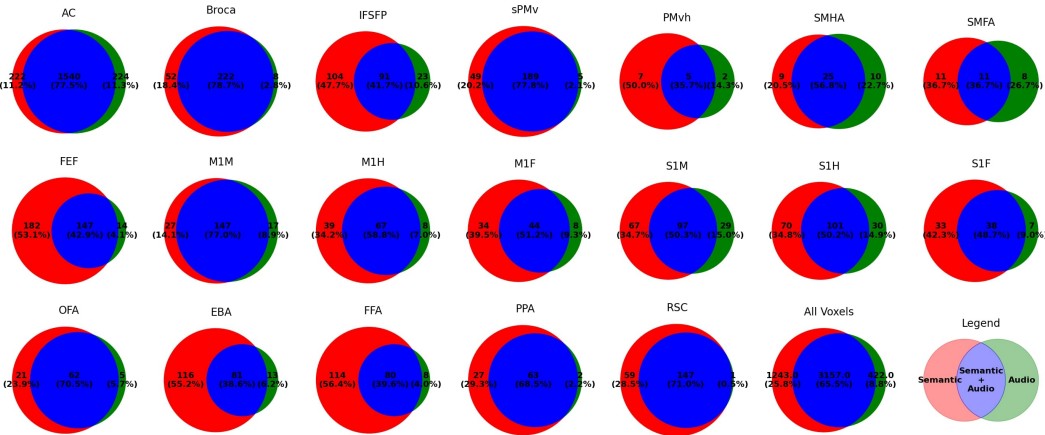

Figure 46: Venn diagrams showing the distribution of explained variance across different brain regions of interest (ROIs) for subject S2, using MLP encoder.Refer to Fig 43 for more detail.

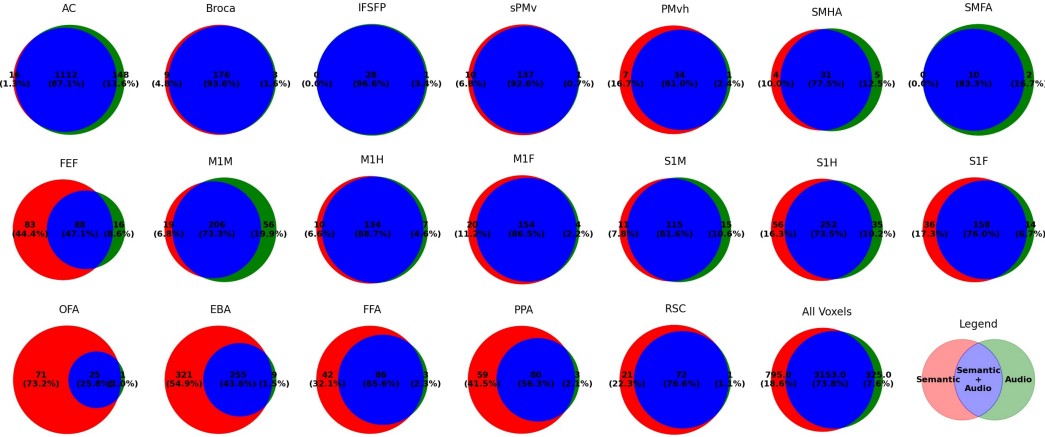

Figure 47: Venn diagrams showing the distribution of explained variance across different brain regions of interest (ROIs) for subject S3, using linear encoder. Refer to Fig 43 for more detail.

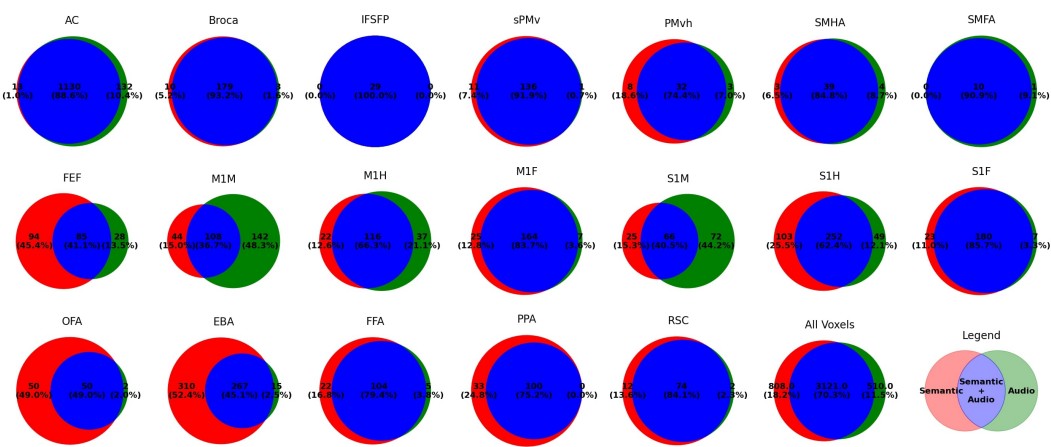

Figure 48: Venn diagrams showing the distribution of explained variance across different brain regions of interest (ROIs) for subject S3, using MLP encoder. Refer to Fig 43 for more detail.

# N  UNIQUE CHALLENGES IN SPEECH ENCODING AND CONTEXTUALIZING MODEL PERFORMANCE

Here we analyze the fundamental methodological disparities between vision and language encoding in neuroscientific research. We first examine the unique challenges of language encoding compared to vision encoding, highlighting why nonlinear models have been difficult to implement in language neuroscience, and how our work begins to address these longstanding methodological barriers. We then contextualize the magnitude of our performance improvements by conducting a comparative analysis with established benchmarks from recent literature.

## N.1  CHALLENGES OF SPEECH ENCODING COMPARED TO VISION ENCODING

Table 6: Comparison of vision and speech encoding datasets (from Allen et al. (2022) and LeBel et al. (2023)).

| Characteristic | Natural Scenes Dataset (Vision) Allen et al. (2022) | Lebel et al. Language Dataset LeBel et al. (2023) |
| --- | --- | --- |
| Stimulus Presentation | 4 seconds per image (3s image, 1s gap) | 2 seconds, ~5 words spoken |
| Voxel Prediction Space | ~15k voxels (occipital areas) | 80–90k voxels (whole cortex) |
| Data Collection | 30–40 hours per subject | 20 hours per subject |
| Prediction Complexity | Primarily perceptual | Cortex-wide, including higher-order semantic areas |
| Number of Subjects | 8 | 3 |
| Spatial/Temporal Resolution | 1.8mm × 1.8mm × 1.8mm, 1.6s | 2.6mm × 2.6mm × 2.6mm, 2s |
| Field Strength (Tesla) | 7T | 3T |

While vision encoding has long benefitted from nonlinear models, speech encoding presents unique challenges, as illustrated by the stark differences between the Natural Scenes Dataset (NSD) Allen et al. (2022) and the Lebel et al. Language Dataset LeBel et al. (2023), as outlined in Table 6.

The temporal dynamics of stimulus presentation fundamentally differ between these datasets. NSD presents visual stimuli for 4 seconds (3 seconds of image presentation with a 1-second gap), whereas the Lebel et al. Language Dataset captures linguistic stimuli over 2 seconds, with approximately 5 words spoken during that interval. This rapid and continuous linguistic information flow creates significant complexity in encoding neural representations.

The prediction space for these datasets also reveals substantial methodological challenges. NSD focuses on predicting neural activity in approximately 15,000 voxels primarily within occipital areas, which are predominantly perceptual. In contrast, the Lebel et al. Language Dataset requires predicting 80-90,000 voxels across the entire cortex, encompassing higher-order semantic areas. Predicting neural activity in non-perceptual, higher-level regions like the prefrontal cortex introduces considerable noise and computational complexity.

Data collection further highlights the intrinsic difficulties. While NSD collected 30-40 hours of data per subject with 8 participants and high-resolution 7T imaging, the Lebel et al. Language Dataset gathered 20 hours from only 3 subjects using lower-resolution 3T imaging. These constraints make developing sophisticated encoding models particularly challenging for language processing.

These fundamental differences underscore why nonlinear encoding models, which have become standard in vision research Yang et al. (2023); Scotti et al. (2024), have been difficult to implement in language neuroscience. Our work represents a critical step towards bridging this methodological gap.

## N.2  TYPICAL IMPROVEMENT MAGNITUDES IN FMRI SPEECH ENCODING STUDIES

To contextualize the improvements reported in our study, we present a comprehensive comparison of typical improvement magnitudes ($\Delta r$) observed in leading fMRI speech encoding research. This analysis demonstrates that nonlinearity reveals a wealth of information contained within the language and speech model embeddings.

Table 7: Comparison of typical improvement ranges ($\Delta r$) in language fMRI encoding studies

| Study | Analysis Type | Typical $\Delta r$ Range | Notes |
|---|---|---|---|
| *ROI-wise Analysis* | | | |
| Caucheteux et al. (2023) | ROI-wise | -0.005 to 0.015 | "Forecast score" in Fig. 2(f) |
| Lamarre et al. (2022) | ROI-wise | 0.025 to 0.050 | For AC, Broca, sPMv ROIs (no statistical testing) |
| Millet & King (2021) | ROI-wise | 0 to 0.015 | From Fig. 3(D) |
| **Our Study** | **ROI-wise** | **0.025 to 0.075** | **AC: 0.06, Broca: 0.025-0.050, IFSFP: 0.050-0.075** |
| *Voxel-wise Analysis* | | | |
| Aw & Toneva (2022) | Voxel-wise | -0.2 to 0.2 | From Fig. 4 |
| Jain & Huth (2018) | Voxel-wise | -0.2 to 0.2 | From Fig. 3 |
| Millet & King (2021) | Voxel-wise | -0.008 to 0.008 | Varied ranges (-0.06 to 0.06 also) reported |
| Caucheteux et al. (2023) | Voxel-wise | 0.004 to 0.020 | Relative gains of 0-5% |
| **Our Study** | **Voxel-wise** | **-0.5 to 0.5** | **17.2% average voxelwise $r^2$ improvement over baseline** |

This comparative analysis reveals two critical insights. First, analyzing and deriving conclusions from modest $\Delta r$ improvements is standard practice in the language fMRI encoding field. Second, our improvements are substantially larger than those typically reported in comparable studies. Notably, while influential works like Caucheteux et al. (2023) report ROI-wise $\Delta r$ values ranging from -0.005 to 0.015, our study demonstrates much larger improvements in key regions like the Auditory Cortex (0.06), Broca's area (0.025-0.050), and IFSFP (0.050-0.075).

For voxel-wise analyses, our improvements (Appendix K.1 and Figure 24) span a wider range (-0.5 to 0.5) than other studies, with a substantial 17.2% increase in average $r^2$ compared to semantic-only linear models. Subject-wise analyses in Figure 29 reveal even more pronounced effects in some ROIs, with $\Delta r$ values exceeding 0.100.

In all the studies referenced above, even modest ROI-wise and voxel-wise improvements played pivotal roles in deriving significant scientific conclusions. Given that our improvements are more pronounced by comparison, we believe our research provides robust empirical evidence for the benefits of nonlinear, multimodal approaches in language encoding models.

## O  LICENSES OF THE ASSETS

**LeBel et al. fMRI dataset:** We use the fMRI dataset from LeBel et al. LeBel et al. (2023). This dataset is licensed under the Creative Commons Zero (CC0) license. It can be accessed at `https://openneuro.org/datasets/ds003020/versions/3.0.0`.

**Llama models: Llama models:** We use Llama models spanning Llama-1 (7B, 13B, 33B, 65B) Touvron et al. (2023a), Llama-2 (7B) Touvron et al. (2023b), and Llama-3 (8B) Dubey et al. (2024). All models were accessed via Hugging Face at `https://huggingface.co/meta-llama` and were used under Meta Llama Community Licenses, which permit research use but restrict redistribution and commercial applications.

**Whisper models:** We use Whisper models Radford et al. (2023) from OpenAI, released under the MIT License. This license allows free use, modification, and distribution with minimal restrictions. The models were accessed via Hugging Face at `https://huggingface.co/docs/transformers/en/model_doc/whisper`.

