# OpenReview forum: "Aligning the Brain with Language Models Through a Nonlinear and Multimodal Approach"
_ICLR.cc/2026/Conference — Submitted to ICLR 2026_

### Official Review · Reviewer_JRKU · 2025-10-16

**Soundness:** 4
**Presentation:** 3
**Contribution:** 4
**Rating:** 10
**Confidence:** 4

**Summary:**

Instead of classical linearized unimodal speech models to predict voxel-wise cortical activity, the present paper propose a nonlinear multimodal prediction model, integrating audio and semantic features. The paper describes characteristics of the model and reports several elements of evaluation and comparison with other models. In short and in average, the gains are approximately of 15%. The paper also analyzes the results and shows that improvements are mainly due to the nonlinear aspects. Interestingly enough, the analyses also suggest strong relations with a series of neurolinguistic theories (including motor theory of speech perception, the convergence-divergence zone and embodied semantics).

**Strengths:**

Very interesting paper with an impressive amount of experimental work. The descriptions are very pedagogical and the paper does not remain at the experimental level only, but also develop more theorical interpretations.

**Weaknesses:**

A related aspect could be developed more, concerning the spatial and temporal complexity of the approach. It is important to have higher performances, but at which cost (in time and energy).

**Questions:**

Even if the model becomes very large, could it be possible to think even larger, by integrating other modalities, either directly related to linguistic aspects or even wider to other perceptive (eg visual) dimensions ? Or are we here at a limit ?

---

> ### Author Response · Authors · 2025-12-04
>
> We sincerely thank the reviewer for the positive assessment of our work and for recognizing both the experimental depth and the theoretical contributions. We also appreciate the constructive suggestions and address them briefly below.
>
> ### Regarding the performance and cost
>
> We agree that evaluating the computational implications of nonlinear approaches is an important direction for future work. Our current study is focused on identifying aspects of cortical computation that linear, unimodal mappings systematically fail to capture; accordingly, we do not position our contribution as one that optimizes or benchmarks computational efficiency. As clarified in our response to Reviewer ZbwB, the motivation for introducing nonlinear mappings is not to pursue maximal performance, but to more faithfully approximate the nonlinear interactions inherent in multimodal speech processing. We also note that, because our nonlinear approach operates on dimensionally reduced representations via PCA rather than learning voxel-wise parameters, its computational footprint does not scale with the number of predicted voxels, which partially mitigates concerns about increasing cost.
>
> ### On scaling to additional modalities
>
>  We appreciate the reviewer’s constructive question regarding the integration of further perceptual modalities. Our work is intentionally focused on speech encoding, where linguistic and acoustic inputs represent the core signals shaping cortical responses. This scope allows us to isolate the nonlinear interactions that are obscured by strictly linear unimodal models. While our fusion mechanism is general and could, in principle, incorporate additional streams—such as prosodic or visual articulatory cues present in natural communication—doing so requires aligned multimodal speech-fMRI datasets that are not yet widely available.
>
>
> Once again, we thank the reviewer for thoughtful feedback.

---

### Official Review · Reviewer_ZbwB · 2025-10-31

**Soundness:** 3
**Presentation:** 3
**Contribution:** 2
**Rating:** 4
**Confidence:** 5

**Summary:**

This work is part of a broader research effort to explore the alignment between representations from language models and human brain activity during speech comprehension. While prior brain encoding studies typically used linear mapping between text- or speech- stimulus features from unimodal models and voxel activations of speech-evoked brain activity, the current work performs nonlinear mapping between stimulus features to predict speech-evoked fMRI brain activity. They also combined text- and speech features, defined as multimodal features, to predict the brain activity. The authors argue that use of nonlinear mapping introduces a complex integration of auditory signals with linguistic features for better brain predictions. Using the nonlinear multimodal approach, the evaluation focuses on comparing brain predictivity with baseline models such as linear regression, multi-layer linear regression, and delayed interaction MLP methods for brain encoding. Furthermore, the authors test nonlinear multimodal models to examine which components are really driving brain activity. Overall, proposed nonlinear MLP shows strong encoding performance (>17% improvement) over linear mapping predictions, suggesting that nonlinear and multimodal approaches are alternative approaches to brain encoding.

**Contributions:**

* *Nonlinear, multimodal brain encoding:* The study uses a nonlinear MLP and multimodal features by concatenating MLP hidden-layer features of text- and speech-based language-model representations, and uses these multimodal features to predict fMRI brain response to speech-evoked activity. Specifically, the authors introduce an MLP on each modality’s features as well as another MLP mapping from the integrated multimodal representation to brain activity, an approach that is less explored relative to linear mappings between stimuli and brain activity.
* *Comprehensive evaluation:* The study evaluates nonlinear, multimodal early-fusion (MLP) and compares the brain predictivity against baseline models including linear regression, multilayer linear regression, Delayed Interaction MLP. For each encoding model, the authors report Pearson correlation and normalized correlation coefficient. Further, they use RED-based (Relative Error Based) clustering methods to examine the role of nonlinearity after dimensionality reduction. Also, the use of variance partition methods explains the contribution of each modality  in nonlinear cross-modal interactions across language, sensory and motor ROIs.

**Technical summary:**
This is primarily an empirical study, and its methodology involves the following components:
* *Linear and nonlinear encoding models:* The authors define several encoding models to map the alignment between language model representations and brain activity. Linear regression follows standard prior brain encoding studies. The nonlinear MLP uses a single hidden layer with a hidden dimension of 256 units. The MLLinear model is an MLP with no dropout, no batch normalization, and no nonlinear activation; they use identity as activation function, meaning everything is linear. The Delayed interactionMLP performs an MLP on each modality first; the reduced modality representations are concatenated and then linearly mapped to brain activity.
* *Normalized correlation coefficient (CCnorm):* To compute CCnorm, the authors estimate a ceiling (CCmax) for each voxel, and model performance divided byCCmax gives CCnorm.
* *Relative Error Based clustering:* RED quantifies the temporal advantage of one feature set over another. Specifically, the RED method identifies the time points at which semantic features from LLaMA predict better than auditory features from Whisper and vice versa. RED is computed by the signed difference of absolute model errors.

**Experimental design/evaluation:**
* *Encoding performance of linear and nonlinear encoders:* The authors evaluate the encoding performance of linear model, an MLP with single hidden layer, an MLLinear, and Delayed interaction MLP. To perform encoding, authors use a naturalistic Moth Radio Hour fMRI dataset in which 3 participants listened to 95 stories. To build encoding models, they extract text-modality representations from LLaMA and speech-modality representations from Whisper and report encoding performance in unimodal and multimodal settings (concatenated features used as multimodal for linear and MLLinear).
* *Role of nonlinearity vs. complex, rich representations:* This analysis evaluates whether nonlinearity is the factor that drives better encoding performance or whether complex rich representations drive better predictivity. The authors perform a linear mapping on PCA-reduced dimension and MLLinear (an MLP without nonlinear activation, using full dimension). If both settings result in similar or lower performance, this implies that nonlinearity is a key factor driving better encoding performance.
* *Role of nonlinearity vs. multimodal integration:* This analysis tests whether nonlinearity improves cross-modal integration in predicting brain activity over within-modality nonlinearity with a linear mapping to brain activity. If nonlinear multimodal integration improves over other settings, it implies that the nonlinear approach provides better cross-modal interaction than linear models.
* *Contribution of each modality in nonlinear, multimodal encoders:* The authors use variance partitioning approach to examine the explainable variance of semantic features (LLaMA) in multimodal integration across language, sensory and motor ROIs. The same analysis is performed to examine the explainable variance of speech features in multimodal integration.


**Main findings:**
According to the authors’ interpretation, the main findings are as follows:
* Across four encoder model settings, nonlinear multimodal encoding achieves higher brain predictivity over linear models.
* The superior encoding performance of nonlinear approaches over linear approaches is examined by disentangling nonlinearity from reduced feature dimensionality, and finding that nonlinearity fundamentally drives superior encoding performance.
* The ROI analysis using modality-specific contributions reveals that adding audio features enhances predictions in auditory, primary motor, somatomotor, and precuneus regions, while adding semantic features enhances predictions across all cortical ROIs except the auditory region, indicating that semantic features show a broad influence on neural activity.
* The variance-partitioning analysis reveals a contribution of auditory information in higher visual areas, suggesting that auditory signals may provide information beyond visual or linguistic features alone.

**Strengths:**

I found this work to have the following strengths:
* *Clarity:* The manuscript introduction, dataset, and key methodological details are well written and well structured. The encoding model details are easy to follow and clearly present differences in model training for nonlinear and linear approaches. Later, the noise ceiling estimate, and RED metrics are described clearly. The results section clearly reports encoding performance across unimodal and multimodal settings, for all voxels and with PCA on brain data, using four encoding models; further, authors applied the control of modality in multimodal and explain the variance across brain ROIs, and then presents variance partitioning analysis to describe the shared and unique variance of unimodal and joint features of the MLP.
* *Originality:*  The idea of using nonlinear mappings to build encoding models and controlling model parameters similar to the MLLinear setting, is simple yet methodologically novel. Further, the authors perform disentangling of nonlinearity vs. feature dimensionality using linear approaches to predict brain responses and learn subject-specific models. This paper offers a comprehensive analysis of using nonlinear approaches in brain encoding, compares them with linear regression, and provides implications for using nonlinear approaches over linear models.
* *Significance:* This work is significant in that it contributes to a better understanding of the contribution of nonlinear approaches for better encoding performance during speech comprehension. The comprehensive analysis on the Moth Radio Hour dataset shows that nonlinear approaches show higher degree of brain predictivity over linear brain encoding. Beyond prediction accuracy, the work contributes a reproducible evaluation and analysis framework that clarifies when nonlinear multimodal encoders help in naturalistic speech brain encoding, especially with lower parameter settings.

**Weaknesses:**

From my perspective, the primary weaknesses of this study arise from the lack of statistical significance testing, limited  and limited evaluation:
* *Motivation for nonlinear approaches:* The authors argue that “nonlinear approaches are common in computer vision” and they aim to explore this in speech brain encoding (Yang et al., 2023; Chen et al., 2023; Scotti et al., 2024)”. However, Yang et al. (2023) use a linear voxel-wise readout on nonlinear deep features; this is standard linear mapping in brain encoding studies, where complex rich representations from non-linear language models are mapped to brain activity with a linear model.  Also, Chen et al., 2023; Scotti et al., 2024 are brain decoding studies that reconstruct visual stimuli from fMRI brain activity; projecting brain voxels into a common embedding space is common in brain decoding, and the objective is different. Given this, authors' motivation to use nonlinear approaches for speech brain encoding is not well  justified by these prior vision studies. I recommend that the authors clarify the motivation with more appropriate prior literature.
* *Interpretability vs. prediction accuracy:* The authors show that nonlinear, multimodal encoding exhibits a higher degree of brain predictivity over linear regression, with 17.2% (unnormalized) and 17.9% (normalized). However, the major question arises with interpretability. The main objective of aligning language models with brain language processing is to understand information processing in the two systems. Since the representations from language models are rich and complex while brain recordings are noisy with low SNR, researchers use simple linear models to map the systems. The use of nonlinear approaches resulting in increased prediction accuracy is not surprising. However, nonlinearity adds complexity and obscures the interpretability objective. Although the authors provide some justification that nonlinearity is the factor driving better performance over feature dimensionality, interpretability requires disentangling features into semantics, syntax, discourse, etc., and examining which regions process this information. Adding nonlinearity to the representations makes things more complex and makes it hard to understand which linguistic properties are really driving the activity. I suggest the authors discuss the motivation of prioritizing prediction accuracy using nonlinear approaches versus interpretability using linear approaches, and clarify the text to make the paper stronger.
* *No statistical significance reports:* The paper reports no statistical significance tests (e.g., block permutation and each model predictions and FDR correction across model groups) to determine significant voxels. Without any significance tests, the performance of MLP over other models should be considered as descriptive rather strong empirical evidence.
* *No standard error or standard deviation across subjects:* The results reported in Table 1 state that average voxelwise r^2 and normalized correlation coefficient (CCnorm) are reported for all models, without accompanying measures of variability such as standard deviation or standard error. These metrics are critical for evaluating the robustness of the findings.
* *Nonlinearity vs. dimensionality reduction:* Section 3.1 concludes that “nonlinearity that fundamentally drives superior encoding performance”, because MLP outperforms linear (PCA) and MLLinear(no activation function). Since MLP(PCA) and MLLinear have parameter match, the performance improvement is marginal. Without any statistical significance tests on the differences between models, the claim is not conclusive.
* *Lack of modality control and interpretation:* Authors use RED metric (signed difference of absolute errors) to compare multimodal vs. unimodal predictors and to examine the “impact” of semantic (text) vs. audio features. However, RED is a comparative error metric, not a control procedure: it does not remove shared/collinear information between modalities, nor does it quantify unique vs. shared variance. As a result, subtracting one model’s error from another’s does not establish which modality is causally responsible for a prediction improvement.


For a complete and detailed account of both major and minor issues, please refer to the “Questions” section.

**Questions:**

I would like to thank the authors for the interesting comparison between nonlinear and linear approaches for fMRI brain encoding during speech comprehension. However, there are several points that I believe require further attention/work. I have divided these into major issues, which should be prioritized, and minor ones, which should be addressed for a strong version of current work.

**Major Comments/Questions:**
* *Motivation and positioning:* As discussed in weakness1, I request authors to clarify why nonlinearity should help in speech brain encoding? Also, please correct Line 56: nonlinear approaches are more popular in brain decoding; this should not be presented as direct motivation for encoding.
* *Interpretability vs. prediction accuracy:* I recommend authors to provide a discussion on the trade-off between interpretability (linear mapping) and prediction accuracy (nonlinear approaches) for the paper's main objective.
* *Statistical significance and clarity on Table 1:* I suggest authors following
Please perform blocked permutation tests with Benjamini–Hochberg FDR correction (across voxels), report the adjusted p-values and effect sizes on model groups, and clearly indicate which model pairs are statistically significant.
Select voxels which are significant for each model and report prediction performance between models.
* *Reporting variability:*  Please report standard deviations (or standard errors) alongside mean values to provide a clearer picture of variability in Table 1.
* *Lack of modality control:* Prior studies use methods like residual approach to control one feature over other by regressing linearly correlated information [Toneva et al. 2022, Oota et al. 2024]. The other approach is to fit Text-only, Audio-only, and Text+Audio models with the same approach; report joint − best unimodal with paired bootstrap testing across voxels/subjects, and correct voxelwise tests with FDR [Reddy et al. 2021, Oota et al. 2023]. I recommend authors perform any of these methods to perform modality controlled contribution. To further validate this, authors can test any simple downstream task evaluation to see the impact of these controlled modality features.

Toneva et al. 2022, Combining computational controls with natural text reveals new aspects of meaning composition, Nature Computational Science, 2022

Oota et al. 2023, How does brain process syntax structure during listening, ACL Findings 2023

Reddy et al. 2021, Can fMRI the representation of syntactic structure in the brain, NeurIPS 2021


* *Clarity on multimodal integration vs. fusion vs. multimodal model:* The authors state that (i) MLP on text, (ii) MLP on audio, and (iii) concatenated these features considered as cross-model integration. Concatenation performed before is multimodal fusion. The paper should clearly define these terms and avoid confusion with multimodal models . In addition, please position multimodal setup against recent multimodal text+speech models (e.g., CLAP, Pengi, AudioLM) that learn paired text+audio embeddings via explicit joint objectives or cross-attention; explain how the concatenation+MLP in paper differs and why it is preferable for this study.
* *Clarity on section 3.3.1:* Section 3.3.1 reports that adding audio features enhance activity in other brain regions as well such as primary motor, somatosensory, occipital, and precuneus, while adding semantic information excel in the whole cortex except auditory cortex. Do these claims align with any prior literature? Oota et al. (2024) report that predictions of speech models (e.g., Whisper) in language regions are due to low-level features. Similarly, text-based language models significantly predict the auditory cortex, and this is due to low-level features. Should the authors clarify whether adding nonlinearity to these model representations yields higher brain-relevant semantics? Is the language component’s contribution in the multimodal features due to the MLP itself?
* *Clarity on Linear (all voxels) vs. MLP (PCA):* I recommend that the authors perform PCA on the weight matrix learned during the linear approach with all voxels and compute the similarity with the MLP model weights in the same PCA dimension. This analysis would provide how similar the encoding models are between the two approaches. The authors can extend this comparison across all four models to see which voxels cluster together in the projection.

**Minor Comments/Typos:**
While addressing the following points may not be critical to the paper’s core contributions, doing so would enhance the overall quality.
* Please clarify whether the semantic and auditory “controls” shown in Fig. 2 are derived from the RED method. In the text, explicitly describe how each control is constructed and applied.
* What statistical test was used to report significant differences across ROIs in Fig. 2e? How were multiple comparisons handled (e.g., FDR)? Please report the test details and effect sizes.
* Line 476: (Yang et al., 2023) -> Yang et al., (2023)
* Line 75: Appendix section number is missing.
* Line 108: Authors extracted semantic features representations from three LLaMA models. What model results are reported in Table 1?
* Did authors multikernel banded ridge approach? Banded ridge approach works on multiple embedding sources yielding contribution of each feature space in predicting brain activity.

**General Advice:**
The manuscript presents a nonlinear multimodal approach that integrates text and speech features to perform speech brain encoding during speech comprehension. Using a nonlinear approach, the authors evaluate subject-specific models, and compare with linear approaches under different settings. However, the current version lacks a clear motivation for nonlinear approaches, lacks modality control, and reports results without any statistical significance testing. Tables should include subject-level variability and statistical significance tests with FDR correction across models. Addressing these points and the above mentioned weaknesses and major comments would make the work stronger.

---

> ### Author Response · Authors · 2025-12-04
>
> Thank you for taking the time and providing thoughtful and valuable feedback. We have added new content (blue in the revised PDF). Below, we address each of your comments and questions in detail.
>
>
> ### W1. Motivation for nonlinear approaches
> > “The authors argue that ‘nonlinear approaches are common in computer vision’ and they aim to explore this in speech brain encoding (Yang et al., 2023; Chen et al., 2023; Scotti et al., 2024)… Given this, authors' motivation to use nonlinear approaches for speech brain encoding is not well justified by these prior vision studies. I recommend that the authors clarify the motivation with more appropriate prior literature.”
>
> Thank you for pointing this out. We agree that our original framing over-emphasized vision work and did not clearly articulate why nonlinear mappings themselves are important for speech encoding.
>
> Our core motivation is not that “vision has nonlinear models, so language should too”, but that cortical and language computations are intrinsically nonlinear, with rich interactions across features, time, and modalities. Nonlinear mappings are therefore often more appropriate for neuroscience questions such as in-silico experimentation, testing feature relevance, and assessing the joint contribution of feature sets, as argued by prior work on nonlinear brain models and encoding beyond linear regression.
>
> To clarify this, we have substantially revised the “nonlinearity” part of the introduction:
>
> - We now motivate nonlinearity primarily from **biological and computational evidence** (e.g., nonlinear neural dynamics, and nonlinear speech perception and multimodal integration), and cite corresponding work.
>
> - We explicitly reference prior arguments that **relaxing the linearity assumption** in encoding models can reveal functional organization patterns and enable more robust in-silico perturbation analyses.
>
> - We reposition the **vision literature**: instead of using it as a direct justification for our architecture, we now use it to highlight a *methodological gap*. In vision, nonlinear encoders are common partly because there are fewer voxels and simpler temporal structure; in speech fMRI, models must predict ~80k–90k voxels and capture rapid continuous dynamics, which has discouraged nonlinear encoding despite similarly rich nonlinear interactions.
>
> These changes are reflected in the revised line 54 – line 74 of the introduction.
>
> We believe this clarified framing directly addresses the concern about the motivation for nonlinearity and the role of the vision references.

---

> ### Author Response · Authors · 2025-12-04
>
> ### W2. Interpretability vs. prediction accuracy
> > “However, the major question arises with interpretability.”
> >
> > “ interpretability requires disentangling features into semantics, syntax, discourse, etc., and examining which regions process this information”
> >
> > “I suggest the authors discuss the motivation of prioritizing prediction accuracy using nonlinear approaches versus interpretability using linear approaches, and clarify the text to make the paper stronger.”
>
> As clarified in our response to **W1**, our primary motivation for using nonlinear mappings is not to maximize prediction accuracy per se, but to better approximate the **nonlinear nature of cortical language and multimodal processing**. Here, we focus on how this choice relates to interpretability.
>
> We agree that linear models have clear advantages for decomposing representations into semantics, syntax, discourse, etc., and mapping these components to cortical regions. Our goal in this paper is complementary: we aim to show that current *linear, unimodal* practice leaves a substantial amount of structured variance unexplained, particularly variance arising from nonlinear multimodal integration. In other words, while linear models remain the right tool for dissecting individual representational dimensions, our results suggest that important aspects of how the brain combines modalities and features are missed if we restrict ourselves to linear mappings. In the revised text, we explicitly state that our nonlinear model is intended to complement, rather than replace, such linear analyses (lines 499–506).
>
>
> To make this clearer, we have:
> - **Revised the first contribution bullet (lines 79-86) to shift the emphasis away from “beating baselines” and toward what the performance gains imply**. The contribution now highlights that a simple nonlinear **multimodal** encoder reveals robust, spatially organized gains over standard linear semantic baselines and prior linear ensembles, indicating that current linear, unimodal practices leave substantial **structured nonlinear multimodal variance** unexplained.
>
> - **Added a short discussion of the trade-off you raise in the Discussion section (lines 499–506)**: when the true brain–model mapping is approximately linear and the main goal is fine-grained attribution of feature weights, linear models remain preferable. However, if the underlying computation involves strong nonlinear interactions, strictly enforcing linearity can misattribute variance or underestimate the role of interactions. In this regime, a simple nonlinear mapping—such as the one we use—can provide a more faithful and thus more informative approximation of the underlying computation, while linear analyses remain essential for fine-grained feature-level interpretation.

---

> ### Author Response · Authors · 2025-12-04
>
> ### W3/W4. No statistical significance reports / No standard error or standard deviation across subjects
> > “no statistical significance tests (e.g., block permutation and each model predictions and FDR correction across model groups) to determine significant voxels…”
> > “… Table 1 … without accompanying measures of variability such as standard deviation or standard error…”
>
> Thank you for raising this point. We agree that the original submission did not clearly describe the statistical testing and variability measures we already use, and it also lacked explicit tests of *relative* improvements between models.
>
> First, we clarify that **significance tests were already applied to our voxel- and ROI-wise analyses**:
> - For the voxel-wise variance-partitioning plots, we only display voxels that pass an FDR-corrected significance threshold (Figure 3 and Appendix Figures 34–42; \(q<0.01\)).
> - For the ROI-wise plots (Figure 1e), we perform FDR-corrected significance tests for each ROI (\(q<0.05\)).
>
> Second, in response to the reviewer’s comment, we have added a dedicated **Statistical Testing** subsection in Appendix C, together with new per-subject tables and pairwise comparison figures that make between-model differences explicit:
> - Using voxelwise prediction scores (r² and CC_norm), we compute mean ± standard error of the mean (SEM) **across voxels** for each model and subject; these values are reported in the new Appendix Tables 2,3 (per-subject r² and CC_norm). Voxels are treated as repeated measurements of model performance within a subject, which is also the basis for the paired tests below.
> - Using the same voxelwise scores, we perform paired t-tests across voxels for every pair of models (A,B) and each subject, testing the null hypothesis that the mean difference A–B is zero. We also perform a pooled analysis in which we aggregate voxelwise score differences across all subjects and repeat the same paired testing procedure. The resulting t-statistics, Bonferroni-corrected over all model pairs, are visualized as heatmaps in the new figures (Figures 4,5) (pairwise model comparison for r² and CC_norm); significant cells are colored (red: A > B, blue: A < B), and non-significant comparisons are masked in white, with panels (a–c) showing individual subjects and panel (d) showing the pooled results.
> - Overall performance levels differ substantially across subjects and we have only three subjects, so an “across-subject SEM” would be dominated by between-subject offsets and is not very informative. Therefore, in the main-text table we report in the “Across subjects” column only the arithmetic mean of the subject-wise means (plus percent change relative to the text-linear baseline), and we base our formal inference on the within-subject voxelwise statistics described above. This rationale is now explained in Appendix C (lines930).
>
> Taken together, these clarifications and new analyses show that (i) our voxel/ROI results are based on formal significance tests, (ii) the relative gains of the multimodal nonlinear models over the baselines are statistically reliable, and (iii) the reported variability measures accurately reflect the uncertainty that is most relevant for our conclusions.

---

> ### Author Response · Authors · 2025-12-04
>
> ### W5. Nonlinearity vs. dimensionality reduction
>
> > “Section 3.1 concludes that ‘nonlinearity … fundamentally drives superior encoding performance’, because MLP outperforms linear (PCA) and MLLinear (no activation function). Since MLP(PCA) and MLLinear have parameter match, the performance improvement is marginal. Without any statistical significance tests on the differences between models, the claim is not conclusive.”
>
> Thank you for pointing out the lack of explicit significance testing between models.
>
> We would first like to clarify that the gains from adding nonlinearity alone (MLLinear → MLP, at fixed PCA dimensionality) are not “marginal” in the context of fMRI speech encoding. As discussed in Appendix N.2 “Typical Improvement Magnitudes in fMRI Speech Encoding Studies”, prior work often bases conclusions on much smaller ROI-wise and voxel-wise Δr values. In contrast, our nonlinear models (MLP) show clear improvements over their linear counterparts (MLLinear), for example in the multimodal condition the average voxelwise \(r^2\) increases from 4.10\% to 4.29\% (relative performance gain of 4.6\%), with similar gains in the text-only and audio-only settings. These effect sizes are comparable to or larger than the improvements typically interpreted as meaningful in the speech-encoding literature.
>
> As requested, we now support this claim with formal tests in the new **Statistical Testing** section of Appendix C (see W3/W4). Using voxelwise scores, we perform paired t-tests between **MLP (PCA)** and **MLLinear (PCA)** for each subject and in the pooled analysis, and find that the nonlinear models yield **statistically significant** improvements in all subjects. Combined with the observations that (i) MLLinear and Linear models achieve very similar performance, and (ii) MLPs without PCA overfit and perform worse, this justifies our conclusion that “while dimensionality reduction enables tractable modeling, it is nonlinearity that fundamentally drives superior encoding performance.” In the revised text we now explicitly reference these tests (line 257).
>
>
>
> ### W6. Lack of modality control and interpretation (RED metric)
>
> > “... RED is a comparative error metric, not a control procedure: it does not remove shared/collinear information between modalities, nor does it quantify unique vs. shared variance.”, “... subtracting one model’s error from another’s does not establish which modality is causally responsible for a prediction improvement.”
>
>
> Thank you for this thoughtful comment. We fully agree that the RED metric, by construction, is a **comparative** measure of prediction error and does not by itself provide a causal decomposition or a true unique/shared feature  analysis between modalities.
>
> Our intent with RED was more modest and descriptive:
>
> - RED was designed to summarize **where and by how much** text or audio model reduces error relative to the best unimodal alternative, and to visualize **spatial patterns** of voxels that benefit more from text-like vs. audio-like information *given the model as trained*, rather than to claim that one modality is causally responsible. In particular, we used RED as an example of how nonlinear encoders, when paired with appropriate interpretability tools, can reveal **quantitatively different spatial patterns** compared to standard linear encoders.
>
> - For a more formal treatment of unique vs. shared contributions, we rely on our variance-partitioning analyses (Figure 3), which explicitly quantify unique and shared variance attributable to text and audio predictors.
>
> A fully causal analysis of modality effects (e.g., via explicit interventions or causal designs) is beyond the scope of this work and would be an interesting direction for future research.

---

### Official Review · Reviewer_Fs8L · 2025-11-09

**Soundness:** 2
**Presentation:** 3
**Contribution:** 2
**Rating:** 6
**Confidence:** 2

**Summary:**

The manuscript proposes a nonlinear, multimodal brain-encoding model that fuses features from a speech model (Whisper) and a language model (Llama) to predict voxelwise fMRI during naturalistic listening. Instead of a classic linear mapping, the authors train a compact MLP encoder (after PCA compression) that allows cross-modal interactions, and they report clear gains in prediction accuracy over strong linear baselines and functional-connectivity benchmarks. Beyond accuracy, the nonlinear model yields more coherent cortical organization in data-driven clustering (e.g., somatomotor body-part structure, canonical ventral-visual modules, and dorsal speech pathway alignment), suggesting it captures structured spatiotemporal relationships in brain responses. Overall, the results argue that modest nonlinear fusion of audio and semantic features—without fine-tuning huge foundation models—improves both predictive performance and the interpretability of large-scale cortical organization during speech comprehension.

**Strengths:**

Timely question and clear advance: Shows that modest nonlinear, multimodal fusion (audio + semantics) can outperform strong linear speech-encoding baselines on naturalistic fMRI.

Scale: Uses a compact MLP  avoiding billion-parameter fine-tuning while still delivering sizeable gains

Captures cross-modal interactions: Architecture can model audio×semantic synergies that linear sums miss; analyses tie improvements specifically to nonlinearity, not just dimensionality reduction.

Cortical organization emerges: Nonlinear encoders yield more coherent, neuroanatomically sensible clusters (somatotopy, ventral visual modules, dorsal speech pathway), adding interpretive value beyond prediction.

**Weaknesses:**

The clustering improvement is modest. Modularity rises 0.145→0.155 (vs. FC 0.068). Stability/uncertainty metrics (split-half ARI/NMI) would help to strengthen the claim.

Because frequency and length correlate with both acoustic rate and major axes of semantic embeddings—and explain substantial variance in psycholinguistic RTs—I would recommend variance-partitioning controls in this case.  Encoding approaches like this are powerful, but it is possible that what seems to be semantic encoding instead is the encoding of quasi-semantic features such as frequency, which of course have a strong effect on the brain signal because high frequency words and concepts are easier to process.

The observation of semantic effects in somatosensory/motor cortices is intriguing, but I would urge caution in linking these results to embodied semantic memory. The semantic effect in motor region can again be largely explained by quasi-semantic features like frequency (more frequent words are easier to simulate acoustically), and have nothing to do with embodied simulation of meaning such as: I listen to the sentence "she grasped the idea in a minute" and I activated the motor neurons involved in grasping. These theories are highly controversial and this paper lacks adequate evidence to support them.  On the other hand, the argument in support to the Motor theory of speech perception is more solid and indeed the acoustic features explain much of the variance in motor regions.

While the clustering and encoding results in higher-order visual cortices are interesting, I would be cautious about interpreting them as support for convergence-zone theory or modality-independent semantic representations. First, it remains unclear which specific semantic features drive these effects. Second, activity in ventral visual areas could plausibly reflect concurrent visual mental imagery during narrative listening (hence modality-specific), or relatively automatic feedback from higher-order semantic regions (e.g., along a concreteness/abstractness dimension). I recommend softening the interpretation (e.g., “consistent with” rather than “support for”) unless additional controls demonstrate that these signals are not explained by imagery or feedback and are truly modality-invariant.

**Questions:**

Accounting for hemodynamics. Specify/justify the temporal model (FIR/HRF) used within the MLP pipeline and show that results are robust to reasonable choices (e.g., different lag windows), to rule out timing advantages unrelated to feature content.

Stability/uncertainty metrics (split-half ARI/NMI) would help to strengthen the claim of clustering improvement

I don't want to overload the authors with analysis involving precise linguistic of semantic features because this is clearly not the objective of this kind of approach, however, I would at least rule out the effect of frequency, since it is well know to be crucial for comprehension and to modulate brain activity a lot.

I will tone down or even avoid claims in support of Embodied semantics or Convergence zones unless the authors are able to provide stronger evidence.

---

> ### Author Response · Authors · 2025-12-04
>
> Thank you for the valuable feedback. We have added new content (blue in the revised PDF) and below we respond to your comments and questions.
>
> ### W3/W4. Interpretation of motor and visual semantic effects
> > “The observation of semantic effects in somatosensory/motor cortices is intriguing, but I would urge caution in linking these results to embodied semantic memory… These theories are highly controversial and this paper lacks adequate evidence to support them. On the other hand, the argument in support to the Motor theory of speech perception is more solid and indeed the acoustic features explain much of the variance in motor regions.”
> >
> > “While the clustering and encoding results in higher-order visual cortices are interesting, I would be cautious about interpreting them as support for convergence-zone theory or modality-independent semantic representations… I recommend softening the interpretation (e.g., ‘consistent with’ rather than ‘support for’) unless additional controls demonstrate that these signals are not explained by imagery or feedback and are truly modality-invariant.”
>
> Thank you for these careful comments. We agree that our original wording suggested stronger theoretical commitments than our data can fully justify, especially regarding embodied semantics in motor cortex and convergence-zone / modality-independent theories in higher-order visual cortex.
>
> For the **motor and somatosensory regions**, our goal was to highlight that both semantic and acoustic features contribute to prediction in sensorimotor areas during naturalistic listening, not to claim definitive evidence for embodied semantic memory (e.g., literal reactivation of grasp-related neurons when hearing “she grasped the idea”). In the revision:
>
> - We now explicitly acknowledge alternative explanations in the Results/Discussion, including quasi-semantic factors such as lexical frequency, predictability, and articulatory difficulty, and note that our current design cannot disentangle these from embodied-semantic accounts. (lines 427-430)
>
> - We have rephrased the relevant passages so that the emphasis is on well-supported mechanisms, such as motor/sensorimotor involvement in speech perception and production-like prediction, and we present embodied semantics as *one possible interpretation* (lines 424) rather than as something “supported” by our data.

---

> ### Author Response · Authors · 2025-12-04
>
> —--
>
> For the **higher-order visual cortices**, our intention was to connect our findings to prior work showing semantic tuning near visual cortex during both movie viewing and narrative listening, rather than to claim a definitive demonstration of modality-independent convergence zones.
>
> We would first like to clarify that the presence of semantic contributions near higher visual areas and linking them to high-level convergence zones is not entirely novel. Prior encoding work by Popham et al. (2021) used voxelwise semantic models in the same participants during two separate experiments—silent movie viewing and narrative story listening—and showed that semantic maps derived from visual and linguistic stimuli are aligned along the border of visual cortex, forming a contiguous map across modality-specific visual regions and adjacent language-driven regions. They interpreted this alignment as evidence for high-level semantic convergence zones near higher visual areas. In the revision, we now describe this work more explicitly (lines 444-448) in the results section and position our findings as **extending this line of research**—by showing that multimodal text–audio models also predict activity in overlapping high-level visual ROIs during naturalistic listening—rather than as independently proving fully modality-invariant “convergence zones.”
>
> We also soften the theoretical claims: phrases that previously read as “support for convergence-zone theory” are now rephrased as “are consistent with convergence-zone accounts” (lines 447), and we explicitly acknowledge that our auditory-only paradigm cannot rule out alternative explanations such as visual imagery or top-down feedback from higher-order semantic regions.
>
> Taken together, these changes tighten the link between our empirical results and the theoretical claims: we retain the main scientific message—that semantic information from multimodal models engages motor/somatosensory and high-level visual regions under naturalistic speech—while clearly acknowledging alternative explanations and limiting our language to “consistent with” rather than strong “support for” claims.
>
> Popham, Sara F., et al. "Visual and linguistic semantic representations are aligned at the border of human visual cortex." Nature neuroscience 24.11 (2021): 1628-1636.

---

### Author Response · Authors · 2025-12-04
**General response**

We thank the reviewers for their careful and constructive feedback. We are encouraged that all reviewers found the paper timely, clearly written, and technically solid, and we appreciate the specific ways in which they highlighted the contribution of this work.

Our paper introduces a **nonlinear, multimodal brain-encoding framework** that integrates acoustic representations from Whisper and semantic representations from LLaMA to predict voxelwise cortical responses during naturalistic speech comprehension. Unlike prior work relying on linear, unimodal mappings, our approach employs a **compact nonlinear encoder** that supports cross-modal interactions, enabling it to uncover structured variance that linear projections systematically fail to capture. We demonstrate that this nonlinear multimodal encoding yields substantial improvements in prediction accuracy (reported by Reviewer ZbwB as 17–18% normalized gains) and, more importantly, reveals **coherent cortical organization** aligned with known functional hierarchies (as emphasized by Reviewer Fs8L).

Across reviewers, several strengths were consistently noted:
- All reviewers agreed that the paper advances the field by clarifying **when and why nonlinear multimodal approaches help**, without requiring large-scale finetuning or billion-parameter modification of foundation models—an important point given the current landscape of brain–ML alignment research (Fs8L, ZbwB, JRKU).


- The nonlinear multimodal approach goes beyond improving prediction accuracy to reveal **neuroanatomically coherent cortical organization** that linear baselines fail to expose (Fs8L).


- The **methodological novelty** of disentangling nonlinearity from feature dimensionality provides a principled framework for understanding when nonlinear encoding is beneficial in speech comprehension (ZbwB).


- The work combines **substantial experimental depth** with clear exposition and theoretically meaningful interpretations (JRKU).


Below we summarize how we addressed the main points raised by the reviewers and how the revised manuscript has changed.


### 1. Motivation and positioning of nonlinearity (Reviewer ZbwB: W1)

Reviewer ZbwB asked us to clarify why nonlinearity should help in speech brain encoding and to avoid over-reliance on analogies to vision.
- We have **substantially revised the introduction** to motivate nonlinearity from **neurobiological and computational principles**, not from the fact that nonlinear models are popular in computer vision. We now emphasize that cortical language and multimodal processing are inherently nonlinear (e.g., nonlinear neural dynamics, context-dependent integration across time and modalities), and that enforcing a linear mapping is a modeling assumption rather than a ground truth.

- The vision work is now explicitly framed as highlighting a **methodological gap**: in vision, nonlinear encoders are common, whereas in speech fMRI—despite similarly complex structure—linear voxelwise mappings remain the norm, partly due to the high voxel count and temporal complexity.

- We also clarify that our goal is not to show that “nonlinear is always better,” but to identify **regimes where linear, unimodal mappings systematically leave structured variance unexplained**, especially in multimodal integration.

### 2. Interpretability vs. prediction accuracy (Reviewer ZbwB:W2)

Reviewer ZbwB emphasized the important trade-off between interpretability (linear mappings) and predictive performance (nonlinear mappings).

- We added a **dedicated discussion of this trade-off**. We explicitly state that linear models remain the preferred tool when the aim is **fine-grained attribution** of specific features (e.g., semantics vs. syntax vs. discourse).

- Our nonlinear model is positioned as **complementary**, not a replacement: when the underlying computation has strong nonlinear interactions (e.g., multimodal speech integration), strictly enforcing linearity can under-estimate these interactions. In that regime, a simple nonlinear mapping can provide a **more faithful approximation** of the computation, while linear analyses remain essential for dissecting individual feature contributions.

- We revised the first contribution bullet to de-emphasize “beating baselines” and instead highlight what the performance gains imply: **standard linear, unimodal practices leave substantial structured nonlinear multimodal variance unexplained**.

---

> ### Author Response · Authors · 2025-12-04
>
> ### 3. Statistical testing and variability (Rev. ZbwB: W3/W4)
>
> Reviewer ZbwB rightly pointed out that the original draft did not clearly present statistical significance tests and variability measures.
>
> - We added a dedicated “Statistical Testing” subsection (Appendix C) that unifies and clarifies all statistics used.
>
> - We explicitly note that voxelwise and ROI-wise analyses already used FDR-corrected significance thresholds; we now ensure this is consistently applied and clearly stated in the main text.
>
> - We introduce paired voxelwise t-tests between all model pairs (including MLP vs. MLLinear), for each subject and in a pooled analysis across subjects. These tests are Bonferroni-corrected over model pairs and visualized as heatmaps, making relative improvements fully explicit.
>
> - For each subject and model, we now report mean ± SEM across voxels for both r^2 and normalized correlation in new Appendix tables. We explain why an “across-subject SEM” is not informative with only three subjects and large between-subject offsets; instead, inference is based on within-subject voxelwise statistics, while across-subject means are reported descriptively in the main table.
>
>
>
> ### 4. Nonlinearity vs. dimensionality reduction (Reviewer ZbwB: W5)
>
> Reviewer ZbwB questioned whether the gains attributed to nonlinearity are marginal and possibly confounded by dimensionality reduction.
>
> - We clarify that the critical comparison is at **fixed PCA dimensionality** (MLLinear vs. MLP) with matched parameter counts.
>
> - We note that the effect sizes (e.g., ≈4–5% relative gains in voxelwise r² for multimodal models) are **in line with, or larger than**, improvements typically treated as meaningful in speech-encoding work, and we point to an Appendix section that contextualizes these magnitudes.
>
> - The new paired voxelwise tests show that **MLP significantly outperforms MLLinear** across all subjects and in the pooled analysis, whereas MLLinear and standard Linear baselines behave similarly and MLP without PCA overfits. Together, this supports the statement that **PCA enables tractable modeling, but it is nonlinearity that drives the additional reliable gains**.
>
>
> ### 5. Modality control and interpretation of RED (Reviewer ZbwB: W6)
>
> Reviewer ZbwB noted that RED does not provide true modality “control” or causal attribution.
>
> - We now state explicitly that **RED is a comparative error metric**, not a control or causal measure. Its role is to map *where* the multimodal model reduces error relative to the best unimodal alternative and to visualize **spatial patterns of relative advantage**, not to claim that a particular modality is uniquely responsible.
>
> - For unique vs. shared contributions, we direct readers to our **variance-partitioning analyses**, which quantify unique and shared variance attributable to text vs. audio.
>
> - We explicitly acknowledge stronger modality-control procedures from prior work (e.g., residualization, joint–best-unimodal contrasts with FDR/bootstrapping) and frame their adaptation to our nonlinear framework as a **natural direction for follow-up work**, particularly for studies whose primary goal is causal modality attribution.
>
>
> ### 6. Theoretical interpretations: motor and visual effects, quasi-semantic factors (Reviewer Fs8L: W3,W4)
>
> Reviewer Fs8L urged caution regarding embodied semantics and convergence-zone interpretations, and raised the role of quasi-semantic features such as frequency.
>
> - We have **softened the language** throughout, replacing strong claims (e.g., “support for”) with more cautious phrasing (e.g., “consistent with”) and explicitly mentioning alternative explanations such as word frequency, predictability, articulatory difficulty, imagery, and top-down feedback.
>
> - For motor/somatosensory regions, we now emphasize well-supported mechanisms (e.g., **motor involvement in speech perception and prediction**) and present embodied semantics as *one possible* interpretation rather than a conclusion.
>
> - For higher-order visual regions, we connect our results more explicitly to prior work (e.g., Popham et al., 2021) and position our findings as **extending that line of work** rather than independently proving modality-invariant convergence zones.
>
> - We note that additional controls (e.g., explicit frequency/predictability regressors, clustering stability metrics) would further refine these interpretations and identify them as **future extensions** rather than central claims of the current manuscript.

---

> ### Author Response · Authors · 2025-12-04
>
> ### 7. Temporal modeling and computational cost (Reviewer  JRKU “cost/complexity”)
>
> - In response to Reviewer  JRKU, we clarify that our nonlinear encoder operates on **PCA-compressed features** and predicts all voxels jointly. The number of parameters scales with the feature dimension, not with voxel count, so the practical training cost is **comparable to or lower than** voxelwise linear regression on full voxel sets.
> - We mention that the same architecture could, in principle, incorporate additional modalities given appropriate datasets, but we intentionally focus on speech to isolate the role of acoustic–semantic interactions in this work.
>
> ---
>
> Overall, we believe these revisions substantially strengthen the paper: they provide a clearer and more principled motivation for nonlinearity, a nuanced treatment of interpretability vs. prediction accuracy, rigorous statistical testing and variability reporting, a more precise treatment of modality contributions, and appropriately cautious theoretical interpretations. At the same time, the core message emphasized by all reviewers remains unchanged: **a compact nonlinear multimodal encoder, applied to fixed text and audio representations, reveals robust and spatially organized cortical structure that linear unimodal mappings systematically miss.**
>
> We hope the Area Chair will view the revised manuscript as a solid, timely, and conceptually grounded contribution to modeling how the brain integrates acoustic and linguistic information during naturalistic speech comprehension.

---

### Meta-Review · Area_Chair_8gSU · 2026-01-12

**Summary:**

This paper proposes a nonlinear multimodal fMRI encoding model that fuses Whisper audio and LLaMA semantic features via a compact MLP, reporting ~15–17% gains over linear baselines and more coherent cortical clustering. Reviewers agree the work is clearly written and technically competent, with strengths in careful model comparisons and analyses suggesting that nonlinear multimodal interactions can improve prediction and reveal structured cortical organization (Fs8L, ZbwB).

However, several substantive weaknesses limit the strength of the claims. A central concern is insufficient control and interpretability: multiple reviewers note that semantic effects—particularly in motor and higher-order visual cortices—may be explained by quasi-semantic confounds such as lexical frequency or imagery, and the paper does not adequately rule these out (Fs8L). Relatedly, the RED analysis does not provide proper modality control, making conclusions about modality-specific contributions fragile (ZbwB). Reviewers also question the motivation and positioning of nonlinear encoding for neuroscience, arguing that gains in prediction do not clearly translate into improved mechanistic insight, especially given limited discussion of the trade-off with interpretability (ZbwB). Finally, despite revisions, concerns remain about the robustness and generality of the reported improvements given the small number of subjects and modest clustering gains (Fs8L).

Overall, while promising, the paper’s claims about semantic organization and theoretical implications are not sufficiently supported for acceptance at ICLR.

**Reviewer Concerns:**

The submission faces several critical weaknesses that preclude acceptance. A primary concern raised by Reviewer ZbwB is the trade-off between prediction accuracy and interpretability; they noted that "nonlinearity adds complexity and obscures the interpretability objective," making it difficult to isolate which specific linguistic properties drive neural activity. Furthermore, Reviewer Fs8L highlighted significant theoretical overreach, arguing that the results regarding "embodied semantics" and "convergence zones" lack adequate evidence and could be explained by quasi-semantic features like word frequency.

While the authors provided statistical updates in the rebuttal, the decision to reject is based on the limited scientific gain beyond predictive metrics. The modest improvement in cortical clustering and the lack of robust modality controls (as noted regarding the RED metric) suggest that the current framework does not yet provide the necessary neuroscientific insights required for a high-impact contribution.

**Reviewer Scores:**

It is hard to tell. The authors did address several of the concerns raised by the reviewers.

---

### Decision · Program_Chairs · 2026-01-26

Reject